# Mitochondrial RNA modifications shape metabolic plasticity in metastasis

Sylvain Delaunay[1], Gloria Pascual[2], Bohai Feng[3,4], Kevin Klann[5], Mikaela Behm[1], Agnes Hotz-Wagenblatt[1], Karsten Richter[1], Karim Zaoui[3], Esther Herpel[6,7], Christian Münch[5], Sabine Dietmann[8], Jochen Hess[1,3], Salvador Aznar Benitah[2,9] & Michaela Frye[1 ✉]

Aggressive and metastatic cancers show enhanced metabolic plasticity[1], but the precise underlying mechanisms of this remain unclear. Here we show how two NOP2/Sun RNA methyltransferase 3 (NSUN3)-dependent RNA modifications— 5-methylcytosine (m⁵C) and its derivative 5-formylcytosine (f⁵C) (refs. [2–4])—drive the translation of mitochondrial mRNA to power metastasis. Translation of mitochondrially encoded subunits of the oxidative phosphorylation complex depends on the formation of m⁵C at position 34 in mitochondrial tRNA$^{Met}$. m⁵C-deficient human oral cancer cells exhibit increased levels of glycolysis and changes in their mitochondrial function that do not affect cell viability or primary tumour growth in vivo; however, metabolic plasticity is severely impaired as mitochondrial m⁵C-deficient tumours do not metastasize efficiently. We discovered that CD36-dependent non-dividing, metastasis-initiating tumour cells require mitochondrial m⁵C to activate invasion and dissemination. Moreover, a mitochondria-driven gene signature in patients with head and neck cancer is predictive for metastasis and disease progression. Finally, we confirm that this metabolic switch that allows the metastasis of tumour cells can be pharmacologically targeted through the inhibition of mitochondrial mRNA translation in vivo. Together, our results reveal that site-specific mitochondrial RNA modifications could be therapeutic targets to combat metastasis.

Metastasis is the main cause of cancer-related deaths[5]. As a multistep process, metastasis begins with the invasion of tumour cells through the basement membrane and intravasation into the surrounding vasculature or lymphatic system, and ends with colonization at secondary tumour sites[5]. To successfully metastasize, tumour cells must dynamically adapt to the constantly changing microenvironment[1]. Metabolic plasticity allows tumour cells to survive in adverse conditions, including hypoxia and starvation, in particular during proliferation-independent processes such as dissemination from the primary tumour[1,6].

Mitochondria are bioenergetic, biosynthetic and signalling organelles that are integral to stress sensing. Because mitochondria allow fast cellular adaptations to environmental cues, they are important mediators of most aspects of tumorigenesis[7]. For example, mitochondrial translation efficiency is vital for controlling cytosolic protein homeostasis and nuclear stress signalling, and thereby directly determines cellular lifespan[8]. However, the precise molecular mechanisms that underlie how human tumour cells rapidly adjust the balance between mitochondrial and glycolytic energy production remain largely unclear.

The mitochondrial metabolic pathway of oxidative phosphorylation (OXPHOS) contains over 100 proteins. The mitochondrion itself translates 13 OXPHOS subunits. To translate these essential subunits of the respiratory chain complex, the mitochondrial genome contains 22 tRNAs that get modified at 137 positions by 18 types of RNA modifications[9]. The function of these RNA modifications is to determine the accuracy and optimal rate of translation[10]. The mitochondrial protein synthesis machinery differs in many ways from translation in the cytoplasm. For instance, human mitochondria only use one tRNA (tRNA$^{Met}_{CAU}$) to read AUG and AUA codons as methionine[11]. To decipher AUA as methionine, mitochondrial tRNA$^{Met}$ contains the RNA modification f⁵C at the wobble position of the anticodon (position 34). As f⁵C34 enables tRNA$^{Met}$ to recognize AUA as well as AUG codons, the modification regulates both translation initiation and elongation efficiency[2,12]. The biogenesis of f⁵C34 is initiated through the formation of m⁵C by NSUN3, and is completed by ALKBH1 (refs. [2–4,13]). Loss-of-function mutations in the human *NSUN3* gene are linked to deficiencies in the mitochondrial respiratory chain complex, which are caused by severe defects in mitochondrial translation[4]. The functions of mitochondrial tRNA modifications in cancer are at present largely unknown.

Here we show that mitochondrial cytosine-5 RNA methylation is essential for the dynamic regulation of mitochondrial translation rates, and thereby shapes metabolic reprogramming during metastasis. We

[1]German Cancer Research Center – Deutsches Krebsforschungszentrum (DKFZ), Heidelberg, Germany. [2]Institute for Research in Biomedicine (IRB Barcelona), The Barcelona Institute of Science and Technology (BIST), Barcelona, Spain. [3]Department of Otolaryngology, Head and Neck Surgery, University Hospital Heidelberg, Heidelberg, Germany. [4]Department of Otorhinolaryngology, The Second Affiliated Hospital of Zhejiang University School of Medicine, Hangzhou, China. [5]Institute of Biochemistry II, University Hospital, Goethe University Frankfurt, Frankfurt am Main, Germany. [6]Institute of Pathology, University Hospital Heidelberg, Heidelberg, Germany. [7]NCT Tissue Bank, National Center for Tumor Diseases (NCT), Heidelberg, Germany. [8]Washington University School of Medicine in St. Louis, St. Louis, MO, USA. [9]Catalan Institution for Research and Advanced Studies (ICREA), Barcelona, Spain. ✉e-mail: M.Frye@dkfz.de

reveal that tumour cells with low mitochondrial levels of $m^5C$ reduce OXPHOS and rely on glycolysis for energy production instead. However, the reverse metabolic switch from glycolysis to OXPHOS promotes tumour cell invasion and metastasis. Only cancer cells that have high levels of $m^5C$ and $f^5C$, to enhance mitochondrial translation rates and fuel OXPHOS, invade the extracellular matrix and disseminate from primary tumours. Preventing the formation of $m^5C$ and pharmacological inhibition of mitochondrial translation both inhibit metastasis in vivo. Together, our study shows that mitochondrial RNA modifications regulate the metabolic reprogramming that is required for the invasion and dissemination of tumour cells from primary tumours.

## Results

### Cytosine modifications in mitochondrial tRNA[Met]

$f^5C$ is a unique tRNA modification in mammalian mitochondria that derives directly from $m^5C$ (ref.[9]). To simultaneously detect $f^5C$ and $m^5C$ at single-nucleotide resolution, we performed chemically assisted bisulfite sequencing (fCAB-seq), a method that uses *O*-ethylhydroxylamine to protect $f^5C$ sites in bisulfite conversion protocols[14] (Extended Data Fig. 1a). Out of all mitochondrial encoded tRNAs, only two showed consistently high levels of modification in normal and cancer cells (Fig. 1a and Extended Data Fig. 1b,c). As expected, one site corresponded to cytosine 34 (C34) in the 'wobble position' of mitochondrial (mt)-tRNA[Met] (Fig. 1a,b). The second site corresponded to mt-tRNA[Ser2], which does not carry $f^5C$ but contains three NSUN2-mediated consecutive $m^5C$ sites in the variable loop[15] (Extended Data Fig. 1d). Thus, our data confirm that tRNA[Met] is the only mitochondrial tRNA that carries $f^5C$.

To quantify $f^5C$ and $m^5C$ modifications individually, we next performed bisulfite RNA sequencing (RNA-seq) alongside fCAB-seq. The levels of $f^5C$ are calculated by subtracting bisulfite-protected cytosines from *O*-ethylhydroxylamine-protected sites (Fig. 1c–j). The most prevalent modification in mitochondrial tRNA[Met] was $m^5C$ (more than 50%), followed by $f^5C$ (around 35%) and unmodified cytosines (less than 10%) in all tested cell lines (Fig. 1c–j). To inhibit the formation of both tRNA[Met] modifications, we depleted the methyltransferase NSUN3 in four oral squamous cell carcinoma (OSCC) cell lines using two different short hairpin RNAs (shRNAs) (Extended Data Fig. 1e). Parallel bisulfite RNA-seq and fCAB-seq revealed an increase of around eightfold in unmodified cytosines at position C34 when NSUN3 was depleted (Fig. 1k–n and Extended Data Fig. 1e–h). Notably, the $f^5C$ modification was virtually absent in mt-tRNA[Met] in NSUN3-depleted cells (Fig. 1k–n).

NSUN3 and ALKBH1 form an enzymatic cascade to synthesize $f^5C34$ (refs. [2–4,13]), but we did not detect equal stoichiometry of $m^5C$ and $f^5C$ (Fig. 1c–j). To test whether the expression of ALKBH1 was compromised in cancer cells, we measured the RNA levels of *NSUN3* and *ALKBH1* (Extended Data Fig. 1i). Although the RNA levels of both enzymes increased in cancer cells, the expression of *NSUN3* was more variable (Extended Data Fig. 1i). In addition, *ALKBH1* expression was largely unaffected when *NSUN3* was depleted (Extended Data Fig. 1j). ALKBH1 and NSUN3 protein levels correlated in cancer cell lines[16] (Cancer Cell Line Encyclopedia) (Extended Data Fig. 1k), and a significant correlation of expression was also present in oesophageal carcinoma and head and neck squamous cell carcinoma (HNSCC) datasets, as well as the corresponding normal tissues (Extended Data Fig. 1l,m). We conclude that NSUN3 and ALKBH1 are co-expressed, but that their enzymatic activity or import into mitochondria may be differently regulated.

Together, our data show that depletion of NSUN3 causes a robust reduction of both $m^5C$ and $f^5C$ at C34 in mitochondrial tRNA[Met].

### $m^5C$ modulates mitochondrial function

As mt-tRNA[Met] is needed for both translation initition and elongation of mitochondrial mRNA, we asked whether loss of $m^5C$, and consequently also $f^5C$, affected global rates of mitochondrial translation.

We confirmed that there was a significant decrease of nascent protein synthesis in NSUN3-depleted mitochondria by quantifying the incorporation of *O*-propargyl-puromycin (OP-puro) into nascent peptide chains (Fig. 2a,b and Extended Data Fig. 2a,b). Thus, hypomethylation of mt-tRNA[Met] downregulates the protein levels of mitochondrially encoded genes[2–4].

To test how the mitochondrial metabolism adapted to the downregulation of protein synthesis, we quantified metabolites of the tricarboxylic acid (TCA) cycle using mass spectrometry (Extended Data Fig. 2c). We measured a slight reduction in most TCA metabolites when the expression of NSUN3 was reduced (Fig. 2c). To confirm that $m^5C$ directly regulated mitochondrial activity, we overexpressed a wild-type (WT) or an enzymatic dead (MUT) version of the NSUN3 protein[4] (Extended Data Fig. 2d–f). Similar to NSUN3 depletion, TCA metabolite levels were also lower in methylation-deficient cells (Fig. 2c). Consequently, maximal cellular respiration was reduced when mt-tRNA[Met] was hypomodified (Fig. 2d,e and Extended Data Fig. 2j,k), and the basal extracellular acidification rate (ECAR) was higher overall in NSUN3-deficient cancer cells (Fig. 2f and Extended Data Fig. 2g–i,l–o). The differences in mitochondrial metabolism were not driven by increased cell death, altered mitochondrial DNA copy number or enhanced production of reactive oxygen species (ROS) (Extended Data Fig. 3a–c). Instead, the cancer cells directly rebalanced their overall ATP production towards glycolysis (Fig. 2g).

Mitochondria adapt their shape and structure to sustain their cellular functions[17]. Accordingly, we measured a significant reduction in mitochondria length in cells that overexpressed methylation-deficient NSUN3 (Extended Data Fig. 3d,e). Similarly, the morphology of mitochondria in NSUN3-deficient cells appeared more circular and the number of cristae per mitochondrion was decreased (Fig. 2h–k and Extended Data Fig. 3f–k). We conclude that the RNA modifications $m^5C$ and $f^5C$ in mt-tRNA[Met] act as a sensor of cellular energy requirements and adapt mitochondrial functions accordingly (Fig. 1l).

### Mitochondrial $m^5C$ regulates metastasis

Owing to their integral role in stress sensing, mitochondria have been implicated in most aspects of tumorigenesis[7]. To investigate which stages of tumorigenesis involved $m^5C$, we performed orthotopic transplantation assays into host mice[18]. We transplanted three NSUN3-deficient human metastatic OSCC lines: SCC25 and the patient-derived lines VDH01 and VDH15 (ref. [18]; Fig. 3a and Extended Data Fig. 4a). In addition, we overexpressed the wild-type (WT) or enzymatic dead (MUT) NSUN3 protein to identify methylation-dependent cellular functions during tumorigenesis[4] (Fig. 3a).

All human cancer lines formed tumours with 100% penetrance and the vast majority (around 75%) formed metastases (Supplementary Table 1). Metastases into both the lymph nodes and lungs were consistently decreased in tumours that lacked a functional NSUN3 protein (Fig. 3a,b, Extended Data Fig. 4a,b and Supplementary Table 1). On average, inhibition of mitochondrial $m^5C$ formation reduced the development of lymph node metastasis from 80% to 20% (Fig. 3b, Extended Data Fig. 4b and Supplementary Table 1). However, primary tumour growth was largely unaffected (Extended Data Fig. 4d–f). To fully exclude the possibility that differences in metastases were driven by primary tumour size, we normalized the dimension of the secondary tumour to its matching primary tumour. This ratio was consistently lower when NSUN3 was depleted or inhibited (Fig. 3c and Extended Data Fig. 4c). We conclude that mitochondrial tRNA[Met] modifications at C34 are required for efficient tumour metastasis, but not for primary tumour development and growth.

### $m^5C$ and $f^5C$-deficient tumours are glycolytic

To understand how tumour formation was affected by reduced levels of $m^5C$ in mitochondria, we histologically examined the primary tumours. Consistent with our observation that inhibition of

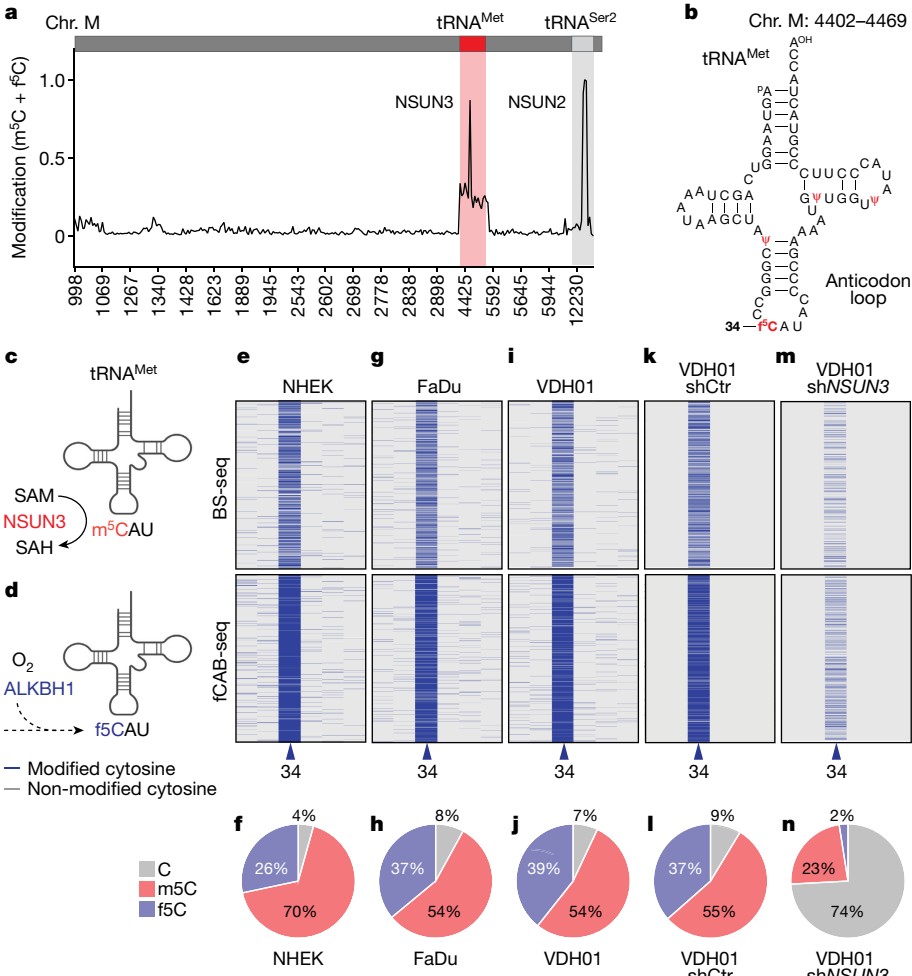

**Fig. 1 | Detection of m⁵C and f⁵C in mitochondrial tRNAs. a**, Detection of m⁵C and f⁵C sites in the mitochondrial tRNA transcriptome using fCAB-seq in cancer cells (VDH01). Plotted are all cytosines with a coverage of more than 100 in both independent replicates. The two peaks correspond to C34 of mt-tRNA$^{Met}$ mediated by NSUN3 and C47, C48 and C49 in mt-tRNA$^{Ser2}$ mediated by NSUN2, respectively. **b**, Mitochondrial tRNA$^{Met}$ secondary structure with all known modifications highlighted in red. **c,d**, Schematic overview of m⁵C (**c**) and f⁵C (**d**) formation. SAM, *S*-adenosyl-L-methionine; SAH, *S*-adenosyl-L-homocysteine. **e**–**n**, Heat maps (**e,g,i,k,m**) of mt-tRNA$^{Met}$ centred on position C34 and adjacent cytosines showing modified (blue) and unmodified (grey) cytosines from gene-specific parallel fCAB-seq and bisulfite sequencing (BS-seq) in the indicated cell lines; and quantification (**f,h,j,l,n**) of cytosines that are unmodified, m⁵C- or f⁵C-modified as shown in the heat maps (average of three sequencing reactions per condition).

methylation enhanced glycolysis in vitro, NSUN3-depleted tumours also exhibited an increase in glycolysis and upregulation of glucose transporter 1 (GLUT1) in vivo (Fig. 3d,e and Extended Data Fig. 4g). As cancer cells readily use glycolysis for energy production, even when oxygen is available (the Warburg effect), it was unsurprising that primary tumour growth was unaffected by the loss of NSUN3. Accordingly, NSUN3-depleted primary tumours were histologically highly similar to controls (Extended Data Fig. 4h–t).

To identify mitochondria-driven gene expression signatures that correlated with metastasis, we transcriptionally profiled the primary tumours expressing or lacking NSUN3 (Extended Data Fig. 5a–f and Supplementary Table 2). We identified a set of 1,708 transcripts that were commonly deregulated in NSUN3-depleted tumours (Extended Data Fig. 5g). As expected, the differentially expressed transcripts were enriched for genes that encode regulators of mitochondrial activity such as NADH dehydrogenase, oxidoreductase and electron transfer (Extended Data Fig. 5h). The most significantly enriched Gene Ontology (GO) category comprised genes that encode structural constituents of both mitochondrial and cytoplasmic ribosomes (GO:0003735; Extended Data Fig. 5h). Notably, all identified ribosomal regulators were downregulated in the absence of NSUN3

(Extended Data Fig. 5i), confirming that global protein homeostasis had adapted to low levels of protein synthesis in the mitochondria. Principal component analyses using all transcribed mitochondrial genes (*n* = 1158; MitoCarta2.0) clearly separated the tumours according to NSUN3 expression (Extended Data Fig. 5j), an effect that was highly reproducible and independent of the shRNA (Extended Data Fig. 5k).

We next assessed how the mitochondrion itself was affected by NSUN3 depletion. RNAs that encode proteins of the OXPHOS pathway were the most repressed in the absence of NSUN3 (Fig. 3f,g). All mitochondrially encoded proteins that form complex I, III, IV and V were downregulated (Fig. 3g). The only exception was complex II, which does not contain mitochondrially encoded subunits and was therefore not directly affected by the depletion of NSUN3 (Fig. 3g). Gene set enrichment analysis (GSEA) further confirmed a decrease in the expression of OXPHOS regulators, but also identified positive correlations with regulators of hypoxia and inhibitors of metastasis (Fig. 3h).

Thus, our data show that the transcriptional profile of NSUN3-depleted tumours is driven by mitochondrial activity, and that hypomodified mitochondrial tRNA$^{Met}$ metabolically reprograms primary tumours.

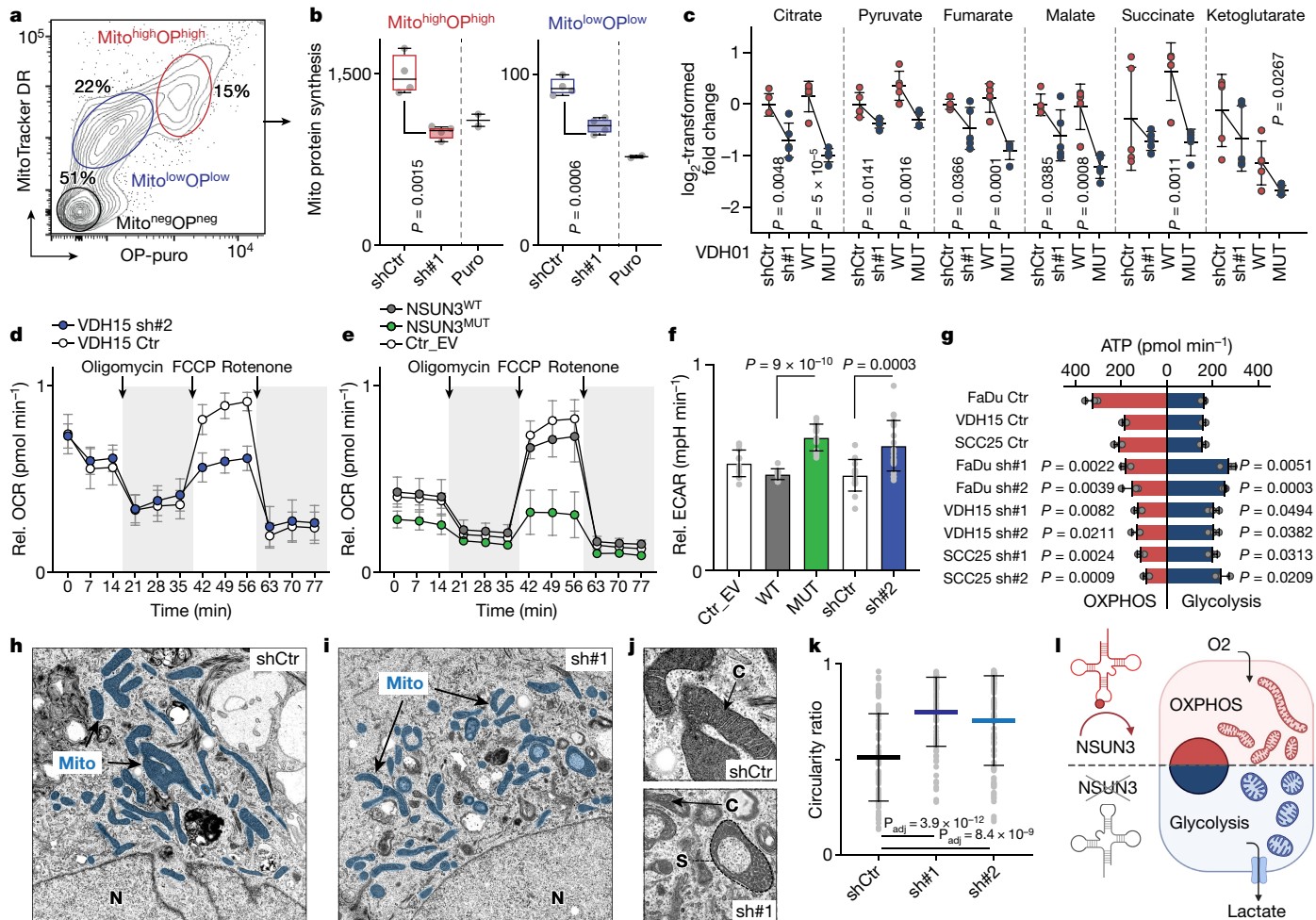

**Fig. 2 | Mitochondrial m⁵C controls energy metabolism in tumour cells.**
**a**, Representative flow cytometry plot using MitoTracker DR (Mito) and OP-puro (OP) to isolate mitochondria. **b**, Quantification of mitochondrial protein synthesis in the cell populations shown in **a** infected with control shRNA (shCtr) or *NSUN3* shRNA (sh#1), or treated with puromycin (Puro) (shCtr, sh#1: n = 4 flow sorts; Puro: n = 2 treatments). Box plots show minimum value, first quartile, median, third quartile and maximum value. **c**, Log₂-transformed fold change of normalized TCA metabolite levels (pmol per 10⁶ cells) (n = 5 mass spectrometry runs per condition). **d**, Oxygen consumption rate (OCR) in VDH15 cells infected with Ctr or *NSUN3* shRNAs (sh#2) (shCtr: n = 5 injections; sh#2: n = 7 injections). **e**, OCR in VDH15 cells infected with empty vector (Ctr_EV) or with constructs overexpressing wild-type (WT) or mutant (MUT) NSUN3 proteins (Ctr_EV, NSUN3^WT: n = 4 injections; NSUN3^MUT: n = 6 injections). **f**, Quantification of basal extracellular acidification rate (ECAR) in

VDH15 cells infected with shRNAs (shCtr, sh#2) or constructs overexpressing wild-type or mutant NSUN3 proteins. The empty vector (Ctr_EV) served as a control (Ctr_EV, WT: n = 12; MUT: n = 18; shCtr: n = 15; sh#2: n = 21 injections). **g**, Metabolic flux analysis quantifying mitochondrial and glycolytic ATP production in FaDu, VDH15 and SCC25 cells infected with shRNAs (shCtr, sh#1 and sh#2; n = 3 injections). **h–j**, Electron microscopy of VDH01 cells infected with shCtr (**h**) or sh#1 (**i**). Higher magnifications are shown in **j**. C, cristae; Mito, mitochondria; N, nucleus; S: structure (representative images from 10 cells per condition; 2 infections). **k**, Relative circularity ratio of mitochondria in VDH01 cells infected with shCtr, sh#1 or sh#2 (Ctr: n = 91; sh#1: n = 86; sh#2: n = 93 mitochondria). **l**, Metabolic switch induced by loss of m⁵C in mt-tRNA^Met. Data are mean ± s.d. (**c–g,k**). Unpaired two-tailed *t*-test (**b,c,f,g**) or two-sided Šídák's test (**k**). Exact *P* values are indicated.

## Mitochondrial m⁵C fuels invasion

To determine which tumour cell population relied most on high mito-chondrial translation levels, we labelled primary tumour sections for the mitochondrial markers MTCO1 and MTCO2. Both markers co-localized with CD44 and were most highly expressed in the basal layers of the tumours (Extended Data Fig. 6a–d). The basal layer contains undifferentiated, proliferating cells but also initiates invasion during metastasis. To study whether cell proliferation or invasion was affected by inhibition of mitochondrial m⁵C formation, we subjected the OSCC cells to three-dimensional (3D) culture systems (Fig. 4a).

All spheres (tumoroids) were comparable in size, even when NSUN3 was depleted or m⁵C formation inhibited, which confirms that cell proliferation was largely unaffected (Extended Data Fig. 6e–l). Similar to our findings in 2D-cultured cells and tumours formed in vivo, the spheres also switched to glycolysis in the absence of NSUN3, as shown

by enhanced uptake of glucose (Extended Data Fig. 6m–p). To visualize mitochondrial activity in the tumoroids, we measured the mitochon-drial membrane potential (MMP) through incorporation of the dye MitoTracker CMXROS. The MMP was highest in a small number of cells at the surface of the tumoroids (Fig. 4b and Extended Data Fig. 7a–k), and those cells disappeared in the absence of NSUN3 (Fig. 4c,d).

Mitochondria are involved in both the migration and the invasion of tumour cells[19]. Therefore, we tested the capacity of the tumoroids to disseminate cells for invasion by embedding them into a 3D collagen I matrix[20] (Fig. 4e,f). Invading leader cells showed an upregulation of the MMP and increased mitochondrial lengths (Fig. 4f–i). Both depletion and inhibition of NSUN3 significantly reduced the number of leader cells per tumoroid (Fig. 4j–l and Extended Data Fig. 7l–n). Thus, high levels of mitochondrial tRNA^Met modifications promoted tumour cell invasion.

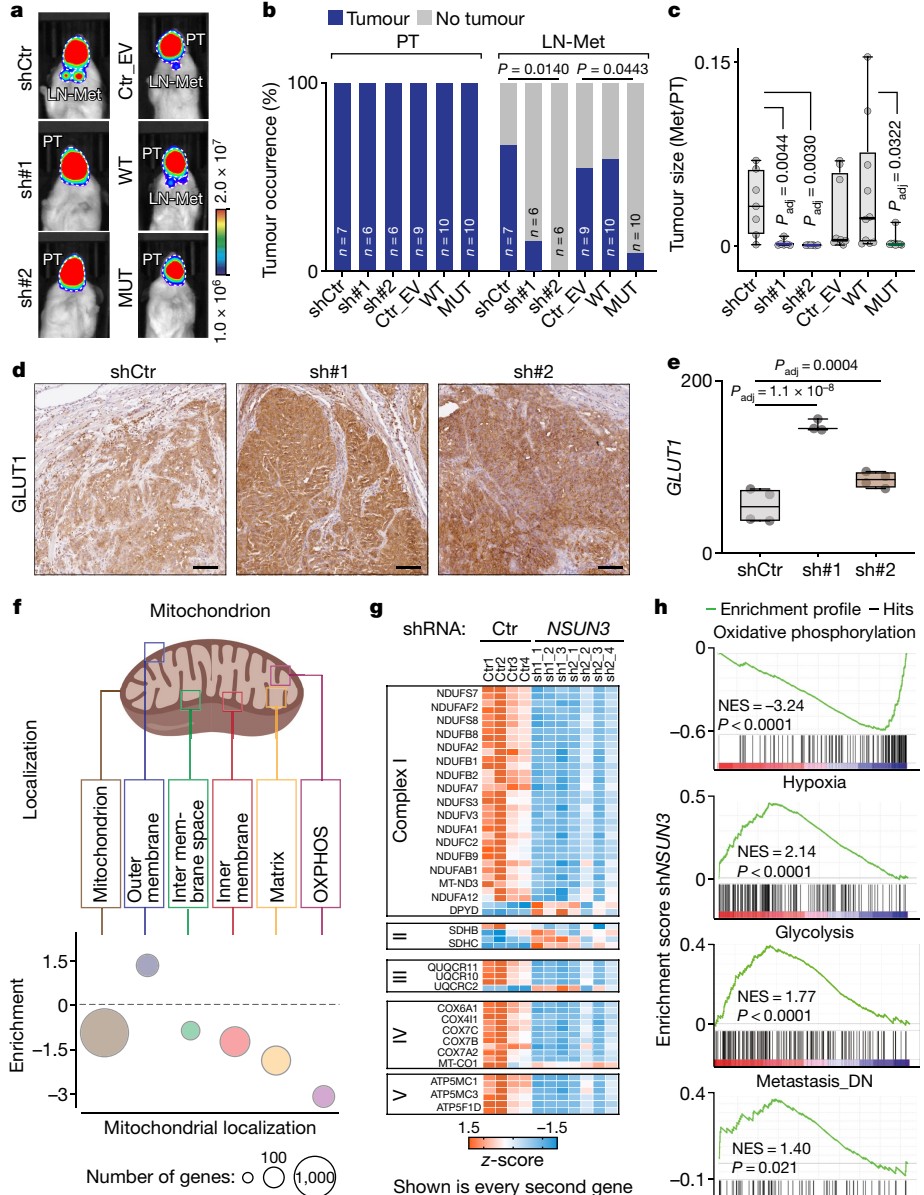

**Fig. 3 | Mitochondrial m⁵C is required for metastasis. a,b**, Bioluminescence imaging (**a**) and tumour occurrence (**b**) of primary tumours (PT) and lymph node metastases (LN-Met) 21 days after orthotopic transplantation into the mouse tongue. Tumours derived from VDH01 cells were infected with control shRNA (shCtr) or *NSUN3* shRNA (sh#1, sh#2) (left) or with an empty vector control (Ctr_EV) or wild-type or mutant NSUN3 overexpression constructs (right). **c**, Dimensions of the metastasis (LN-Met) relative to its matching primary tumour derived from VDH01 cells in the indicated conditions (shCtr: *n* = 7 mice; sh#1, sh#2: *n* = 6 mice; Ctr_EV: *n* = 9 mice; WT, MUT: *n* = 10 mice). **d,e**, Protein (**d**) and RNA (**e**) levels (read counts) of GLUT1 in VDH15-derived tumours transduced with shCtr or *NSUN3* shRNA (sh#1, sh#2) (protein:

representative images from 3 mice per condition; RNA: shCtr, sh#2: *n* = 4 mice or tumours; sh#1: *n* = 3 mice or tumours). Scale bars, 50 μm. **f**, Illustration of mitochondrial compartments (top) and GSEA showing the normalized average enrichment score of mitochondrial regulators in sh*NSUN3* cells in the respective compartments (bottom). **g**, Heat map using *z*-scores of differentially expressed (*P* ≤ 0.05) transcripts from the indicated complexes of the electron transport chain. **h**, GSEA of shCtr versus sh*NSUN3* tumour cells. DN, down; NES, normalized enrichment score. Box plots in **c,e** show minimum value, first quartile, median, third quartile and maximum value. Chi-squared test (**b**), two-sided Dunnett's test (**c**) or Wald test (**e**). Random permutations (**h**). Exact *P* values are indicated.

## m⁵C regulates CD36-driven metastasis

A subpopulation of non-dividing tumour cells co-expressing the cell-surface markers CD44 and CD36 has been identified as metastasis-initiating cells in human oral carcinoma[18]. As CD36 is located on the outer mitochondrial membrane[21], we next tested whether the CD44- and CD36-expressing population correlated with mitochondrial functions. To measure the mitochondrial activity in CD44⁺CD36⁺ tumour cells, we first sorted cancer cells for high (H)

or low (L) expression of CD44 and CD36 (Fig. 5a,b), and then quantified the MMP in the different subpopulations. CD44^H CD36^H cells were consistently the population with the highest MMP (Fig. 5b,c and Extended Data Fig. 8a–c). Accordingly, expression of *NSUN3* and the mitochondrial regulators *MT-CO1* and *TFAM* were also upregulated in the CD44^H CD36^H population (Fig. 5d and Extended Data Fig. 8d,e). Other highly expressed markers included the cell adhesion protein integrin-α6 (*ITGA6*) and the regulator of epithelial-to-mesenchymal transition *SLUG* (Extended Data Fig. 8e).

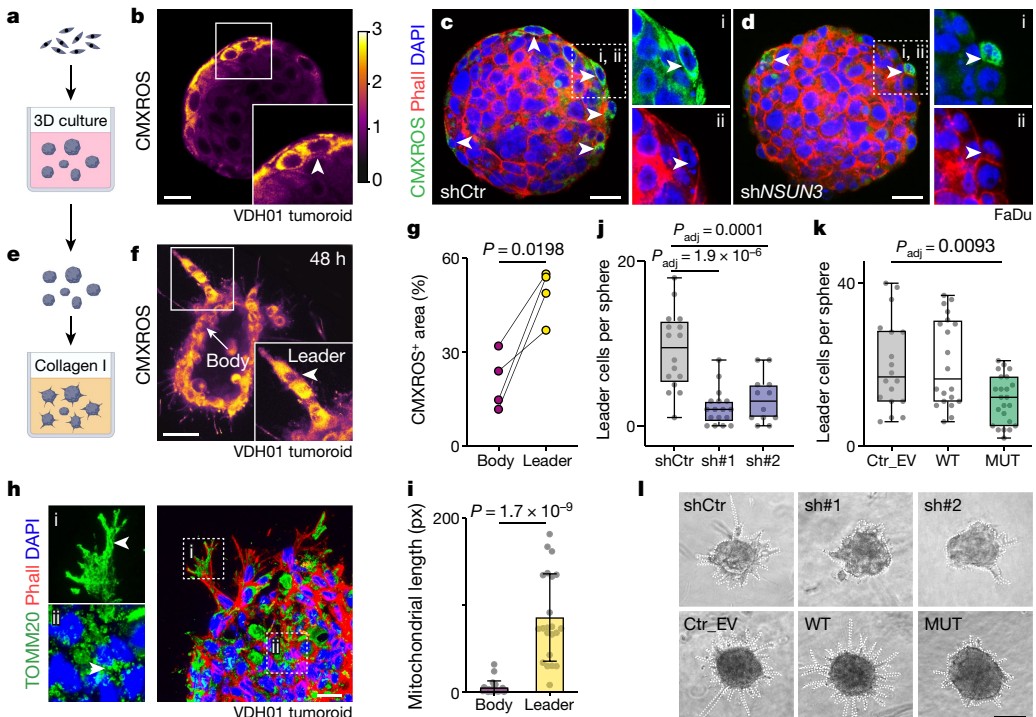

**Fig. 4 | Efficient mitochondrial translation promotes invasion. a**, Scheme of 3D cultures. **b**, Intensity of MitoTracker (CMXROS) in VDH01 tumoroids. White square, magnified area. Arrowhead, measured area. **c,d**, CMXROS and phalloidin staining in FaDu tumoroids expressing shCtr (**c**) or sh*NSUN3* (**d**) constructs. Dotted squares, magnified areas i and ii (right). **e**, Invasion assay: tumoroids were placed into a 3D collagen matrix. **f**, VDH01 tumoroid labelled for CMXROS after 48 h cultured in collagen I. White square, magnified area. Arrowhead, invading leader cells. **g**, Quantification of the CMXROS-positive area in the VDH01 sphere body compared to leader cells after 48 h in culture ($n = 4$ spheres from 4 independent experiments). **h**, Sectioned VDH01-derived invading tumoroids labelled for TOMM20 (green) and phalloidin (red) after 48 h cultured in collagen I. Dotted squares, magnified areas i and ii (left). Arrowhead, mitochondria. **i**, Quantification of the mitochondrial length in leader and sphere body cells from sectioned VDH01-derived invading tumoroids (body: $n = 23$; leader $n = 22$ mitochondria from 9 cells of 3 casted tumoroids). Data are mean ± s.d. **j–l**, Quantification of leader cells per tumoroid (**j,k**) and images of representative VDH01 tumoroids at 9 days (**l**) in invasion assays infected with control shRNA (shCtr) or sh*NSUN3* (sh#1, sh#2) (**j**) or with empty vector control (Ctr_EV), wild-type or mutant NSUN3 overexpression constructs (**k**) (shCtr: $n = 16$, sh#1: $n = 17$, sh#1: $n = 12$, Ctr_EV: $n = 17$, WT: $n = 20$, MUT: $n = 23$ tumoroids from 3 independent experiments). DAPI: nuclear counterstain (**c,d,h**). Representative pictures from a minimum of 3 tumoroids (**b–d,f,h**). Scale bars, 20 μm (**b,h**); 30 μm (**c,d**); 40 μm (**f**); 50 μm (**l**). Box plots in **j,k** show minimum value, first quartile, median, third quartile and maximum value. Paired two-tailed *t*-test (**g**), unpaired two-tailed *t*-test (**i**) or two-sided Šídák's test (**j,k**). Exact *P* values are indicated.

Next, we asked whether mitochondrial m⁵C and f⁵C levels maintained the metastasis-initiating CD44^H CD36^H cell population. Knockdown of NSUN3 was sufficient to downregulate the mRNA levels of both *CD44* and *CD36* (Extended Data Fig. 8f,g). The number of CD44^H-CD36^H cells was about threefold lower in the absence of NSUN3 (Fig. 5e and Extended Data Fig. 8h–j). Similarly, overexpression of the methylation-deficient but not the wild-type NSUN3 protein reduced the number of CD44^H CD36^H cells by more than fourfold (Extended Data Fig. 8k–m). Finally, we also analysed orthotopically transplanted tumours and their matching lymph node metastases for CD44^H CD36^H cells (Fig. 5f). Both depletion and inhibition of NSUN3 reduced the number of CD44^H CD36^H cells by threefold (Fig. 5g). Notably, the lymph node metastases and their matching primary tumours contained a comparable number of CD44^H CD36^H cells, indicating that the formation of the double-positive metastasis-initiating cell population was reversible (Fig. 5h). Together, our data show that depletion of the mitochondrial RNA modifications m⁵C and f⁵C is sufficient to diminish the metastasis-initiating cell population.

### Proteome of metastasis-initiating cells
The mitochondrial translation rate is intricately linked to nuclear gene expression and cytosolic mRNA translation, and this cross-talk is required to integrate mitochondrial activity into the cellular context[22]. To study how mRNA translation was altered in CD36-dependent metastasis-initiating cells, we quantified nascent protein synthesis in all three subpopulations (CD44^H CD36^H, CD44^H CD36^L and CD44^L-CD36^L) in tumoroids using a comparative quantitative proteomics approach[23] (Fig. 5i and Supplementary Table 3). Hierarchical clustering of the subpopulation's proteome segregated newly translated mRNAs into seven different clusters (Fig. 5j and Supplementary Table 4). Only cluster 3 contained mRNAs for which the translation rates progressively increased, with the rate being highest in CD44^H CD36^H cells (Fig. 5j). Proteins involved in regulating mitochondrial activity were significantly enriched in cluster 3 (Fig. 5k and Extended Data Fig. 9a–e). Cluster analyses using all translated mitochondrial genes (MitoCarta2.0) present in all replicates and conditions ($n = 656$) confirmed a consecutive translational upregulation of mitochondria-related proteins, with metastasis-initiating cells (CD44^H CD36^H) having the highest translation rates (Fig. 5l). Thus, CD36-driven metastasis-initiating cells are defined by a metabolic translatome that promotes mitochondrial respiration.

Our data so far have linked mitochondrial activity to the CD44^H CD36^H metastasis-initiating tumour cells, but it is unclear whether the downregulation of CD36 is a cause or a consequence of m⁵C- and f⁵C-dependent mitochondrial metabolic reprogramming. To directly test whether the reduction of the NSUN3-deficient metastasis-initiating population was due to the downregulation of CD36 rather than the loss of the mitochondrial RNA modifications, we stably overexpressed CD36 in NSUN3-depleted cancer cells (Extended Data Fig. 9f–j). Overexpression of CD36 was not sufficient to rescue oxygen consumption rates in NSUN3-depleted cells or alter tumoroid growth (Extended

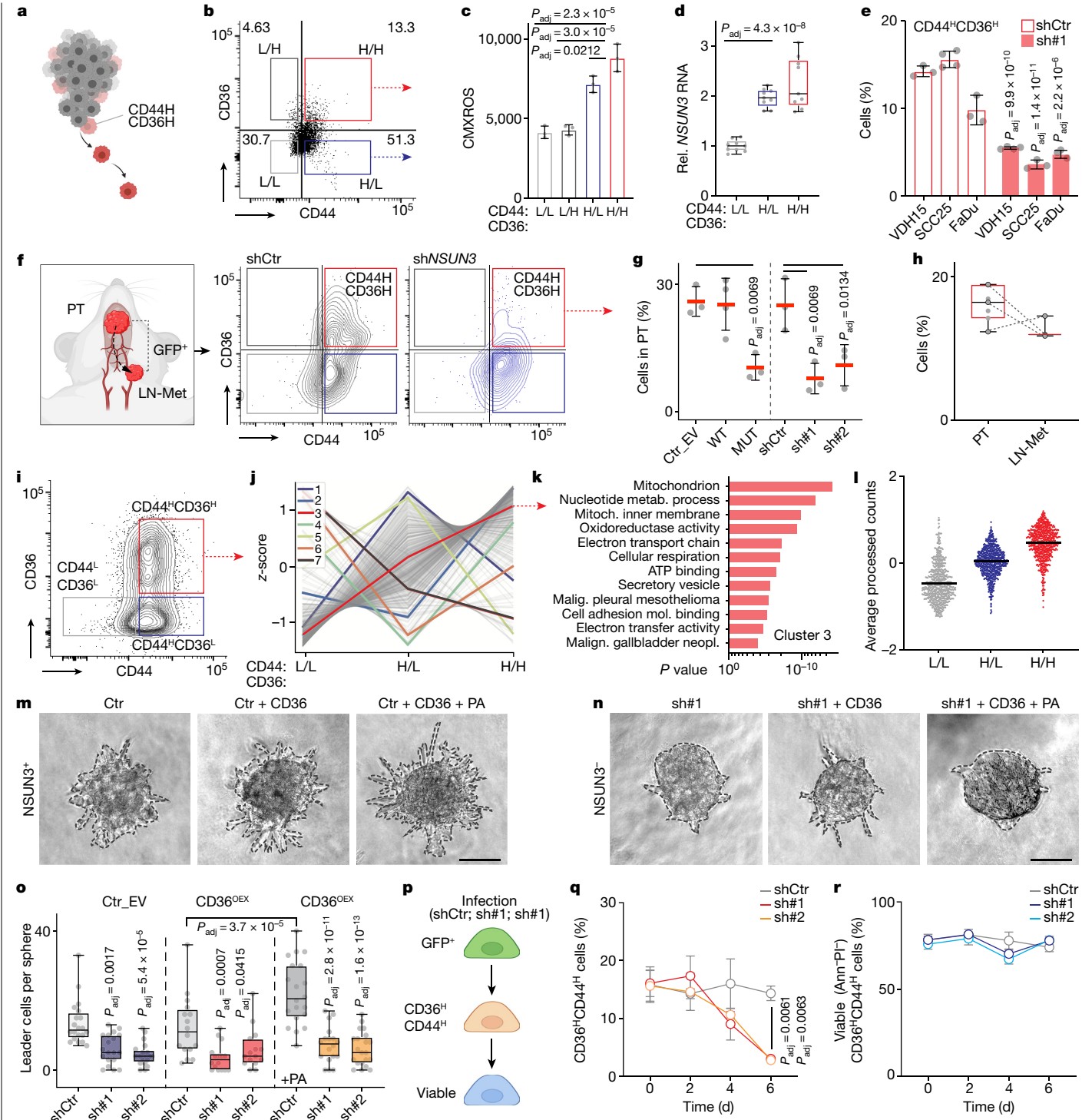

**Fig. 5 | Metastasis-initiating cells require methylated mitochondrial RNA.**
**a**, Metastasis-initiating CD44⁺CD36⁺ cells. **b,c**, Flow cytometry (**b**) and CMXROS quantification (**c**) of CD36 and CD44 high (H) and low (L) VDH01 cells (n = 3 sorts). **d**, *NSUN3* RNA in FaDu subpopulations (n = 9; 3 quantitative PCR with reverse transcription (RT–qPCR) runs; 3 infections). **e**, CD44ᴴCD36ᴴ population in VDH15, SCC25 and FaDu tumoroids. shCtr, control; sh#1, *NSUN3* shRNA (shCtr VDH15, shCtr FaDu, sh#1 SCC25, sh#1 FaDu: n = 3 flow sorts; shCtr SCC25, sh#1 VDH15 n = 4 flow sorts). **f**, GFP⁺ primary tumours (PT) and lymph node metastases (LN-Met) (left) and flow cytometry (middle and right) of primary tumours 21 days after transplantation. **g**, CD44ᴴCD36ᴴ cells in infected primary tumours (VDH01): empty vector (Ctr_EV), wild-type NSUN3, mutant NSUN3 or shCtr, sh#1 and sh#2 (Ctr_EV, MUT, shCtr, sh#1, sh#2: n = 3 mice; WT: n = 4 mice). **h**, CD44ᴴCD36ᴴ cells in primary tumours and matching LN-Met (VDH15) (PT: n = 7 mice; LN-MET: n = 3 mice). **i**, CD44 and CD36 flow cytometry using VDH01 tumoroids for translatome analyses. **j**, Unsupervised clustering of nascent protein synthesis levels. **k**, GO analysis (ToppGene) of cluster 3 (**j**). **l**, Cluster analyses (ClustVis; averaged processed counts) of translated MitoCarta2.0 genes (n = 656) (n = 3 flow sorts; 3 infections). **m–o**, Tumoroids (**m,n**) and quantification (**o**) of leader cells per tumoroid (VDH01) in invasion assays infected as indicated or overexpressing (OEX) CD36, untreated or treated with 30 µM of palmitic acid (+PA) (shCtr, n = 20; sh#1, n = 19; sh#2, n = 20; shCtr + CD36ᴼᴱˣ, n = 16; sh#1 + CD36ᴼᴱˣ, n = 16; sh#2 + CD36ᴼᴱˣ, n = 15; shCtr + CD36ᴼᴱˣ + PA, n = 20; sh#1 + CD36ᴼᴱˣ + PA, n = 18; sh#2 + CD36ᴼᴱˣ + PA, n = 21 tumoroids; 3 independent experiments). **p–r**, Illustration (**p**) and quantification (**q**) of CD44ᴴCD36ᴴ cells and their viability (**r**) (n = 3 infections). Ann, Annexin V; PI, propidium iodide. Scale bars, 50 µm (**m,n**). Data in **c,e,g,q,r** are mean ± s.d. Box plots in **d,h,o** show minimum value, first quartile, median, third quartile and maximum value. Two-sided Šídák's test (**c–e,g,o**) or Dunnett's test (**q,r**). Random sampling of whole genome (**k**). Exact P values are indicated.

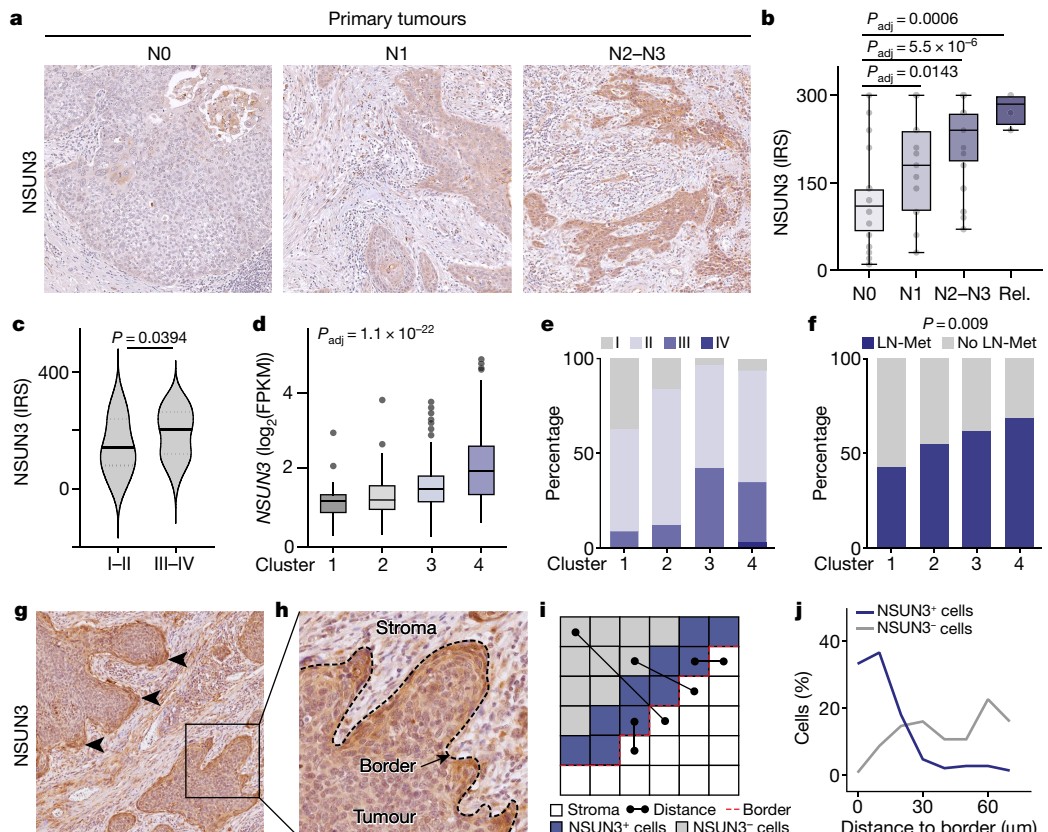

**Fig. 6 | An NSUN3-driven gene signature is predictive for metastasis in patients with HNSCC. a,b,** Representative immunohistochemistry (**a**) and quantification (**b**) of NSUN3 protein expression in primary tumours from patients with HNSCC classified by pathological N-stage at diagnosis with no (N0: $n = 28$), one (N1: $n = 19$) or several (N2–N3: $n = 32$) metastases, or patients who have relapsed (Rel.: $n = 4$ relapse tumours from 3 patients). IRS, immunoreactivity score. **c,** Quantification of NSUN3 protein expression (IRS) for the indicated pathological stages (I–II: $n = 25$ patients, III–IVa–b: $n = 52$). Violin plot shows median with quartiles. **d,** Unsupervised cluster analyses identified four clusters of patients with HNSCC (TGCA) according to *NSUN3*-related gene expression (cluster 1, $n = 141$; cluster 2, $n = 127$; cluster 3, $n = 174$; cluster 4, $n = 58$ patients). FPKM, fragments per kilobase million.

**e,f,** Frequency of the indicated pathological stages (**e**) and lymph node metastases (LN-Met) (**f**) in clusters identified in **d. g,h,** Immunohistochemistry for NSUN3 in primary HNSCC with metastases. Arrowheads, cells at the tumour–stroma border. Black square, higher magnification shown in **h.** Dotted line with arrow, tumour–stroma border. **i,** Strategy to quantify the Euclidian distance of NSUN3⁺ and NSUN3⁻ cells from the tumour–stroma border. **j,** Proportion of NSUN3⁺ and NSUN3⁻ cells at the indicated distance to the tumour–stroma border ($n = 5$ patients). Representative staining from 5 patients (**g,h**). Box plots in **b,d** show minimum value, first quartile, median, third quartile and maximum value. Two-sided Šídák's test (**b**), two-tailed unpaired *t*-test (**c**), ordinary one-way ANOVA (**d**) or chi-squared test (**f**). Exact *P* values are indicated.

Data Fig. 9k–m). Thus, CD36 signalling required NSUN3-dependent mitochondrial changes.

As expected, overexpression of CD36 significantly induced the formation of leader cells in invading tumoroids when the cells were pre-treated with palmitate to activate CD36 signalling ($P_{adj} = 3.7 \times 10^{-5}$) (Fig. 5m,o and Extended Data Fig. 9n). However, CD36-driven activation of invasion depended on the presence of NSUN3 (Fig. 5n,o). In conclusion, upregulation of CD36 signalling was not sufficient to rescue the invasion processes in NSUN3-depleted cells.

In summary, CD36 signalling requires the mitochondrial RNA modifications m⁵C and f⁵C for palmitate-induced invasion of tumour cells. Moreover, reducing mitochondrial RNA modification levels is sufficient to diminish the metastasis-inducing tumour cell population.

## Metabolic reprogramming is dynamic

Our data have shown that low levels of m⁵C and f⁵C repress mitochondrial translation and thereby force tumour cells to use glycolysis as energy source, and that this metabolic pathway allows cell growth, but prohibits cell invasion and metastasis. However, the fate of NSUN3-depleted cells during invasion is unclear. To test whether CD36 signalling in the absence of NSUN3 simply causes cell death, we quantified the number of viable CD44ᴴCD36ᴴ cells after depletion

of NSUN3 in a time-course experiment (Fig. 5p–r and Extended Data Fig. 9o). We measured a significant decrease in CD44ᴴCD36ᴴ cells six days after NSUN3 depletion (Fig. 5q), but the number of viable cells in the CD44ᴴCD36ᴴ population remained unchanged (Fig. 5r). Thus, the lack of metabolic plasticity in NSUN3-deficient cells inhibited CD36-dependent invasion but it did not cause cell death.

We conclude that the metastasis-initiating CD44ᴴCD36ᴴ population is dynamically formed within the primary tumour as a result of metabolic events that require high levels of tRNAᴹᵉᵗ RNA modifications.

## NSUN3-driven gene signature in cancer

To assess the predictive value of mitochondrial m⁵C levels for metastasis in patients with HNSCC, we stained 78 tumour samples for NSUN3 (ref. [24]). Expression of NSUN3 protein was highest in primary tumours from patients with regional lymph node metastases at the time of diagnosis, and was associated with higher pathological staging of the primary tumours (Fig. 6a–c, Extended Data Fig. 10a,b and Supplementary Table 5). The levels of NSUN3 protein expression were comparable in primary tumours and lymph node metastases (Extended Data Fig. 10c,d), which corroborates our finding that orthotopically transplanted tumours and their matching lymph node metastasis also had similar proportions of CD36ᴴCD44ᴴ cells (Fig. 5h).

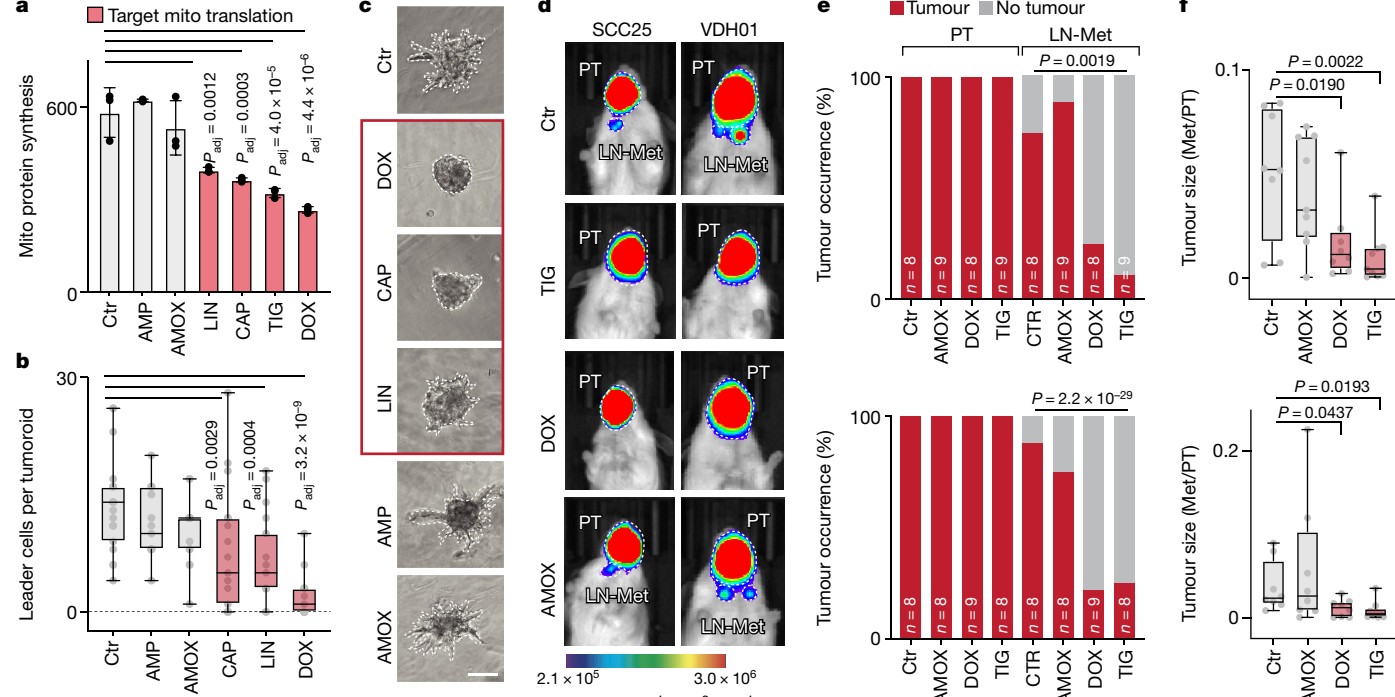

**Fig. 7 | Pharmacological inhibition of mitochondrial translation prevents metastasis. a**, Quantification of mitochondrial (Mito) protein synthesis using OP-puro in VDH01 tumour cells treated with the indicated antibiotics or a vehicle control (Ctr) ($n = 3$ drug treatments). Data are mean ± s.d. **b,c**, Quantification of invading leader cells (**b**) and representative bright field images of tumoroids (**c**) after exposure to the indicated antibiotics or control (Ctr, $n = 19$; AMP, $n = 15$; AMOX, $n = 15$; CAP, $n = 19$; LIN, $n = 19$; DOX, $n = 19$ tumoroids from 3 independent drug treatments). **d,e**, Bioluminescence imaging of SCC25 (left) and VDH01 (right) tumours (**d**) and quantification of tumour occurrence (**e**) in mice with orthotopically transplanted SCC25 (top) or VDH01 (bottom) tumours treated with the indicated antibiotics or phosphate-buffered saline (PBS) as a control (Ctr) for 8 days. **f**, Dimension of the lymph node metastasis relative to its matching primary tumour (PT) of SCC25 (top) or VDH01 (bottom) tumours treated with PBS (Ctr) or the indicated antibiotics (SCC25 Ctr, $n = 8$ mice; AMOX, $n = 9$ mice; DOX, $n = 8$ mice; TIG, $n = 9$ mice; VDH01 CTR, TIG, AMOX, $n = 8$ mice; DOX, $n = 9$ mice). Box plots in **b,f** show minimum value, first quartile, median, third quartile and maximum value. Two-sided Šídák's test (**a,b**), chi-squared test (**e**) or unpaired two-tailed *t*-test (**f**). Exact *P* values are indicated.

To define the clinical features of mitochondrial-driven transcriptional profiles, we identified all differentially expressed genes that correlated with the levels of *NSUN3* in patients with HNSCC (The Cancer Genome Atlas (TCGA); $n = 500$) (Extended Data Fig. 10e and Supplementary Table 6). As expected, the NSUN3-driven signature was enriched for genes that encode regulators of metastasis and hypoxia (Extended Data Fig. 10f and Supplementary Table 7). Unsupervised clustering grouped the patients into four clusters according to the NSUN3-driven gene signature (Fig. 6d and Extended Data Fig. 10g). Both the pathological stage of these cancers and the presence of lymph node metastasis at the time of diagnosis progressively increased with *NSUN3* levels (Fig. 6e,f).

To confirm our observation in tumoroids that metastasis-initiating cells needed high mitochondrial activity for invasion (Fig. 3), we labelled primary tumour sections from patients with lymph node metastasis for NSUN3. NSUN3 protein levels were highest at the invasive front of the tumours (Fig. 6g,h). When we measured the Euclidian distance of NSUN3^high and NSUN3^low cells from the tumour–stroma border, we found that the number of NSUN3^high cells sharply increased at around a 30-μm distance to the border (Fig. 6h–j). Together, our data indicate that the activity of NSUN3 is higher close to the tumour–stroma border, where the activation of invasion occurs.

In summary, the NSUN3-driven gene signature is prognostic for lymph node metastasis and for higher pathological stage in patients with HNSCC. Thus, we have identified a mitochondria-driven gene signature that predicts lymph node colonization and progression-free survival in head and neck cancer.

## Mitochondria regulate metastasis

Small-molecule inhibitors of mitochondrial transcription reduce tumour growth in xenografts of human cancer cells[25], and mitochondrial protein synthesis has central roles in cancer development and progression[26]. In stark contrast to these wide-ranging mitochondrial functions in cancer, inhibiting mitochondrial mRNA translation by targeting m5C and f5C in tRNA^Met exclusively reduced tumour metastasis without affecting cell viability or primary tumour initiation and growth. Therefore, we sought to exclude that other unknown functions of NSUN3 contributed to tumour cell invasion and metastasis.

If regulating mitochondrial translation by modifying mt-tRNA^Met is the only function of NSUN3, then we should be able to recapitulate all observed NSUN3-dependent cellular functions by specifically inhibiting mitochondrial translation without affecting cytoplasmic protein synthesis. Because mitochondrial and prokaryotic protein synthesis machineries are highly similar, several classes of antibiotics such as glycylcyclines, oxazolidinones and amphenicols also target mitochondrial ribosomes[27,28] (Extended Data Fig. 11a). Indeed, only treatment of cancer cells with linezolid (LIN), chloramphenicol (CAP), tigecycline (TIG) or doxycycline (DOX) repressed mitochondrial protein synthesis, reduced the oxidative phosphorylation capacity and increased the extracellular acidification rate (Fig. 7a and Extended Data Fig. 11b–d). Ampicillin (AMP) and amoxicillin (AMOX) both target the bacterial cell wall, and mitochondrial translation and functions were unaffected by those antibiotics (Fig. 7a and Extended Data Fig. 11d).

To study whether repression of mitochondrial translation affected tumour cell invasion similarly to depletion or inhibition of NSUN3, we

exposed OSCC tumoroids to the antibiotics and measured their invasive capacity. Only treatment with CAP, LIN, TIG and DOX reduced the number of invading leader cells in tumoroids and reduced glucose uptake (Fig. 7b,c and Extended Data Fig. 11e–h). Moreover, the expression of CD36 protein and the number of CD36-dependent metastasis-initiating tumour cells (CD44[H]CD36[H]) were reduced about twofold when the cells were exposed to CAP, DOX, LIN and TIG (Extended Data Fig. 11i–m). As described for the depletion of NSUN3, cell viability was unaffected by exposure to the antibiotics (Extended Data Fig. 12a,b).

Finally, we confirmed that treatment with the selected antibiotics also reduced metastases in vivo. We injected two OSCC lines into the tongues of host mice, waited seven days for establishment of primary tumours and then treated the mice daily with TIG, DOX or AMOX. Only treatment with TIG and DOX decreased the number of lymph node metastases from 80% to 20%, whereas AMOX-treated and vehicle-treated tumours showed similar metastasis capacities (Fig. 7d,e and Supplementary Table 1). As described for the inhibition of m[5]C formation, the reduction of lymph node metastases was not driven by primary tumour size when mitochondrial translation was inhibited (Fig. 7f and Extended Data Fig. 12c). In conclusion, inhibition of mitochondrial translation fully recapitulated the loss of a functional NSUN3 protein by preventing cell invasion and reducing the number of CD36-dependent metastasis-initiating tumour cells in vitro and in vivo.

In summary, our data reveal that mitochondrial tRNA modifications regulate mitochondrial translation rates and thereby drive the metabolic reprogramming that is required for metastasis. Moreover, we identify the inhibition of m[5]C formation in mitochondria as a therapeutic opportunity to prevent the dissemination of tumour cells from primary tumours.

## Discussion

Here we show that the dynamic adjustment of mitochondrial RNA modification levels directly contributes to tumour malignancy by promoting metastasis. In contrast to normal tissues, tumours rely to a large extent on aerobic glycolysis to meet the energy demands required for cell division and growth[29]. However, tumours also need a high degree of metabolic plasticity when reacting to cues and stresses in their local environment, which causes heterogeneity in the metabolic status of tumour cells[1]. Although the oxidative phosphorylation pathway strictly depends on both cytosolic and mitochondrial protein synthesis machineries[30], we reveal that reducing the levels of m[5]C in mitochondrial RNA is sufficient to prohibit the metabolic switch from glycolysis to OXPHOS. The consequence of the loss of metabolic plasticity in tumour cells is a low metastatic capacity. We further reveal that CD36-dependent metastasis-initiating cells require mitochondrial m[5]C to activate invasion and dissemination from the primary tumour.

Upregulation of CD36 enhances the uptake of fatty acids for lipid homeostasis but can also fuel mitochondrial respiration under stress conditions[31]. CD36 expression correlates with poor patient survival in various types of cancer[32,33], and when expressed together with CD44 initiates metastasis[18]. We show that the CD44- and CD36-expressing tumour cells only efficiently metastasize when using mitochondria as an energy source. Disrupting mitochondrial responses by depleting m[5]C is sufficient to reduce tumour cell dissemination. We show that m[5]C and f[5]C levels in mt-tRNA[Met] are rate-limiting for the translation of mitochondrially encoded subunits of the OXPHOS complex, thus inhibiting the metabolic plasticity that is required for CD36-dependent metastasis.

We further show that a mitochondria-driven gene signature has predictive value for patients with head and neck cancer. The expression of genes that correlate with high levels of NSUN3—and therefore high levels of m[5]C—predicted lymph node metastases and poor patient prognosis. We propose that mitochondrial RNA-modifying enzymes should be added to the growing list of RNA-modifying anticancer drug targets. NSUN3 is a highly promising drug target because as a stand-alone enzyme, it is solely responsible for mitochondrial m[5]C formation.

A direct regulatory role for mitochondrial RNA modifications in determining tumour cell behaviour is notable. Although mitochondrial and cytosolic translation are known to be rapidly, dynamically and synchronously regulated, it is widely assumed that cytosolic translation processes control mitochondrial translation unidirectionally[34]. For example, invasive breast cancer cells rely on the transcription coactivator PGC-1α (peroxisome proliferator-activated receptor-γ coactivator-1α) to enhance oxidative phosphorylation and mitochondrial biogenesis to undergo metastasis[35,36]. However, it has become increasingly clear that successful metastasis requires reversible metabolic changes that increase the cell's capacity to withstand oxidative stress[37–39]. Moreover, oxidative metabolism is sufficient to drive immortalization in *Drosophila* brain tumours[40].

Certain antibiotics are used as adjuvant agents in the treatment of cancers because of their anti-proliferative, pro-apoptotic or anti-epithelial-to-mesenchymal transition capabilities, but their precise mode of action and effect on cancer therapies is controversial[41]. Our study shows that distinct classes of antibiotics have specific effects on tumours. Antibiotics that target mitochondrial translation inhibit metastasis but do not affect tumour cell viability and growth. Similar to our findings in head and neck cancer, several studies have repurposed antibiotics to prevent OXPHOS-dependent ATP production and have also shown that therapy resistance and tumour-initiating cells can be eradicated by exposure to imatinib and/or tigecycline in leukaemia and melanoma[27,42–44].

Both treatment strategies—inhibition of mitochondrial translation and targeting NSUN3—will, however, not target cancer cells specifically. Long-term systemic inhibition of mitochondrial translation will have side effects. Patients who have a loss-of-function mutation in the *NSUN3* gene survive, yet present with combined mitochondrial respiratory chain complex deficiency[4]. Antibiotics that target mitochondrial translation are often used in the long term and the side effects are well-documented. However, we did not observe any reported side effects in animals treated with the antibiotics, such as swelling of the face or muzzle, skin rashes or diarrhoea.

Metastasis is the major cause of death in patients with cancer. Blocking the dissemination of tumour cells from primary tumours is one approach to stop the successful colonization of tumour cells at distant sites and thus prevent relapse. We propose that the inhibition of mitochondrial RNA modifications is a promising therapeutic opportunity to stop the spread of cancer cells in later stages of tumour development.

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

## Methods

### Cell lines and culture conditions

The squamous cell carcinoma cell lines SCC25 and FaDu were obtained from ATCC (https://www.lgcstandards-atcc.org). The patient-derived lines VDH01 and VDH15 were generated as described in a previous report[18]. Patient samples to generate VDH01 and VDH15 were provided by the Vall d'Hebron University Hospital Biobank (PT17/0015/0047) integrated in the Spanish National Biobanks Network with written informed consent from all participants. The samples were processed following standard operating procedures with the appropriate approval of the Ethical and Scientific Committees. The study followed the guidelines of the Declaration of Helsinki, and patient identity and pathological specimens remained anonymous in the context of the study.

All cells were cultured in a humidified incubator at 37 °C with 5% $CO_2$. SCC25 (ATCC CRL-168TM), FaDu (ATCC HTB-43), VDH01 and VDH15 cells were grown in keratinocyte serum-free medium (K-SFM, Gibco) supplemented with 5 µg ml$^{-1}$ penicillin–streptomycin, 0.025 mg ml$^{-1}$ bovine pituitary extract and 0.2 µg ml$^{-1}$ hEGF. FaDu (ATCCR HTB-43) cells were grown in EMEM (LONZA) supplemented with 5 µg ml$^{-1}$ penicillin–streptomycin and 10% fetal bovine serum (Gibco). LentiX 293T cells grown in Dulbecco's modified Eagle's medium (DMEM) including 10% FBS were used for lentivirus production, after transfection with Lipofectamine 2000 (Thermo Fisher Scientific), according to the manufacturer's instructions. All cells tested negative for mycoplasma contamination.

*NSUN3* shRNA plasmids were obtained from Dharmacon (SMART vector; V3SVHS00_5546488: TTACAAATTCATGTCACCA and V3SVHS00_7499725: TATAGAACAAACACCATCT). Site-directed mutagenesis to generate the *NSUN3*-mutant construct was performed by mutating the nucleotide 640 from T to G and the nucleotide 641 from G to C. Full-length cDNA constructs for NSUN3 wild type (WT) or mutant (MUT) in the pLenti-C-mGFP-P2A-Puro vector were obtained from OriGene. VDH01 cells were transfected with the CD36 wild-type construct CMV-mCherry-CD36-C-10 (#55011, Addgene) using Lipofectamine 2000 (Thermo Fisher Scientific) according to the manufacturer's instructions.

Cells in 2D or 3D culture were treated for 24 h with tigecycline (Sigma-Aldrich, 10 µM), doxycycline (Sigma-Aldrich, 10 µg ml$^{-1}$), linezolid (Sigma-Aldrich, 45 µM), chloramphenicol (Sigma-Aldrich, 300 µg ml$^{-1}$), amoxicillin (Sigma-Aldrich, 10 µg ml$^{-1}$) and ampicillin (Sigma-Aldrich, 10 µg ml$^{-1}$).

### fCAB and bisulfite sequencing and analyses

Total RNA was isolated from OSCC cells using Trizol, according to the manufacturer's instructions (Thermo Fisher Scientific) and DNase treated. For detection of m$^5$C and f$^5$C in mitochondrial tRNAs, tRNA from total RNA was isolated using the Mirvana kit, as described by the manufacturer (Thermo Fisher Scientific). *O*-ethylhydroxylhamine and bisulfite treatment, library preparation and sequencing were performed as previously described[14].

For targeted gene-specific bisulfite sequencing, RNA isolation, bisulfite conversion reaction and sequencing, approximately 1–2 µg of RNA samples was used[45]. For identification of f$^5$C modification, the RNA was pre-exposed to 10 mM of *O*-ethylhydroxylamine in 100 mM MES buffer (pH 5.0), for 2 h at 37 °C, followed by bisulfite treatment. For m$^5$C, the RNA was bisulfite-treated by mixing with 42.5 µl 40% sodium bisulfite solution (pH 5.0) and 17.5 µl DNA protection buffer supplied with the EpiTect Bisulfite Kit (Qiagen). The reaction mixture was then incubated for three cycles of 5 min at 70 °C, followed by 1 h at 60 °C on a thermal cycler. To desalt the reaction, all samples were passaged through Micro Bio-Spin 6 chromatography columns, following the manufacturer's instructions (Bio-Rad). The samples were then desulfonated by adding an equal volume of 1 M Tris (pH 9.0) to the reaction mixture and incubating for 1 h at 37 °C. Bisulfite-treated RNA samples

were then precipitated overnight with 2.5 volumes of 100% ethanol, 0.1 volumes of 3 M sodium acetate (pH 5.5) and 1–2 µl Glycoblue (AM9516; Ambion) at −80 °C. For cDNA synthesis, reverse transcription reactions were carried out with the SuperScript III Reverse Transcriptase kit (Thermo Fisher Scientific), following the manufacturer's instructions. Five hundred nanograms of bisulfite-converted RNA was used. A gene-specific primer (Fw: AGTAAGGTTAGTTAAATAAGTT) was used in the reverse transcription reaction.

The cDNA was PCR-amplified using primers specific for deaminated sequences (Fw: AGTAAGGTTAGTTAAATAAGTT, Rv: TAATACAA AAAAAATATAACCA). The PCR products were separated from unincorporated primers using low-melting agarose gels using a Gel Extraction Kit (Qiagen) for products in between 50 and 90 bp (amplicon at ±70 bp). Illumina sequencing libraries from the converted tRNA amplicons were generated using the NEBNext Ultra II DNA Library Prep Kit (NEB, #E7645) and indexed using the Multiplex Oligos for Illumina Index Primers Set 1 (NEB, E7335). To enable sequencing of the low-complexity tRNA libraries the final equimolar tRNA amplicon pool was multiplexed in a 1:1 volume ratio with a high-complexity library generated from fragmented human gDNA and sequenced on a MiSeq v2 using the Nano kit in PE100 bp mode.

Raw sequencing fastq files were trimmed with 'TrimGalore!' of the adapter, retaining only reads with a minimum length of 25 nt. Reads were aligned to the GRCh38 (hg38) reference genome using Bismark (v.0.22.3) with the '--non_directional' option and default parameters. mtRNA genomic coordinates for the GRCh38 reference genome were obtained from the ENSEMBL database. Original targeted sequencing reads and their multiple sequence alignments on mtRNAs were extracted from sorted Bismark alignment (bam) files using the R packages RSamtools and GenomicAlignments. Multiple sequence alignments and heat maps were generated with 'matrixplot' from the R package VIM.

### Protein extraction and western blotting

Cells were first rinsed twice with PBS and lysed in ice-cold RIPA buffer (50 mM Tris-HCl (pH 7.4), 1% NP-40, 150 mM NaCl, 0.1% SDS and 0.5% sodium deoxycholate per ml RIPA per T-75 or 100 mm culture dish). RIPA was supplemented with complete Mini EDTA-free Protease Inhibitor Cocktail tablets (Roche), and cells were collected using a cell scraper. The lysates were centrifuged and their supernatant collected and kept on ice. The concentration of each protein sample was assessed using the Pierce BCA Protein Assay kit (Thermo Fisher Scientific) according to the manufacturer's instructions and measured using a spectrophotometer.

Cell protein lysates were mixed with Laemmli buffer (2×) and run on separating polyacrylamide gels and transferred to a nitrocellulose or PVDF membrane (GE Healthcare). Membranes were blocked for a minimum of 1 h at room temperature in 5% (w/v) non-fat milk 10% (w/v) in 1× TBS and 0.1% Tween-20 (TBS-T) (48.4 g Tris Base, 160 g NaCl, and $H_2O$ to 1 L; for 20× TBS buffer at adjusted pH 7.6) and incubated with the primary antibody in blocking solution overnight at 4 °C. Each membrane was washed three times for 10 min in TBS-T before incubation with the appropriate horseradish peroxidase (HRP)-labelled secondary antibody (1:10,000) in TBS-T at room temperature for 1 h (anti-mouse NA931 and anti-rabbit IgG HRP; Millipore). Finally, the membranes were washed as before and the antibodies were detected by using the Amersham ECL Prime Western Blotting Detection Reagent (GE Healthcare). The primary antibodies used were NSUN3 (1:500, GTX46175, GenTex), MTCO1 (1:1,000, ab91317, Abcam), MTCO2 (1:1,000, PA5-26688, Thermo Fisher Scientific) and HSP90 (1:1,000, sc-13119, Santa Cruz).

### RNA isolation and RT–qPCR

Total RNA was prepared using Trizol (Thermo Fisher Scientific) and further purified using TURBO DNase treatment (Thermo Fisher Scientific) according to the manufacturer's instructions. Double-stranded cDNA was synthesized from 1 µg of RNA using Superscript III reverse

transcriptase (Thermo Fisher Scientific), following the manufacturer's instructions with Random Primers (Promega). Each RT–qPCR reaction was set up using predesigned probes: NSUN3, ALKBH1, CD44, CD36 or keratin 10 probes (Hs00222961_m1, Hs00195696_m1, Hs01075864_m1, Hs00354519_m1, Hs00166289_m1). A human 18S rRNA probe (Hs99999901_s1) was used for normalization using the ΔCt method. RT–qPCR and data acquisition were conducted using the QuantStudio qPCR machine (Applied Biosystems).

### Mitochondrial DNA copy number and ROS determination

Total cellular DNA was isolated from OSCC cells using the DNeasy Blood and Tissue kit (Qiagen) according to the manufacturer's instructions. Mitochondrial DNA copy number was determined by qPCR using a mtDNA monitoring primer set kit (7246, Takara).

For measurement of mitochondrial ROS levels, MitoSOX was used according to the manufacturer's instructions. In brief, culture cells were incubated with MitoSOX reagent (2 µM; Thermo Fisher Scientific) for 30 min at 37 °C. After incubation, cells were washed twice and resuspended in PBS. The fluorescence of each sample was measured by the BD LSRFortessa Analyzer (BD Biosciences). Data were further processed by FlowJo software. Fluorescence measurements were visualized by histogram, and the raw fluorescence median values were extracted for quantification.

### Analysis of mitochondrial protein synthesis

To investigate protein synthesis, OSCC cells were treated with OP-puro as previously described[46,47]. Reconstituted OP-puro (50 µM; 10 mM reconstituted stock (pH 6.4); Medchem Source) was added to cultured cells 30 min before collection. For measuring mitochondrial protein synthesis, MitoTracker Deep Red was added together with OP-puro 30 min before collection at a concentration of 200 nM in the culture medium (Thermo Fisher Scientific). An untreated sample served as a negative control in each assay. Cycloheximide (50 µg ml$^{-1}$; Sigma-Aldrich) or puromycin (2 µg ml$^{-1}$; Sigma-Aldrich) treated cells served as positive controls. Cells were rinsed with PBS and then collected with trypsin-EDTA.

For mitochondrial protein synthesis, mitochondria were extracted following the instructions of the manufacturer of the mitochondria isolation kit (Thermo Fisher Scientific). The extracted organelles were then fixed in 0.5 ml PFA (1% w/v in PBS; Santa Cruz) and kept for 15 min on ice in the dark. After fixation, all samples were washed in PBS and permeabilized in PBS supplemented with 3% FBS and 0.1% saponin (Sigma-Aldrich) for 5 min at room temperature. To conjugate OP-puro to a fluorochrome, an azide-alkyne cycloaddition was performed for 30 min at room temperature in the dark. For this, the Click-iT Cell Reaction Buffer Kit (Thermo Fisher Scientific) and 5 µM of Alexa Fluor 488/647-Azide (Thermo Fisher Scientific) were used. To remove excess reagents and reduce the background signal, the cells were washed twice in PBS supplemented with 3% FBS and 0.1% saponin. Finally, all samples were resuspended in PBS and analysed by flow cytometer (Fortessa). Fluorescence of each sample was measured by the BD LSRFortessa Analyzer (BD Biosciences). Data were further processed by FlowJo software. Fluorescence measurements were visualized by histogram, and the raw fluorescence median values were extracted from the selected subpopulations.

To assess the effect of antibiotics on mitochondrial protein synthesis, cells were treated with tigecycline (Sigma-Aldrich, 10 µM), doxycycline (Sigma-Aldrich, 10 µg ml$^{-1}$), linezolid (Sigma-Aldrich, 45 µM), chloramphenicol (Sigma-Aldrich, 300 µg ml$^{-1}$), amoxicillin (Sigma-Aldrich, 10 µg ml$^{-1}$) and ampicillin (Sigma-Aldrich, 10 µg ml$^{-1}$) 24 h before OP-puro incorporation.

### Flow cytometry and analysis

Cells were trypsinized, washed twice with PBS and fixed for 10 min with 1% paraformaldehyde in PBS. After two additional PBS washes, cells were incubated with combinations of antibodies: PE–Cy7-conjugated CD44 (1:300, BD Pharmingen, 560533), FITC- or eFluor 660- conjugated CD36 (1:500, BD Bioscience, 555454 and 50-0369-42, Thermo Fisher Scientific). After incubation for 45 min at 4 °C, cells were washed twice in PBS. Data acquisition was performed on a BD LSRFortessa Analyzer or a cell sorter (BD Biosciences). Data were analysed by FlowJo software.

### Cell death assay

Cell death was measured by flow cytometry (fluorescence-activated cell sorting; FACS) analysis of DNA fragmentation using propidium iodide (PI) and Annexin V staining (BD Biosciences). In brief, the supernatant and trypsinized cells were collected and centrifuged for 5 min at 1,800 rpm at 4 °C. Pellets were resuspended in a binding buffer containing 50 µg ml$^{-1}$ propidium iodide and Annexin V for 15 min at 4 °C. Cells were analysed by FACS within 1 h after staining. Cells were labelled as follow: live cells are PI$^-$ and Annexin V$^-$; early apoptosis cells are PI$^-$ and Annexin V$^+$; late apoptosis cells are PI$^+$ and Annexin V$^+$; and necrotic cells are PI$^+$ and Annexin V$^-$.

For analysis of cell death in the CD44$^H$CD36$^H$ subpopulation, SCC25 cells were infected with the lentivirus containing the GFP-shRNA targeting *NSUN3*. At day 0, 2, 4 and 6 after viral infection, the supernatant and trypsinized cells were collected and stained for 45 min with PE–Cy7-conjugated CD44 (1:300, BD Pharmingen, 560533) and eFluor 660- conjugated CD36 (1:300, 50-0369-42, Thermo Fisher Scientific). Cells were then washed and stained for 15 min with PI and Annexin V (BD Biosciences) according to the manufacturer's instructions. The double-negative population (PI$^-$Annexin V$^-$) was labelled as the live cells. For all cell death assays data are represented as mean ± s.d. of at least three independent experiments carried out in triplicate.

### Oxygen consumption, lactate production and ATP rate assay

Oxygen consumption rate (OCR) and extracellular acidification rate (ECAR) were measured on a Seahorse XFe96 extracellular flux analyzer (Agilent) following the manufacturer's instructions. In brief, cells were seeded at $1 × 10^5$ (SCC25, VDH15) or $5 × 10^4$ (FaDu) cells per well in cell culture microplates (Sigma-Aldrich). After reaching 70–90% confluency, cells were equilibrated for 1 h in XF assay medium supplemented with 10 mM glucose, 1 mM sodium pyruvate and 2 mM glutamine in a non-CO$_2$ incubator. OCR and ECAR were monitored at baseline and throughout sequential injections of oligomycin (1 µM), carbonyl cyanide-4-(trifluoromethoxy)phenylhydrazone (1 µM) and rotenone or antimycin A (0.5 µM each). For measurement of ATP rate assay, only oligomycin (1 µM), followed by rotenone or antimycin A (0.5 µM each) was used. Data for each well were normalized to protein concentration as determined using the Pierce BCA Protein Assay kit (Thermo Fisher Scientific) after measurement on the XFe96 machine.

### Analysis of metabolites through ultra-high performance liquid chromatography–mass spectrometry

For determination of organic acids, three million cells per sample were extracted in 0.2 ml ice-cold methanol with sonication on ice. Fifty microlitres of extract was mixed with 25 µl 140 mM 3-nitrophenylhydrazine hydrochloride (Sigma-Aldrich), 25 µl methanol and 100 µl 50 mM ethyl-3-(3-dimethylaminopropyl) carbodiimide hydrochloride (Sigma-Aldrich) and incubated for 20 min at 60 °C. Separation was performed on the above-described ultra-high performance liquid chromatography (UPLC) system coupled to a QDa mass detector (Waters) using an Acquity HSS T3 column (100 mm × 2.1 mm, 1.8 µm, Waters) which was heated to 40 °C. Separation of derivates was achieved by increasing the concentration of 0.1% formic acid in acetonitrile (ACN; B) in 0.1% formic acid in water (A) at 0.55 ml min$^{-1}$ as follows: 2 min 15% B, 2.01 min 31% B, 5 min 54% B, 5.01 min 90% B, hold for 2 min, and return to 15% B in 2 min. Mass signals for the following compounds were detected in single-ion record (SIR) mode using negative detector polarity and 0.8 kV capillary voltage: lactate (224.3 $m/z$; 25 V cone voltage (CV)), malate (403.3 $m/z$;

25 V CV), succinate (387.3 $m/z$; 25 V CV), fumarate (385.3 $m/z$; 30 V CV), citrate (443.3 $m/z$; 10 V CV), 2-hydroxy-ketoglutarate (417.0 $m/z$; 15 V CV), pyruvate (357.3 $m/z$; 15 V CV) and ketoglutarate (550.2 $m/z$; 25 V CV). Data acquisition and processing were performed with the Empower3 software suite (Waters). Organic acids were quantified using ultrapure standards (Sigma).

## Orthotopic transplantation of OSCC cells

All mice were housed in the DKFZ Central Animal Laboratory. All mouse husbandry and experiments were performed according to the guidelines of the local ethics committee under the terms and conditions of the animal licence G-351/19.

OSCC cells were transduced with a retroviral vector expressing luciferase-GFP (luc-GFP). A total of 75,000 cells were injected into the tongue of NSG mice[18]. Adult male and female mice were used but sex-matched in each experiment. The progress of cancer development was monitored for 21 days and luciferase signal was measured in the oropharyngeal area, the surrounding lymph nodes and the lungs. Luciferase bioluminescent signal was measured immediately after injection (T0) and then, at least once weekly using the Xenogen IVIS Imaging System-100 (Caliper Life Sciences). For this, mice received intraperitoneal injection of 50 µl D-luciferin (Promega; 5 mg ml$^{-1}$ in PBS). Continuous administration of isofluorane gas was provided to maintain the anaesthesia of the mice during imaging. Data were quantified with Living Image software v.4.4 (Caliper Life Sciences). Quantifications were calculated with unsaturated pixels. The minimum and maximum values of the colour scales are shown in the images.

To assess the effect of antibiotics on tumour development, the SCC25 cell line and VDH01 PDC were injected in the tongue of NSG mice. After 7 days, we daily injected 100 µl of tigecycline (Sigma-Aldrich, 50 mg kg$^{-1}$), doxycycline (Sigma-Aldrich, 50 mg kg$^{-1}$), amoxicillin (Sigma-Aldrich, 50 mg kg$^{-1}$) or PBS as a control.

The tumour growth was monitored using in vivo imaging (Xenogen IVIS Imaging System-100; Caliper Life Sciences) once in the first week and then two to three times per week for a maximum of four weeks. Mice were killed before this time point when the tumour reached a maximum diameter of 1.0 cm. None of the mice in this study reached this end-point. Experiments were also stopped immediately when mice showed a hunched posture or weight loss of 20% of their initial weight. Mice with tumours were also killed if they showed signs of necrosis or inflammation associated with tumour growth. Mice with moderate dyspnoea owing to metastases in the lungs were killed. Mice with signs of infection, non-healing, bloody or oozing wounds were also killed. These limits were not exceeded in any of the experiments. For each experiment, mice were killed at the same time, once one experimental group reached the humane end-point.

## Isolation of orthotopic tumour cells

Tumours were separated from mouse tongue tissue and the tissue was chopped in 0.5% trypsin 1-300 (MP Biomedical) in K-SFM medium (Gibco). After complete homogenization, samples were incubated at 37 °C for 2 h with shaking. Homogenates were filtered sequentially in 40-µm BD strainers and centrifuged at 1,000 rpm for 10 min at 4 °C. The supernatant was discarded, and each pellet was resuspended in 1× PBS/4% calcium-chelated FBS for antibody staining and subsequent FACS analysis.

## Tumoroid assays

Cells cultured in 2D were resuspended in K-SFM medium (Thermo Fisher Scientific) and placed in an ultra-low adherent culture dish (STEMCELL Technologies). After 7 days, the number of tumoroids per well was assessed and representative pictures were taken for each condition. For the invasion assay, tumoroids were pelleted (200$g$, 5 min) and grown in the 3D collagen I culture kit (EMD Millipore) for 24 or 48 h. Images of tumoroids were taken using a Leica DMIL microscope.

Quantification of the number of leader cell was done on tumoroids from three independent experiments.

To overexpress CD36, 5 days after seeding 2D cells in an ultra-low adherent culture dish, tumoroids were transiently transfected with CMV-mCherry-CD36-C-10 (55011, Addgene). Tumoroids were washed after 12 h and reseeded in an ultra-low adherent culture dish with 30 µM of palmitate-BSA (Agilent, 102720-100). After 24 h, tumoroids were pelleted and casted in a collagen I culture kit for 48 h.

To assess the effect of antibiotics on tumoroid invasion, tigecycline (Sigma-Aldrich, 10 µM), doxycycline (Sigma-Aldrich, 10 µg ml$^{-1}$), linezolid (Sigma-Aldrich, 45 µM), chloramphenicol (Sigma-Aldrich, 300 µg ml$^{-1}$), amoxicillin (Sigma-Aldrich, 10 µg ml$^{-1}$) and ampicillin (Sigma-Aldrich, 10 µg ml$^{-1}$) were added to the medium directly after seeding the tumoroids in the 3D collagen gel.

## Immunostaining

Extracted mouse tongues containing the tumours were fixed overnight with 4% paraformaldehyde, transferred to 70% ethanol and embedded in paraffin. Samples were sectioned at 4 µm. Sections were permeabilized for 10 min with PBS containing 0.3% Triton X-100 at room temperature and washed three times for 5 min in PBS. To block nonspecific antibody binding, sections were incubated with blocking buffer comprising 3% FBS in PBS with 0.1% Tween-20 (PBST) for 1 h. To detect specific proteins of interest, cells were then incubated with primary antibodies diluted in 1% FBS in PBST at 4 °C overnight. The cells were then washed three times in PBS for 5 min each. To label the detected proteins, cells were incubated with the Alexa Fluor 488-, Alexa Fluor 555-, Alexa Fluor 647-conjugated secondary antibodies diluted in 1% BSA in PBST for 1 h at room temperature, protected from light (1:1,000; Thermo Fisher Scientific). Sections were washed as before and their nuclei were counterstained with DAPI (1:10,000 in PBS; Sigma-Aldrich) for 10 min. Finally, sections were rinsed with PBS and the glass coverslips were mounted using fluorescence mounting medium (S302380-2; Agilent). The primary antibodies used were CD44 (1:200, 14-0441-82, Thermo Fisher Scientific), MTCO1 (1:200, ab91317, Abcam), MTCO2 (1:200, PA5-26688, Thermo Fisher Scientific), cytokeratin 10 (1:200, PRB-159P, Biolegend), and filaggrin (1:200, Covance, PRB-417P-100).

Tumoroids were incubated for 2 h with MitoTracker Red CMXROS (Thermo Fisher Scientific). Then, they were fixed 15 min with 4% paraformaldehyde, washed three times in PBS and counterstained with DAPI (1:10,000 in PBS) for 10 min.

Cells infected with GFP-NSUN3-WT or GFP-NSUN3-MUT constructs were incubated for 30 min with MitoTracker Deep Red (Thermo Fisher Scientific) at 200 nM. Then, they were fixed for 15 min with 1% paraformaldehyde, washed three times in PBS and mounted with a fluorescent mounting medium (Dako). Fluorescence images were acquired using a confocal microscope (Leica SP5) at 1,024 × 1,024-dpi resolution. The length of mitochondria was analysed with Fiji software by taking the average length of 20 mitochondria per cell in different conditions.

To measure mitochondrial length in invading leader cells, tumoroids were casted in collagen I matrix for 48 h. Then, they were fixed for 15 min with 4% paraformaldehyde, washed three times in PBS and incubated at room temperature in 30% sucrose PBS for 12 h, followed by 2 h in OCT–30% sucrose PBS (1:1). Collagen matrix was embedded and frozen in OCT and sectioned at 10 µm. Sections were permeabilized for 10 min with PBS containing 0.3% Triton X-100 at room temperature and washed three times for 5 min in PBS. To block nonspecific antibody binding, sections were incubated with blocking buffer comprising 3% FBS in PBS with 0.1% Triton X-100 for 1 h. To detect specific proteins of interest, cells were then incubated with primary antibodies diluted in 1% FBS in PBST at 4 °C overnight. The cells were then washed three times in PBS for 5 min each. To label the detected proteins, cells were incubated with Alexa Fluor 647-conjugated secondary antibodies diluted in 1% BSA in PBST for 1 h at room temperature, protected from light (1:1,000; Thermo Fisher Scientific). Sections were washed as before and their

nuclei counterstained with DAPI (1:10,000 in PBS; Sigma-Aldrich) and Alexa-555 Phalloidin (Thermo Fisher Scientific) for 10 min. Finally, sections were rinsed with PBS and the glass coverslips were mounted using fluorescence mounting medium (S302380-2; Agilent). The primary antibody used was TOMM20 (1:300, Abcam, ab56783). Fluorescence images were acquired using a confocal microscope (Leica SP5) at 1,024 × 1,024-dpi resolution. All of the images were further processed with Fiji software. Mitochondrial length was measured in pixels for at least 20 mitochondria per cell, with a minimum of 15 cells per condition from 3 independent experiments.

For glucose uptake, tumoroids were cultured for 7 days in non-adherent plates (STEMCELL Technologies). The medium was changed for DMEM no glucose (Thermo Fisher Scientific) for 30 min. Then, tumoroids were treated for 3 h with 50 μM 2-deoxy-D-glucose-IR (Licor). Tumoroids were washed three times in PBS and fixed with 4% paraformaldehyde for 15 min. After washing, tumoroids were labelled with DAPI (1:10,000 in PBS) and F-actin counterstained with Alexa-647 Phalloidin (Thermo Fisher Scientific) for 30 min. Fluorescence images were acquired using a confocal microscope (Leica SP5) at 1,024 × 1,024-dpi resolution. All of the images were further processed with Fiji software.

Immunohistochemistry for NSUN3 was performed as described previously[24]. Patients of the HIPO-HNC cohort (GSE117973) were treated between 2012 and 2016 at the University Hospital Heidelberg, Germany. Patient samples were obtained and analysed under protocols S-206/2011 and S-220/2016, approved by the Ethics Committee of Heidelberg University, with written informed consent from all participants. This study was conducted in accordance with the Declaration of Helsinki.

Formalin-fixed paraffin-embedded (FFPE) tumour sections were labelled for NSUN3 (1:100; Genetex, GTX46175). The specificity of antibody staining was confirmed by immunohistochemistry staining using a rabbit IgG isotype control antibody (1:1,000; DA1E, Cell Signaling Technology) on serial FFPE tissue sections from HNSCC samples. The NSUN3 immunoscore (IRS, ranging from 0 to 300) was calculated as a product of staining intensity (ranging from low = 1, moderate = 2 and high = 3) and the percentage total of positively stained tumour cells (ranging from 0% to 100%). FFPE tumour sections were provided by the tissue bank of the National Center for Tumour Disease (Institute of Pathology, University Hospital Heidelberg, Germany), at which preservation and storage of tumour samples occurs under controlled and standardized protocols (https://www.klinikum.uni-heidelberg.de/pathologie-kooperationen/nct-gewebebank).

To estimate the relative distance of cell populations from the tumour–stroma border, pixels were classified as NSUN3+ or NSUN3− (ref. [48]). For each NSUN3+ or NSUN3− pixel we then identified the nearest neighbouring tumour–stroma border. Using Fiji software, we calculated the Euclidian distance map. Cell counts were then pooled into 50 μm bins.

### Electron microscopy

Cells grown on punched sheets of Aklar-Fluoropolymer films (EMS) were embedded in epoxy resin for ultrathin sectioning according to standard procedures: primary fixation in buffered aldehyde (4% formaldehyde, 2% glutaraldehyde, 1 mM $CaCl_2$, 1 mM $MgCl_2$ in 100 mM sodium phosphate, pH 7.2), post-fixation in buffered 1% osmium tetroxide followed by en-bloc staining in 1% uranylacetate. After dehydration in graded steps of ethanol, the adherent cells were flat-embedded in epoxide (Glycidether, NMA, DDSA: SERVA). Ultrathin sections at a nominal thickness of 60 nm and contrast-stained with lead citrate and uranylacetate were observed in a Zeiss EM 910 at 120 kV (Carl Zeiss) and micrographs were taken using a slow scan CCD camera (TRS).

### Quantitative translation measurements using mass spectrometry and analyses

Multiplexed enhanced protein dynamics (mePROD) proteomics followed by mass spectrometry were performed as previously described[23].

In brief, cell pellets from sorted cell populations or bulk were lysed in 2% SDS, 150 mM NaCl, 10 mM TCEP, 40 mM chloracetamide and 100 mM Tris pH 8. Lysates were incubated at 95 °C, followed by sonication and additional incubation at 95 °C for 10 min. Proteins were isolated using methanol-chloroform precipitation and resuspended in 8 M urea and 10 mM EPPS pH 8.2. Digests were performed overnight after dilution to 1 M urea, 10 mM EPPS pH 8.2 with 1:50 w/w LysC (Wako) and 1:100 w/w trypsin (Promega). Peptides were isolated using tC18 SepPak columns (50 mg, Waters) and subsequently dried. For TMT labelling, peptides were resuspended in 200 mM EPPS pH 8.2 and 10% ACN and mixed 1:2 (w/w) with TMT reagents. For mePROD baseline and boost, completely light and heavy digests were used as described previously. Peptides were fractionated using a Dionex Ultimate 3000 analytical HPLC. Pooled and purified TMT-labelled samples were resuspended in 10 mM ammonium bicarbonate (ABC) and 5% ACN, and separated on a 250-mm-long C18 column (X-Bridge, 4.6 mm ID, 3.5 μm particle size; Waters) using a multistep gradient from 100% solvent A (5% ACN and 10 mM ABC in water) to 60% solvent B (90% ACN and 10 mM ABC in water) over 70 min. Eluting peptides were collected every 45 s into a total of 96 fractions, which were cross-concatenated into 24 fractions and dried for further processing.

Peptides were resuspended in 2% ACN and 0.1% TFA and separated on an Easy nLC 1200 (Thermo Fisher Scientific) and a 35-cm-long, 75-μm ID fused-silica column, which had been packed in house with 1.9-μm C18 particles (ReproSil-Pur, Dr. Maisch), and kept at 45 °C using an integrated column oven (Sonation). Peptides were eluted by a non-linear gradient from 5% to 38% ACN over 120 min and directly sprayed into a QExactive HF mass spectrometer equipped with a nanoFlex ion source (Thermo Fisher Scientific) at a spray voltage of 2.3 kV. Full scan MS spectra (350-1400 $m/z$) were acquired at a resolution of 120,000 at $m/z$ 200, a maximum injection time of 100 ms and an AGC target value of $3 \times 10^6$. Up to 20 most intense peptides per full scan were isolated using a 1 Th window and fragmented using higher energy collisional dissociation (normalized collision energy of 35). MS/MS spectra were acquired with a resolution of 45,000 at $m/z$ 200, a maximum injection time of 86 ms and an AGC target value of $1 \times 10^5$. Ions with charge states of 1 and >6 as well as ions with unassigned charge states were not considered for fragmentation. Dynamic exclusion was set to 20 s to minimize repeated sequencing of already acquired precursors.

Raw files were analysed using Proteome Discoverer (PD) 2.4 software (Thermo Fisher Scientific). Spectra were selected using default settings and database searches were performed using Sequest HT node in PD. Database searches were performed against the trypsin-digested Homo Sapiens SwissProt database (2018-11-21) and FASTA files of common contaminants ('contaminants.fasta' provided with MaxQuant) for quality control. Fixed modifications were set as TMT6 at the N terminus and carbamidomethyl at cysteine residues. One search node was set up to search with TMT6 (K) and methionine oxidation as static modifications to search for light peptides and one search node was set up with TMT6+K8 (K, +237.177), Arg10 (R, +10.008) and methionine oxidation as static modifications to identify heavy peptides. For both nodes, Acetyl (+42.011), Met-loss (−131.040) and Met-loss + Acetyl (−89.030) were set as dynamic modifications at the protein terminus. Searches were performed using Sequest HT. After searching, posterior error probabilities were calculated and peptide-spectrum matches (PSMs) filtered using Percolator with default settings. The consensus workflow for reporter ion quantification was performed with default settings, except that the minimum signal-to-noise ratio was set to 5. Results were then exported to Excel files for further processing by an in-house Python pipeline[23,49].

For co-expression clustering, a type II ANOVA was used on a fitted ordinary least squares linear model for each protein to filter out high-variance proteins. All proteins with an ANOVA $P$ value of less than 0.05 were used for further analysis. The remaining data were $\log_2$-transformed and a Pearson correlation matrix was calculated. ToppGene was used for the GO analysis on the clusters.

## RNA-seq and analyses

Tumours were isolated from mice, and connective tissue was removed to the largest extent possible. Tissue was chopped in 0.5% trypsin 1-300 (MP Biomedical) in K-SFM medium (Gibco) using a McIlwain Tissue Chopper. After complete homogenization, samples were incubated at 37 °C for 90 min with shaking. Homogenates were filtered sequentially in 100-μm, 70-μm and 40-μm BD strainers and centrifuged at 1,000 rpm for 10 min at 4 °C. Supernatant was discarded, and each pellet was resuspended in 1× PBS/4% calcium-chelated FBS. GFP-positive human cancer cells were flow-sorted. Total RNA was extracted using the Trizol protocol and treated with DNase. rRNA-depleted RNA was used to generate the RNA-seq libraries using NEXTflex Directional RNA-seq Kit V2 (Illumina). All 11 samples were multiplexed and sequenced in the HiSeq 4k PE 100 sequencing platform (Illumina).

For all samples, low-quality bases were removed with the Fastq_quality_filter from the FASTX Toolkit (0.0.13) with 90% of the read needing a quality phred score > 20. Homertools 4.7 was used for PolyA-tail trimming, and reads with a length < 17 were removed. Genomic mapping was performed with STAR v.2.3 for the filtered reads with human genome 38 (ref. [50]). For quality checking, PicardTools 1.78 CollectRNASeqMetrics was performed on the mapped reads. Count data were generated by FeatureCounts v.1.4.5-p1 with parameters --minReadOverlap 3 -T 3 -M -O -s 0 using the gencode.v32.annotation.gtf (https://www.gencodegenes.org/) file for annotation[51]. The FPKM values were calculated using the same annotation file with a custom perl script. For the comparison with DESeq2[52], the input tables containing the replicates for the groups to compare were created by a custom perl script. In the count matrix, rows with an average count number of less than 10 were removed, then DESeq2 (v.1.4.1) was run with default parameters. The result tables were annotated with gene information (gene symbol, gene type) derived from the gencode.v32.annotation.gtf file.

## In silico analyses using TCGA-HNSC datasets

RNA expression levels, clinical as well as follow-up data were downloaded from the TCGA-HNSC ($n = 500$) cohort (https://portal.gdc.cancer.gov/) in November 2019. Differentially expressed gene analysis was performed by the 'EdgeR' package in R (ref. [53]), depending on NSUN3 expression with a cut-off at false discovery rate (FDR) < 0.01 and a $\log_2$-transformed fold change either higher than 1 or lower than −1. The GSEA algorithm was used to compute the normalized enrichment score and statistical significance for Molecular Signatures Database (MSigDB) hallmark, C2, C5 and C6 collection terms and gene set permutations were performed 1,000 times for each analysis by GSEA v.4.0.3 software.

For clustering analysis, expression values of the top and bottom 50 differentially expressed genes of *NSUN3* quartile expression depending on the $\log_2$-transformed fold change were $\ln(x + 1)$ transformed, and clustering of patients from the TCGA was performed using Euclidean distance and Ward (unsquared distances) linkage. Heat maps with hierarchical trees were generated by the web tool ClustVis[54].

Differentially expressed gene analysis was performed on VDH15 control and sh*NSUN3* tumours, with a cut-off at FDR < 0.01. Enrichment scores were computed by ssGSEA applying the 'GSVA' package in R[55], using the top and bottom 150 differentially expressed genes. ssGSEA scores of patients from the TCGA were then plotted with regard to the occurrence of lymph node metastasis.

To identify the NSUN3-driven gene signature in progression events, the best cut-off of the top or bottom 150 NSUN3-related signature ssGSEA scores for progression-free interval (PFI) was computed by 'maxstat' (smethod= "LogRank", pmethod= "exactGauss", and abseps=0.01) in the TCGA-HNSC cohort. We defined the patients in the HNSC cohort whose top 150 ssGSEA score was higher than the best cut-off and whose bottom 150 ssGSEA score was lower than the best cut-off as the high NSUN3-driven signature group. On the other side, the patients in the HNSC cohort whose top 150 ssGSEA score was lower than

the best cut-off and whose bottom 150 ssGSEA score was higher than the best cut-off were defined as the low NSUN3-driven signature group.

## Sample sizing and collection

No statistical methods were used to predetermine sample size, but a minimum of three samples were used per experimental group and condition. The number of samples is represented in the graphs as one dot per sample. Samples and experimental mice were randomly assigned to experimental groups. Sample collection was also assigned randomly. Sample collection and data analysis were performed blindly whenever possible. Whenever possible automated quantifications were performed using the appropriate software.

## Reporting summary

Further information on research design is available in the Nature Research Reporting Summary linked to this paper.

## Data availability

Quantitative proteomics data are available at the Proteomics Identifications Database (PRIDE) with accession number PXD021835. RNA-seq data using VDH01 and VDH15 cells are available at the European Genome-phenome Archive (EGA) under the accession number EGAS00001004765, including the attached studies EGAD00001008743 and EGAD00001008742. All other sequencing data are deposited at the Gene Expression Omnibus (GEO) under the accession number GSE201993. Results are in part based on TCGA-HNSC data (accession number phs000178) that were downloaded from TCGA (https://portal.gdc.cancer.gov). Source data are provided with this paper.

## Code availability

Custom perl script to compare groups in RNA-seq input table can be found here: https://zenodo.org/search?page=1&size=20&q=6518420
TrimGalore! https://www.bioinformatics.babraham.ac.uk/projects/trim_galore/
Bismark (v.0.22.3) https://www.bioinformatics.babraham.ac.uk/projects/bismark/
RSamtools https://bioconductor.org/packages/release/bioc/html/Rsamtools.html
GenomicAlignments https://bioconductor.org/packages/release/bioc/html/GenomicAlignments.html
Matrixplot https://rdrr.io/cran/VIM/man/matrixplot.html
ToppGene https://toppgene.cchmc.org/
FASTX-Toolkit http://hannonlab.cshl.edu/fastx_toolkit/
HomerTools http://homer.ucsd.edu/homer/ngs/homerTools.html
STAR v.2.3 https://github.com/alexdobin/STAR
PicardTools v.1.78 http://broadinstitute.github.io/picard/
CollectRNASeqMetrics https://gatk.broadinstitute.org/hc/en-us/articles/360037057492-CollectRnaSeqMetrics-Picard-
FeatureCounts v.1.4.5 https://rnnh.github.io/bioinfo-notebook/docs/featureCounts.html
DESeq2 https://bioconductor.org/packages/release/bioc/html/DESeq2.html
EdgeR https://bioconductor.org/packages/release/bioc/html/edgeR.html
GSVA https://www.bioconductor.org/packages/release/bioc/html/GSVA.html
Maxstat https://cran.r-project.org/web/packages/maxstat/index.html

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

**Acknowledgements** We thank the tissue bank of the National Center for Tumor Disease (Institute of Pathology, University Hospital Heidelberg) for providing FFPE tumour specimens; all of the DKFZ core facilities for their support; S. Schmitt for support with flow cytometry; F. Bestvater for imaging and microscopy advice; the Metabolomics Core Technology Platform of Heidelberg University for support with metabolite analyses; all members of staff of the DKFZ Central Animal Laboratory for their support; and the Quantitative Proteomics Unit at Goethe University for mass spectrometry analyses. Some results in this study are in part based on data generated by the TCGA Research Network (https://www.cancer.gov/tcga). This work was funded by the Helmholtz Association (W2/W3-106), Cancer Research UK (CRUK; C10701/A15181) and Worldwide Cancer Research (21-0223). S. Delaunay was supported by an EMBO long-term fellowship (LTFS48) and by the Leon Fredericq Foundation. B.F. was supported by the China Scholarship Council (CSC; 201708330262). Some figure panels were created with BioRender.com.

**Author contributions** S. Delaunay performed and designed in vitro and in vivo experiments and wrote the manuscript. G.P. taught orthotopic transplantation assays and analyses. B.F. performed computational analyses of patient data. K.K. performed nascent proteomics experiments and analyses. M.B. helped generating RNA bisulfite and fCAP sequencing data. A.H-W. performed computational analyses of the RNA-seq datasets. K.R. performed electron microscopy. K.Z. and E.H. were involved in patient tissue collection and analyses. C.M. supervised nascent proteomics experiments and analyses. S. Dietmann performed computational analyses of the RNA bisulfite and fCAP sequencing data. J.H. performed and analysed histological stainings of patient samples and supervised computational analyses of patient data. S.A.B. supervised G.P. and provided orthotopic transplantation assay protocols and patient-derived tumour cells. M.F. supervised S. Delaunay and M.B., designed and analysed experiments and wrote the manuscript.

**Funding** Open access funding provided by Deutsches Krebsforschungszentrum (DKFZ).

**Competing interests** S.A.B. is a co-founder and scientific advisor at ONA Therapeutics. J.H. receives commercial funding from CureVac AG and acts as a consultant and advisory board member for Bristol-Myers Squibb and MSD Sharp & Dohme. M.F. receives commercial funding from Merck. The other authors declare no competing interests.

**Additional information**
**Correspondence and requests for materials** should be addressed to Michaela Frye.

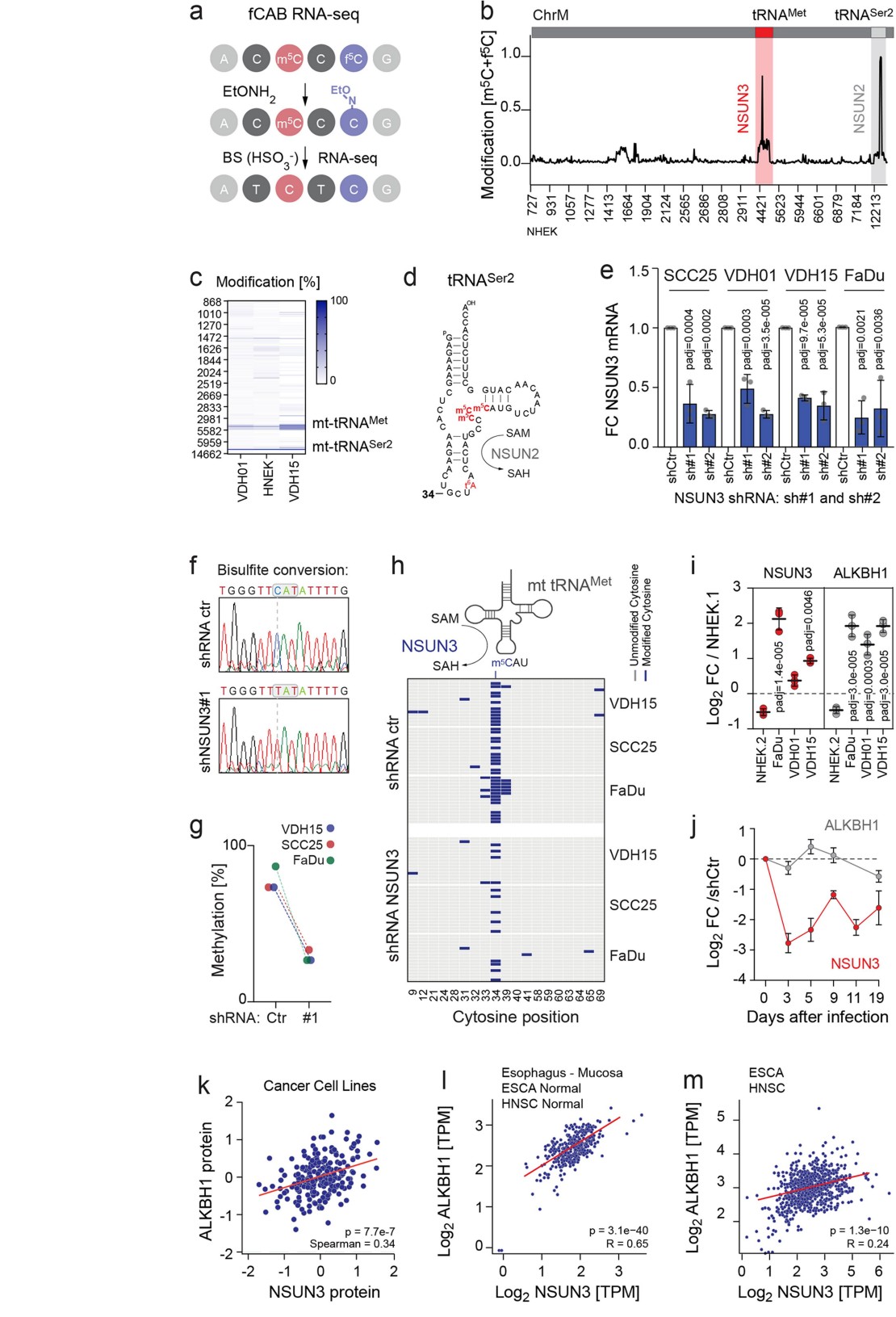

**Extended Data Fig. 1** | See next page for caption.

**Extended Data Fig. 1 | Depletion of NSUN3 removes m⁵C and f⁵C from mt-tRNA^Met.** **a**. Schematic representation of fCAB- and bisulfite (BS) parallel sequencing (seq). **b**. Detection of m⁵C and f⁵C sites in the mitochondrial (mt) tRNA landscape using fCAB-Seq in normal human epithelial keratinocytes (NHEK). Plotted are all cytosines on the mitochondrial chromosome (ChrM) with a coverage >100 in both independent replicates. The two peaks correspond to C34 of mt-tRNA^Met mediated by NSUN3 and C47, 48, 49 in mt-tRNA^Ser2 mediated by NSUN2 respectively. **c**. Heat map comparing modification levels along ChrM in three independent cell lines. **d**. mt-tRNA^Ser2 secondary structure with known modifications highlighted in red. **e**. Relative fold change (FC) of NSUN3 RNA levels in the indicated cell lines infected with control (Ctr) or *NSUN3* shRNAs (sh#1, sh#2) (*n* = 3 independent infections). **f**. Sequence of bisulfite-converted mt-tRNA^Met. **g**. m⁵C levels in the indicated cell lines shown in (h). **h**. Illustration of mt-tRNA^Met in the anticodon loop (upper panel), and heat maps (lower panels) showing methylated (blue) and non-methylated (grey) cytosines identified by RNA BS-seq in the indicated cell lines expressing (shRNA ctr) or lacking NSUN3 (shRNA NSUN3). **i,j**. Log₂ FC of NSUN3 (red) and ALKBH1 (grey) RNA levels in the indicated cell lines versus NHEK cells (i) (*n* = 3 independent infections) or after depletion of NSUN3 in time course (j) (*n* = 3 technical replicates per time point). **k**. Correlation of NSUN3 and ALKBH1 protein levels in cancer cell lines (Cancer Cell Line Encyclopedia). **l,m**. Correlation of ALKBH1 and NSUN3 RNA levels in normal oesophagus mucosa and TCGA normal oesophageal carcinoma (ESCA) and head and neck squamous cell carcinoma (HNSC) tissue (l) and tumour tissue (m). p = p-value of Spearman correlation coefficient. Shown is mean +/- SD (e,i,j). Two-sided Šídák's test (e,i). Exact p-values are indicated.

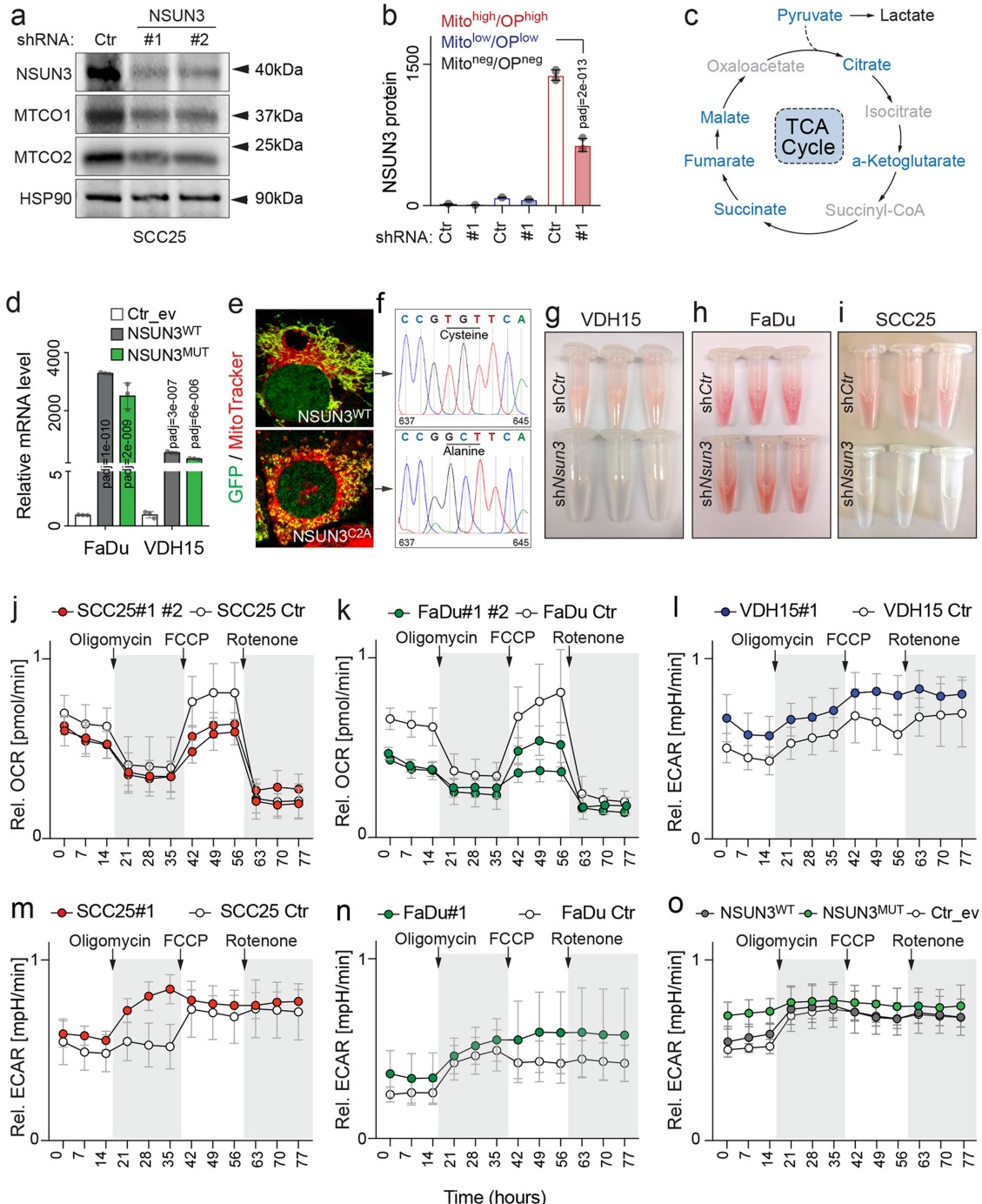

**Extended Data Fig. 2 | NSUN3 regulates mitochondrial activity. a.** Western blot for NSUN3, MT-CO1, and MT-CO2 in SCC25 cells infected with control (Ctr) and *NSUN3* shRNAs (#1, #2). HSP90: loading control. **b.** Quantification of NSUN3 protein levels in mitochondria isolated from the indicated subpopulations (*n* = 4 flow sorts). **c.** Illustration of the TCA cycle and its intermediate metabolites. **d.** Relative RNA levels of *NSUN3* in the indicated cell lines expressing wild-type (WT) or methylation-deficient mutated (MUT) NSUN3 proteins (*n* = 3 RT–qPCRs). **e.** Fluorescence staining of GFP and MitoTracker Deep Red in VDH15 cells infected with WT or MUT NSUN3. Representative immunofluorescence from 30 stained cells. **f.** Sequence of NSUN3 constructs showing the substitution of cysteine (NSUN3^WT) with alanine (NSUN3^MUT). **g-i.** Representative images showing media colour of the indicated cells infected with Ctr or *NSUN3* shRNAs after 48 h in culture. **j-n.** Oxygen consumption rate (OCR) (j,k) and extracellular acidification rate (ECAR) (l-n) in the indicated cell lines infected with Ctr or *NSUN3* shRNAs (sh#1, sh#2). (j: *n* = 5; k: shCtr *n* = 4, sh#1 *n* = 5, sh#2 *n* = 6; l: shCtr *n* = 5, sh#1 *n* = 7; m: shCtr *n* = 5, sh#1 *n* = 6; n: shCtr *n* = 4, sh#1 *n* = 5 injections). **o.** ECAR in VDH15 cells infected with empty vector (ctr_ev), NSUN3^WT or NSUN3^MUT constructs (Ctr_ev, NSUN3^WT: *n* = 4 injections; NSUN3^MUT: *n* = 6 injections). Shown is mean +/- SD (b,d,j-o). Two-sided Tukey's test (b) or Šídák's test (d). Exact p-values are indicated.

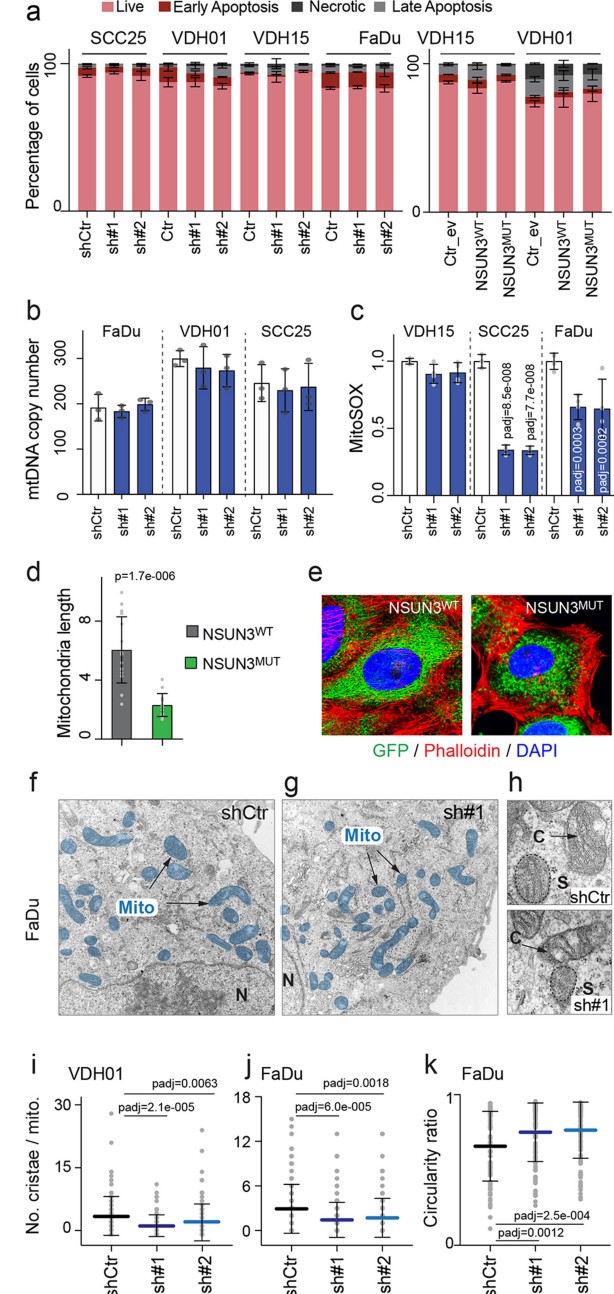

**Extended Data Fig. 3** | See next page for caption.

**Extended Data Fig. 3 | NSUN3 regulates mitochondrial function and shape.**
**a**. Percentage of live or dead cells using Annexin V and Propidium iodide (PI) staining of indicated cell lines infected with Control (Ctr) or *NSUN3* shRNAs (sh#1, sh#2) or NSUN3 overexpression constructs containing wild-type (WT) or mutated (MUT) NSUN3, or the empty vector as a control (Ctr_ev) ($n = 3$ infections). **b**. Quantification of the mitochondrial DNA copy number in shCtr or *NSUN3*-depleted (sh#1, sh#2) FaDu cells ($n = 3$ infections). **c**. MitoSOX flow cytometry to quantify cellular ROS production in the indicated cells ($n = 3$ infections). **d**. Quantification of average mitochondrial length in VDH15 cells infected with NSUN3$^{WT}$ or NSUN3$^{MUT}$ constructs (NSUN3$^{WT}$ $n = 18$; NSUN3$^{MUT}$: $n = 14$ cells). **e**. Fluorescent staining for overexpressed NSUN3 (GFP) and Phalloidin in VDH15 cells infected with NSUN3$^{WT}$ or NSUN3$^{MUT}$ constructs. Representative immunofluorescence out of 30 stained cells. **f-h**. Electron microscopy of FaDu cells infected with control (shCtr) or *NSUN3* shRNAs (sh#1). Higher magnifications of mitochondria are shown in (h). Representative images from 10 cells per condition from 2 independent experiments. (**i-k**) Quantification of cristae per mitochondria (i,j) and circularity ratio (k) in the indicated cells (i: Ctr $n = 118$, sh#1 $n = 116$, sh#2: $n = 117$; j: Ctr $n = 125$, sh#1 $n = 124$, sh#2: $n = 112$; k: Ctr $n = 103$, sh#1 $n = 131$, sh#2: $n = 115$ mitochondria in 10 different cells from 2 independent experiments). Shown is mean +/- SD (a-d, i-k). Two-sided Šídák's test (c,i-k). Two-tailed unpaired t-test (d). Exact p-values are indicated.

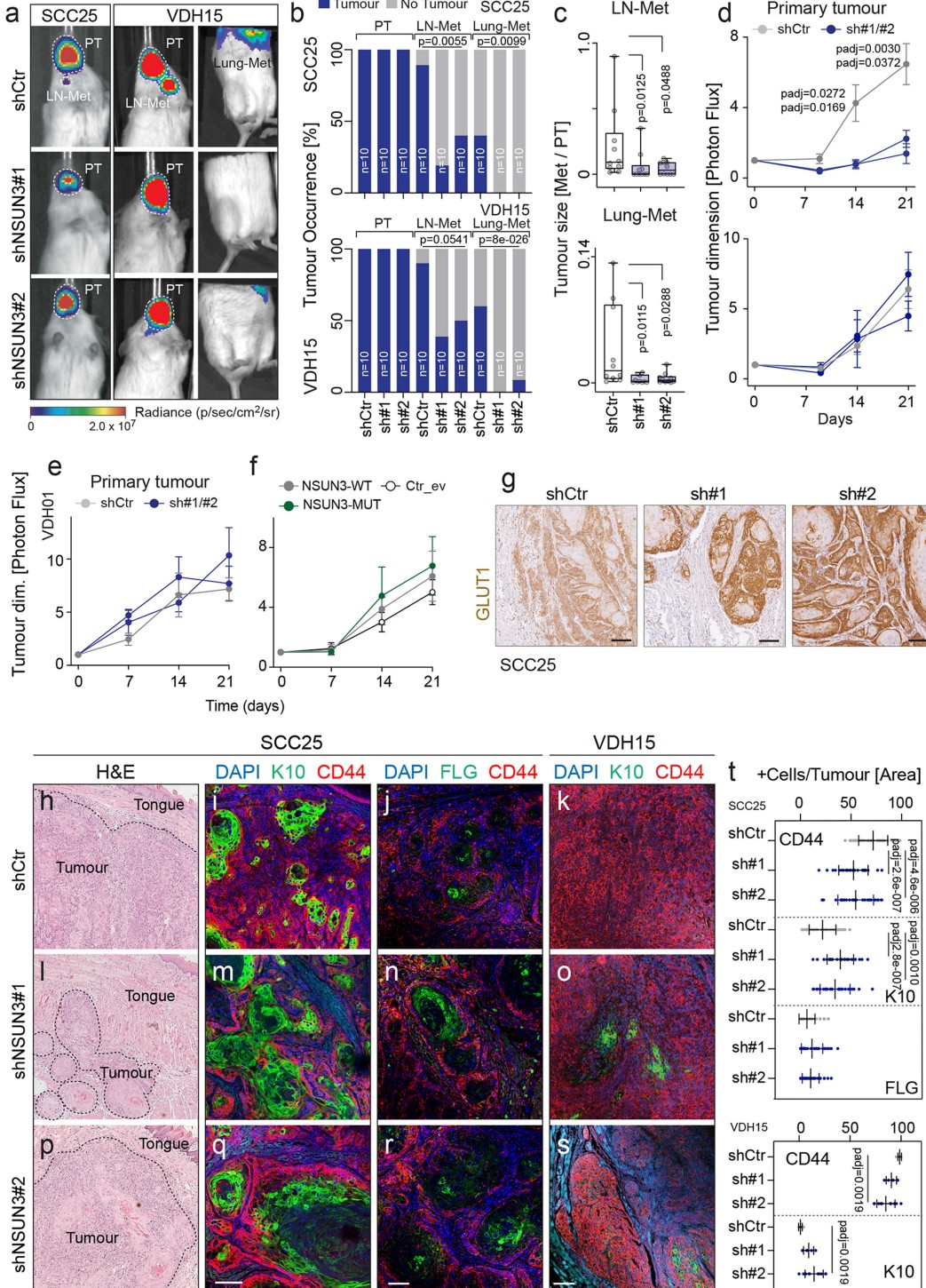

**Extended Data Fig. 4 | NSUN3-deficient tumours switch to glycolysis and fail to metastasize. a, b.** Bioluminescence imaging (a) and tumour occurrence (b) of primary tumours (PT), lymph node (LN) or lung metastases (Met), 21 days after orthotopic transplantations. Tumours derived from SCC25 or VDH15 cells infected with control (shCtr) or two different *NSUN3* shRNAs (#1, #2). **c-f.** Dimension of the metastasis relative to its matching primary tumour (c) quantified by measuring the relative luminescence signal shown in (a), and tumour growth (d-f) of tumours derived from SCC25 (c,d; upper panels; *n* =10 mice per condition), VDH15 (c,d; lower panels; *n* =10 mice per condition) and VDH01 (e,f) cells infected with shCtr or shRNAs #1 or #2) (c,d,e), or with an empty vector control (Ctr_ev), a wildtype (WT), or mutant (MUT) NSUN3 overexpression construct (f; shCTR: *n* =7 mice; sh#1, sh#2: *n* =6 mice; Ctr_ev: *n* =9 mice; WT, MUT: *n* =10 mice). **g.** Immunostaining for GLUT1 in shCtr or

shNSUN3 (sh#1, sh#2) infected SCC25 PTs. **h-t.** Hematoxylin and eosin (h,l,p), immunofluorescence (i-k; m-o; q-s) and quantification (t) of SCC25- and VDH15-derived primary tumours (day 21) infected with shCtr (upper panels), shNSUN3#1 (middle panels) or shNSUN3#2 (lower panels) stained for keratin 10 (K10), CD44, filaggrin (FLG) and DAPI as a nuclear counterstain (SCC25 CD44 and K10 staining: shCTR: *n* =33, sh#1 *n* =26, sh#2 *n* =25 tumours areas from 3 mice; SCC25 FLG staining: shCTR: *n* =46, sh#1 *n* =35, sh#2 *n* =33 tumour areas from 3 mice; VDH15: shCTR: *n* =6, sh#1 *n* =5, sh#2 *n* =11 tumour areas from 3 mice). Representative images from 3 mice per condition (g,h-s). Scale bar: 100 μm. Box plot shows minimum, first quartile, median, third quartile, and maximum (c). Shown is mean +/- SEM (d-f) or SD (t). C*hi*-squared test (b). Two-tailed Mann Whitney test (c), or two-sided Dunnett's test (d) and Šídák's test (t). Exact p-values are indicated.

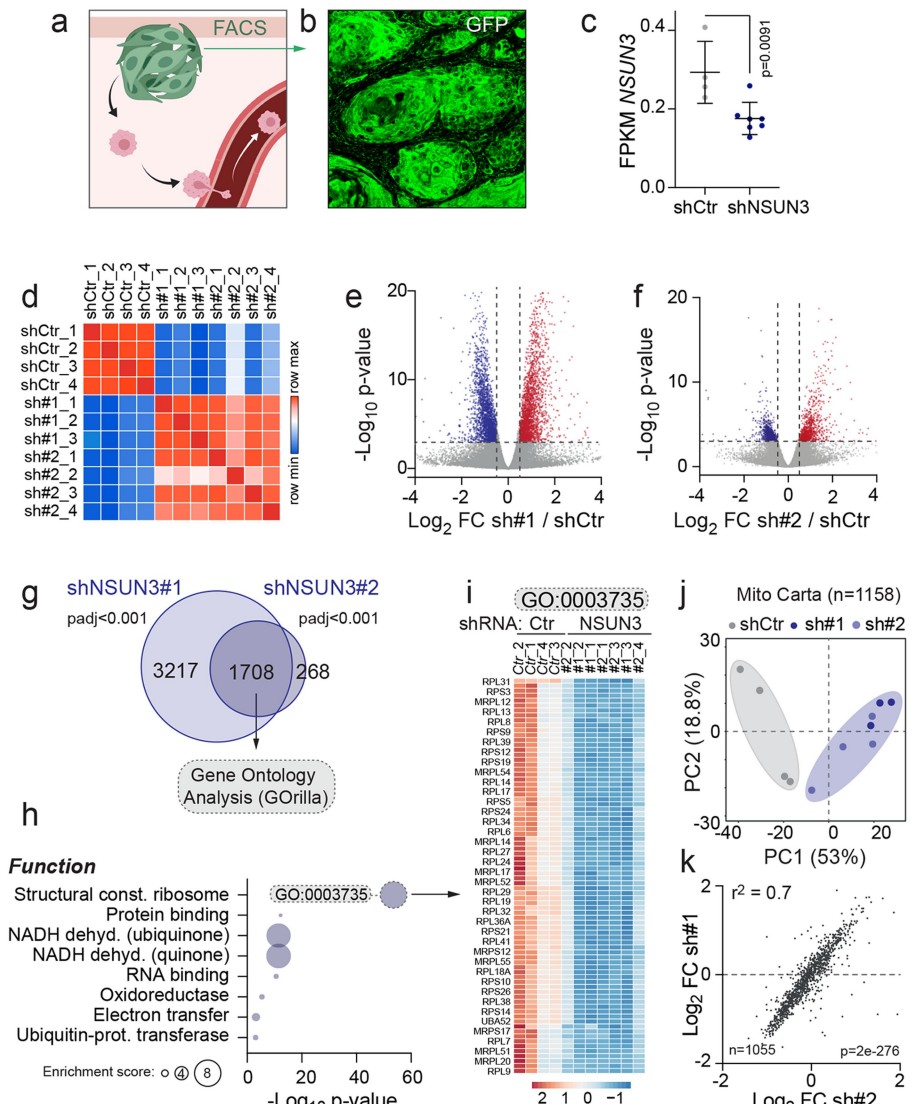

**Extended Data Fig. 5 | Mitochondria-driven gene signatures in primary tumours. a**, **b**. Illustration (a) of GFP-positive primary tumours in mouse tongues (b). **c**. RNA read counts for NSUN3 in control (shCtr) or *NSUN3* shRNAs (sh#1 plus sh#2) primary tumours (shCtr $n = 4$ mice/tumours; shNSUN3: $n = 7$ mice/tumours). **d**. Correlation heat map of VDH15 primary tumour samples subjected to RNA-seq. **e-f**. Volcano plots of differentially expressed genes in VDH15 primary tumours infected with shCtr or the *NSUN3* shRNAs sh#1 (e) or sh#2 (f). **g**. Overlap of differentially expressed genes in tumours infected with shNSUN3#1 or #2. **h**. GO analysis (GOrilla) for commonly differentially

expressed genes shown in (g) (n = 1708). P value: Exact p-value for the observed enrichment. **i**. Heat map showing differentially expressed ($p \leq 0.05$) transcripts from GO:0003735 in control (Ctr) or shNSUN3#1/#2 (NSUN3) primary tumours. **j**. Principal component (PC) analysis using expression values of mitochondrial regulators (MitoCarta2.0) in shCtr or shNSUN3 (sh#1, sh#2) tumour cells. **k**. Correlation of mitochondrial regulators expression in shNSUN3#1 and shNSUN3#2 tumour cells ($r^2$= coefficient of determination). Shown is mean +/- SD (c). Two-tailed unpaired t-test (c). Wald test (e-g). Pearson r (k). Exact p-values are indicated.

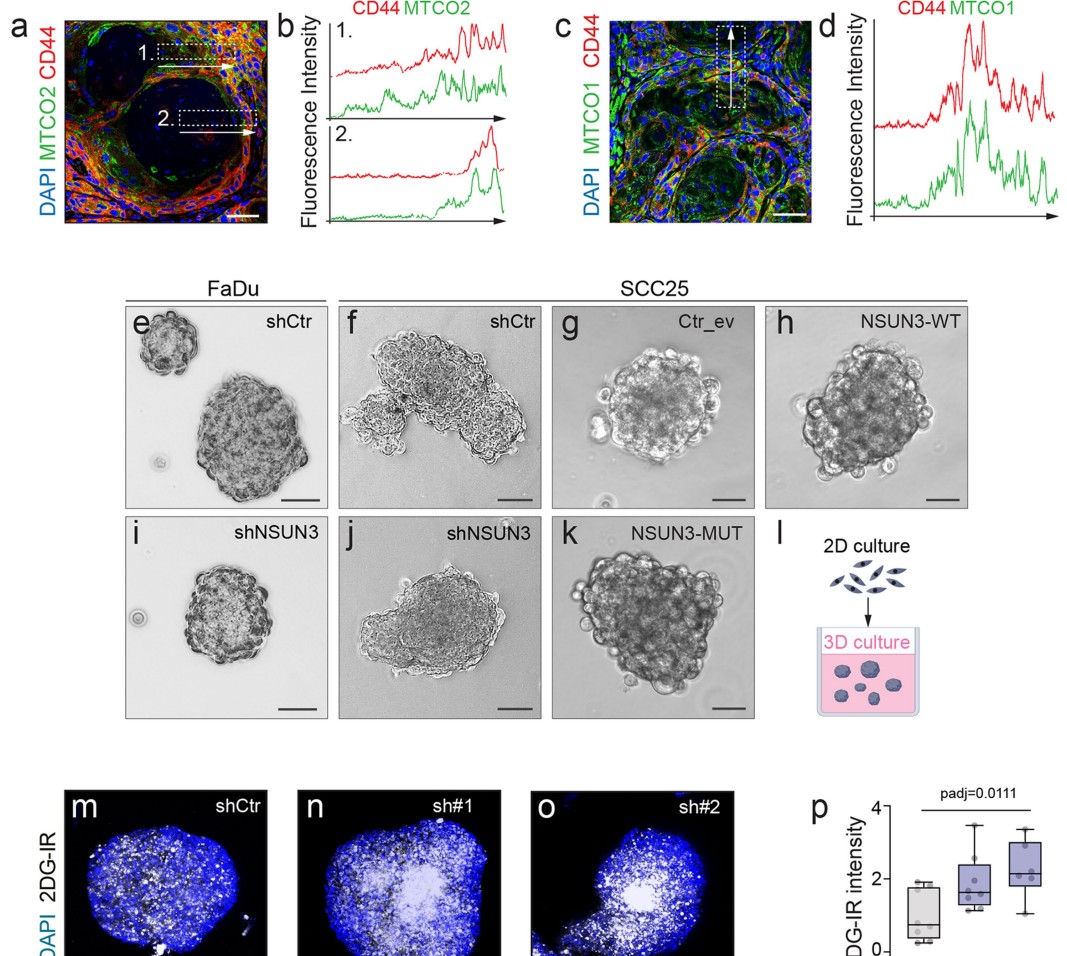

**Extended Data Fig. 6 | NSUN3 is dispensable for tumoroid growth.**
**a-d.** Immunostaining (a,c) and fluorescence intensity (b,d) of MT-CO2 or
MT-CO1 and CD44 in the outlined area (dotted square) in SCC25 primary
tumours at day 21. Shown are representative stainings from 4 mice. **e-l.** Bright
field images (e-k) and illustration of 3D cultures (l) of tumoroids derived from
the indicated tumour cells infected with shRNA control vector (shCtr), shRNA
targeting NSUN3 (shNSUN3), control empty vector (Ctr_ev), NSUN3 wild-type
(WT) or NSUN3 mutated overexpression construct (MUT) after 7 days. Shown

are representative images from 3 independent experiments. **m-p.** Representative
stainings (m-o) and quantification (p) of glucose uptake using 2DG-IR in control
(shCtr) or *NSUN3* shRNAs (sh#1, #2) FaDu-derived tumoroids (shCTR, sh#1:
$n$ = 8; sh#2 $n$ = 6 tumoroids). DAPI: nuclear counterstain (a,c). Scale bar: 50 μm
(a,c,e,f,I,j); 30 μm (g,h,k); 100 μm (m-o). Box plot shows minimum, first
quartile, median, third quartile, and maximum (p). Two-sided Dunnett's test
(p). Exact p-values are indicated.

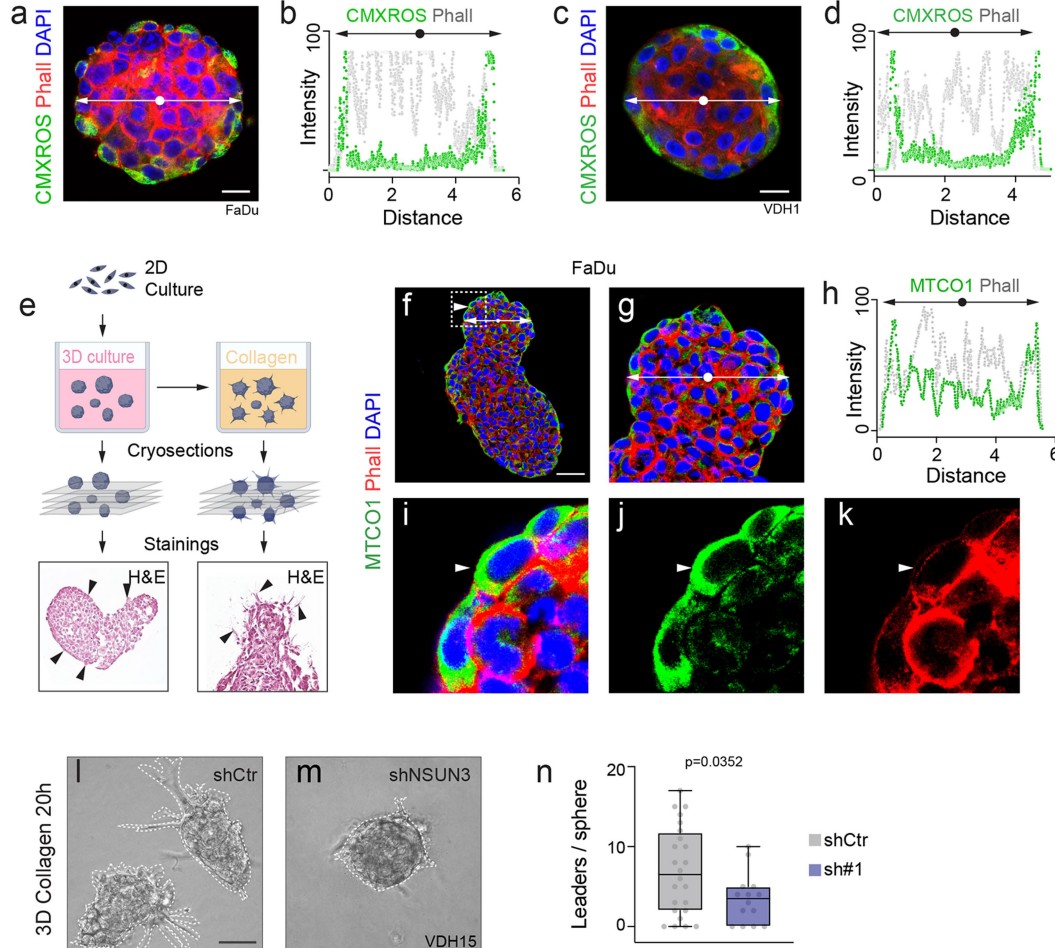

**Extended Data Fig. 7 | Spatial mitochondrial activity in tumoroids.**
**a-d.** Immunofluoresence (a,c) and quantification (b,d) of MitoTracker
(CMXROS) and Phalloidin (Phall) in 7 days cultured Fadu (a,b) or VDH01
(c,d)-derived tumoroids. **e.** Schematic representation of 3D culture assays and
subsequent sectioning. Hematoxylin & Eosin staining of sectioned tumoroids
(lower left) and invading tumoroids (lower right). **f-k.** Immunofluorescence
(f,g,i,j,k) and quantification (h) of MT-CO1 and Phalloidin (Phall) in sectioned

7 days cultured FaDu-derived tumoroids. Shown are representative images
from at least 3 independent experiments (a,c,f,g,i-k). **l-n.** Representative
images (l,m) and quantification (n) of invading leader cells per tumoroid in
shCtr and shNSUN3 infected VDH15 tumoroids. (shCTR $n$ = 23; sh#1 $n$ = 13
tumoroids). Scale bars: 30 μm (a); 40 μm (f,l,m); 50 μm (c). Box plot shows
minimum, first quartile, median, third quartile, and maximum (n). Two-tailed
unpaired t-test (n). Exact p-values are indicated.

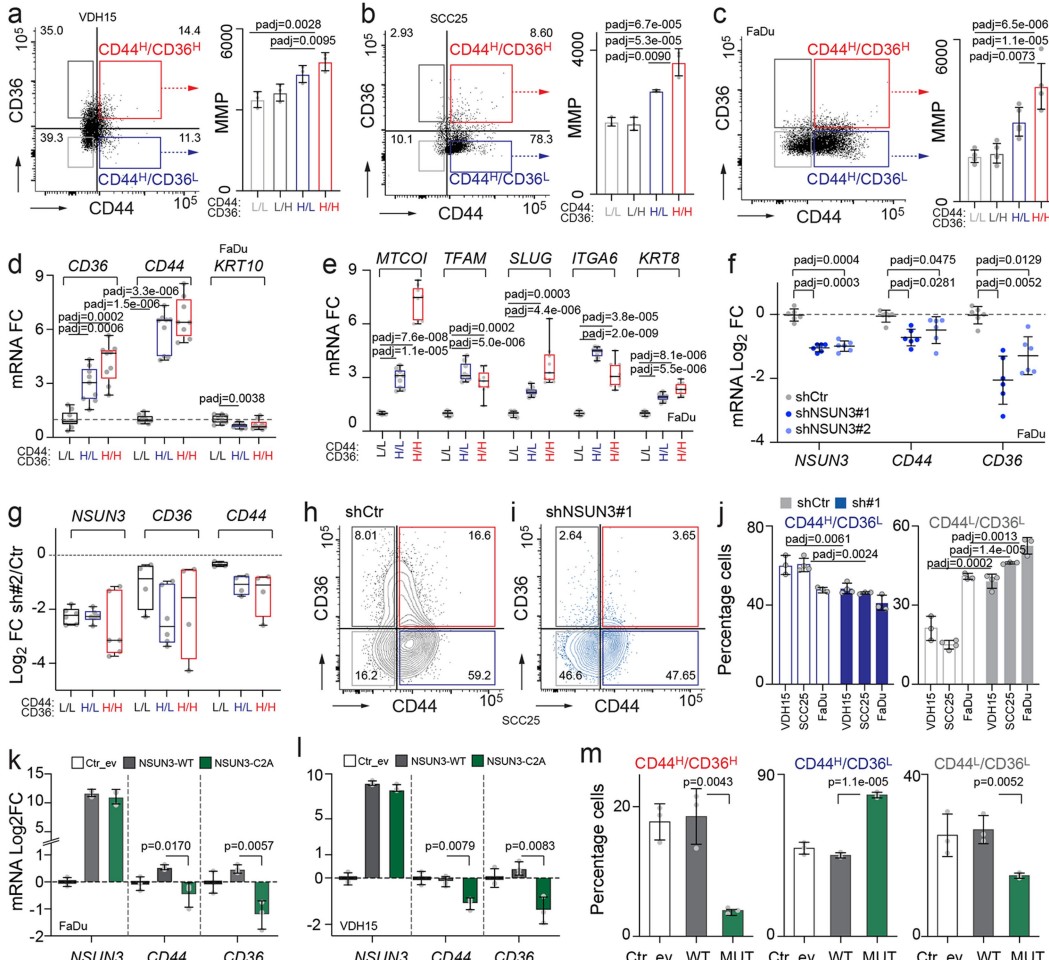

**Extended Data Fig. 8 | Metastasis-initiating cells require NSUN3. a-c.** Flow cytometry of VDH15 (a), SCC25 (b) and VDH01 (c) cells for CD36 and CD44 (left panels) and quantification of the MMP (right panels; *n* = 3 flow sorts) in the indicated subpopulations. **d**, **e**. Fold change (FC) of mRNA levels of CD36, CD44, KRT10 (d) and MT-CO1, TFAM, SLUG, ITGA6 and KRT8 (e) in FaDu subpopulations (*n* = 9: 3 RT–qPCRs from 3 infections). **f.** Log$_2$ FC in RNA levels of NSUN3, CD44 and CD36 in FaDu cells infected with a control shRNA (shCtr) or shRNAs for NSUN3 (sh*NSUN3*#1/#2) (*n* = 6: 2 RT–qPCRs from 2 infections). **g.** Log$_2$ FC of NSUN3, CD36 and CD44 RNA levels in FaDu subpopulations infected with sh#2 relative to shCtr (*n* = 6 or 4 RT–qPCRs from two flow sorts). **h-i.** Representative flow cytometry plots for CD36 and CD44 from tumour cells expressing (shCtr) (h) or lacking (sh*NSUN3*#1) NSUN3 (i). **j.** Percentage of cells

in the indicated sorted populations in from VDH15, SCC25 and FaDu tumoroids infected with shCtr or sh#1 (shCtr VDH15, shCtr FaDu, sh#1 SCC25, sh#1 FaDu: n = 3 flow sorts; shCtr SCC25, sh#1 VDH15 n = 4 flow sorts). **k-l.** Log$_2$ FC of NSUN3, CD44 and CD36 RNA levels in FaDu (k) or VDH15 (l) cells infected with an empty vector control (Ctr_ev), NSUN3 wild-type (WT) or NSUN3 mutated (MUT) overexpressing construct (*n* = 3 RT–qPCRs). **m.** Percentage of cells in the indicated sorted populations from VDH15 tumoroids infected with Ctr_ev, NSUN3$^{WT}$ or NSUN3$^{MUT}$ constructs (*n* = 3 infections). Shown is mean +/- SD (a-c,f,k-m). Box plot shows minimum, first quartile, median, third quartile, and maximum (d,e,g). Two-sided Tukey's test (a-f), or Šídák's test (j). Two-tailed Unpaired t-test (k-m). Exact p-values are indicated.

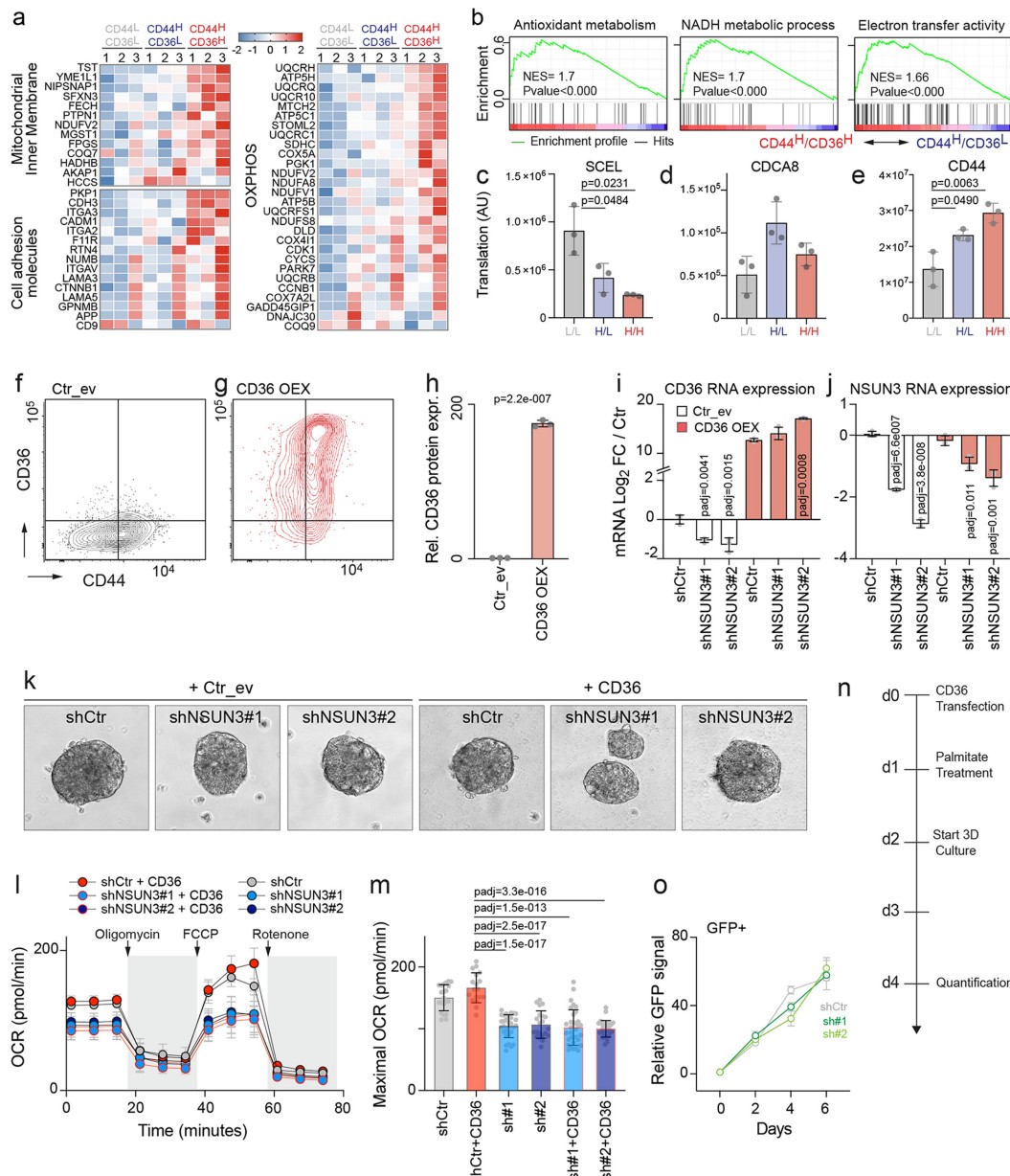

**Extended Data Fig. 9 | A mitochondria-driven proteome defines metastasis-initiating cells. a.** Heat map of differentially translated mRNAs (p < 0.05) encoding proteins of the mitochondrial inner membrane, the oxidative phosphorylation pathway or cell adhesion molecules in the indicated CD44/CD36 subpopulation. H: high; L: low. **b.** GSEA on CD44^HCD36^L versus CD44^HCD36^H newly synthesized proteins showing normalized enrichment scores (NES) and adjusted p-values. **c-e.** Translation level of the differentiation marker sciellin (SCEL) (c), the proliferation marker cell division cycle associated 8 (CDCA8) (d), and CD44 (e) in the indicated cell populations. **f-h.** Representative flow cytometry plot for CD36 and CD44 (f, g) and quantification of CD36 protein levels (h) in tumour cells expressing control empty vector (Ctr_ev) or CD36 overexpression constructs (CD36 OEX). **i-j.** Relative log₂ fold change (FC) of CD36 (i) and NSUN3 (j) RNA in VDH01 cells

infected with a control shRNA (shCtr), two different shRNAs for NSUN3 (shNSUN3#1/#2), or the Ctr_ev and CD36OEX constructs. (n = 3 infections). **k.** Representative bright field images of VDH01 spheres infected with shCtr or shNSUN3#1 or #2), Ctr_ev, or CD36 OEX constructs (n = 3 infections). **l-m.** Quantification of maximal oxygen consumption rate (OCR) in VDH15 cells infected shCtr, shNSUN3#1 or #2, Ctr_ev, or CD36 OEX constructs (n = 7 injections). **n.** Treatment and time line for invasion assays. **o.** Quantification of the GFP signal in cells over time after infection with shCTR or shRNA targeting NSUN3 (n = 3 infections). Shown is mean +/- SD (c-e, h-j,l-o). Random permutations (b). Unpaired, two-sided Student's t-test in which P values were adjusted by Benjamini-Hochberg FDR correction (c-e). Unpaired two-tailed t-test (h) or two-sided Šídák's test (i,j,m). Exact p-values are indicated c-e,h-j,l,m.

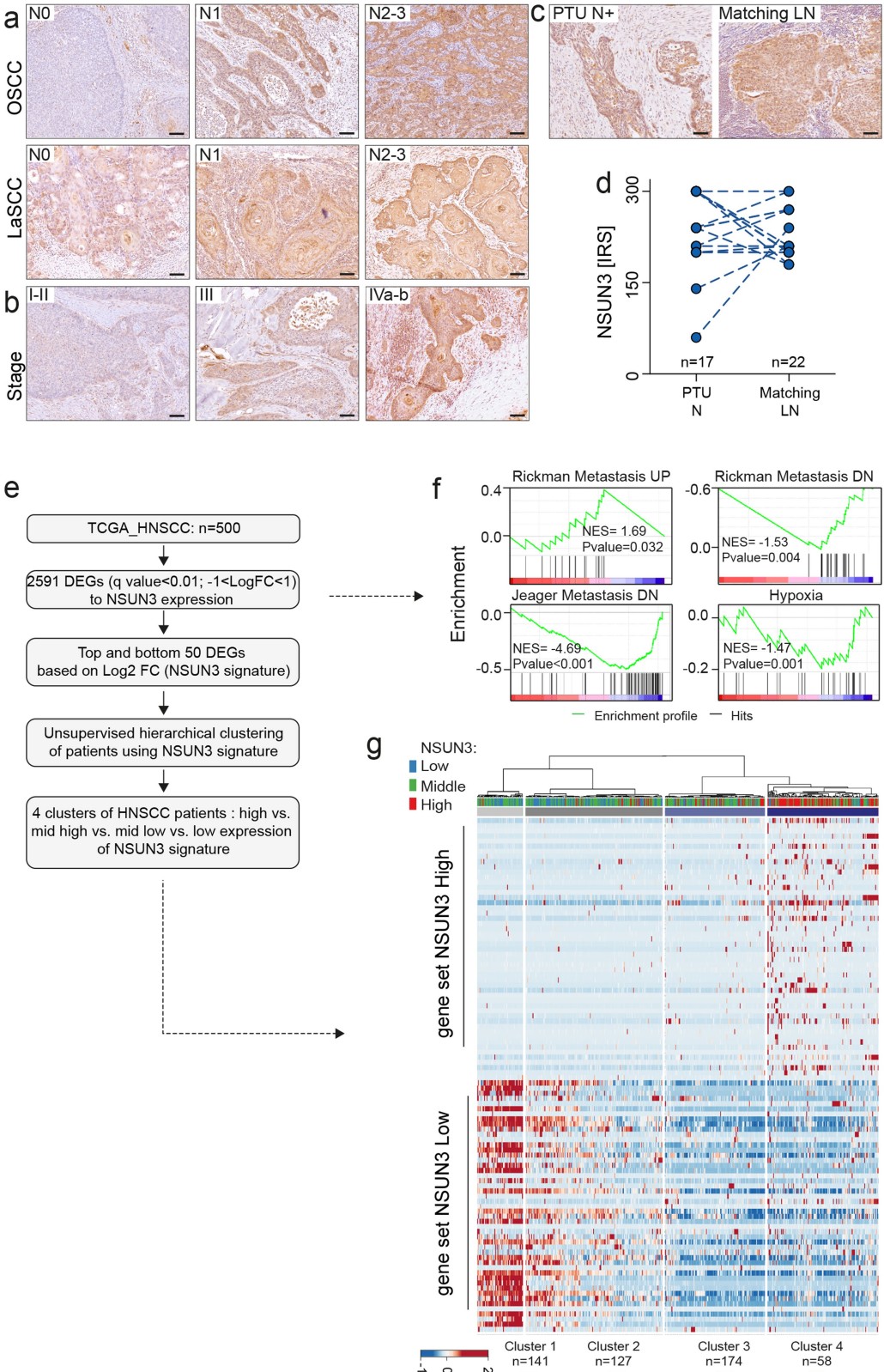

**Extended Data Fig. 10 | NSUN3-driven gene signatures in patients with HNSCC. a, b**. Representative immunohistochemistry for NSUN3 in primary oral (OSCC) or laryngeal (LaSCC) squamous carcinomas from 28 patients classified by pathological N stage with no (N0), 19 patients with one (N1: 19) or 32 patients with multiple (N2-3) metastases (a) or in the indicated pathological stages (I–II: $n$ = 25 patients, III–IVa-b: $n$ = 52) (b). Scale bar: 100 µm.
**c, d**. Immunohistochemistry (c) and quantification (d) of NSUN3 protein levels in primary tumours (PTU; $n$ = 17) and matching lymph node metastases (LN; $n$ = 22) at the time of diagnosis. Dotted line: matching LN metastases.
**e**. Flowchart to identify 4 clusters (C1-C4) of HNSCC patients (TCGA) based on NSUN3-driven gene signatures. **f**. Normalized enrichment scores (NES) and adjusted p-values (random permutations) for the indicated differentially expressed gene sets. **g**. Heat map showing 4 clusters (C1-C4) for unsupervised hierarchical clustering based on transcript levels of top 50 genes (gene set NSUN3High) and bottom 50 genes (gene set NSUN3Low). NSUN3 expression in each tumour is represented as low (blue), middle (green), and high (red).

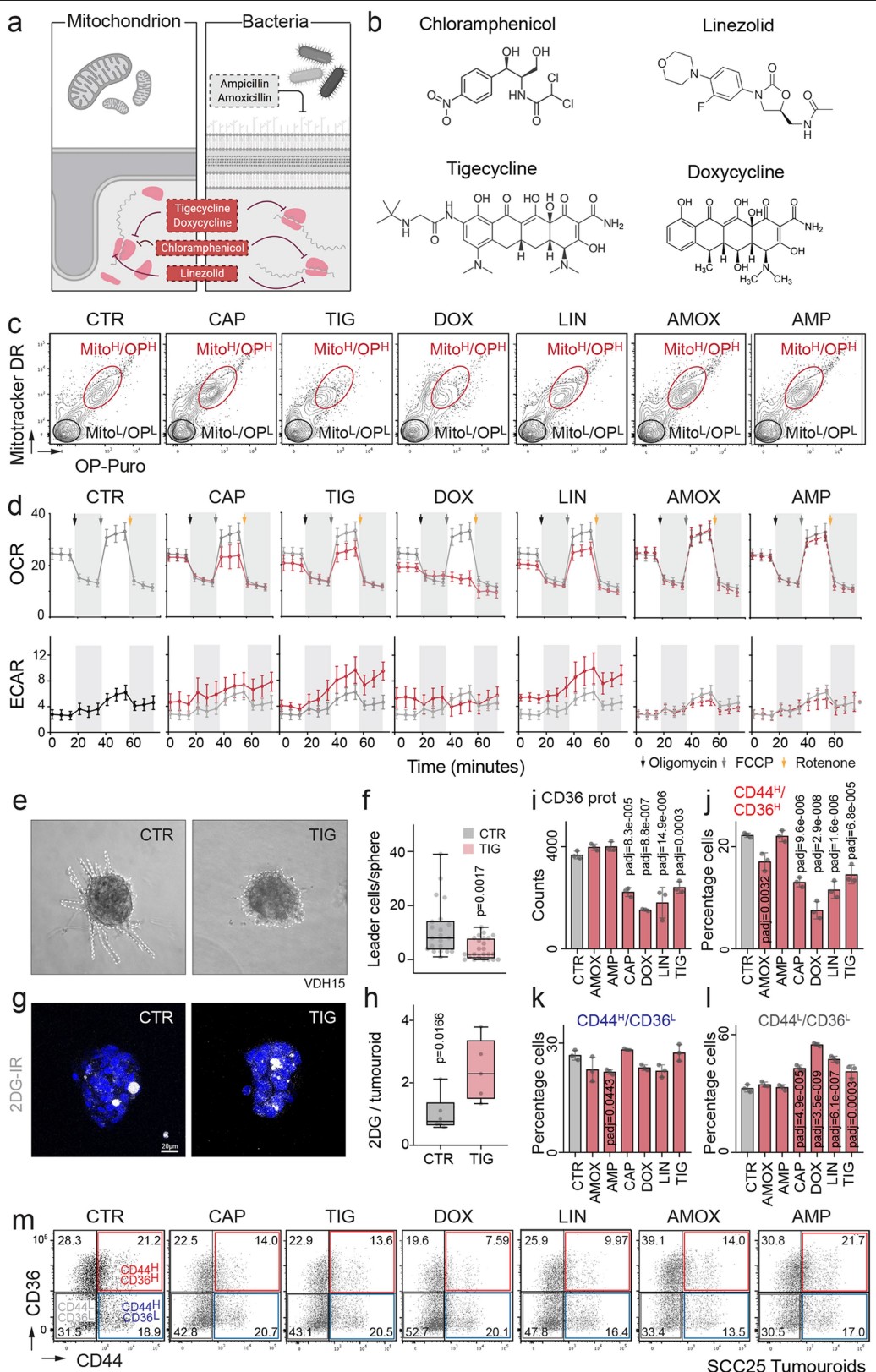

**Extended Data Fig. 11 |** See next page for caption.

**Extended Data Fig. 11 | Pharmacological inhibition of mitochondrial translation mimics loss of mitochondrial m⁵C and f⁵C. a**. Schematic overview of inhibition of mitochondrial translation by selected antibiotics. **b**. Molecular structure of the indicated antibiotics. **c**. Flow cytometry plots for MPP (MitoTracker DR) and OP-puro incorporation in FaDu cells exposed to the indicated antibiotics or vehicle control (CTR) for 48 h. **d**. Oxygen consumption rate (OCR; upper panels) and extracellular acidification rate (ECAR; lower panels) in CTR or antibiotics-treated cells (CTR: $n = 11$, CAP: $n = 8$, TIG: $n = 5$, DOX: $n = 8$, LIN: $n = 6$, AMOX: $n = 8$, AMP: $n = 6$ independent injections). **e**, **f**. Collagen-invading VDH01-derived tumoroids treated with tigecycline (TIG) or vehicle control (CTR) for 48 h (e) and quantification of leader cells (f) (CTR: $n = 20$, TIG: $n = 23$ tumoroids). **g**, **h**. Immunolabelling (g) and quantification (h) of glucose uptake through 2DG-IR in CTR- or TIG-treated tumoroids for 48 h (CTR: $n = 6$, TIG: $n = 5$ tumoroids). **i-m**. CD36 protein expression (i) and quantification of the indicated CD36/CD44 cell populations (j-l) in flow-sorted SCC25-derived tumoroids (m) (n = 3 drug treatments). Shown is mean +/- SD (d, i-l). Box plot shows minimum, first quartile, median, third quartile, and maximum (f,h). Two-tailed unpaired t-test (f,h) or two-sided Šídák's test (j-l). Exact p-values are indicated.

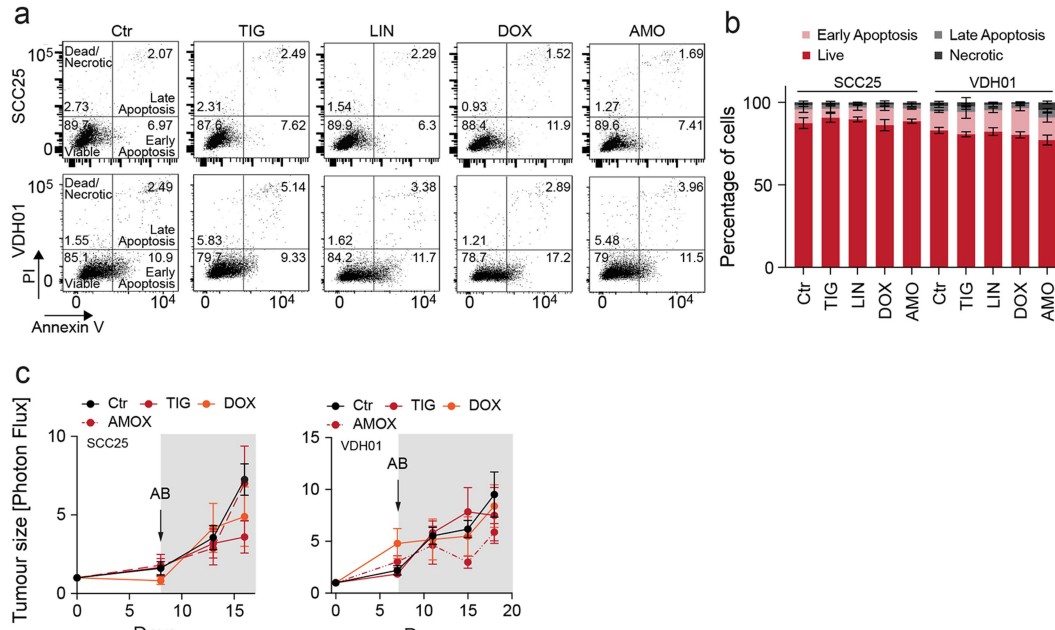

**Extended Data Fig. 12 | Cell survival and tumour growth are unaffected by inhibition of mitochondrial translation. a, b.** Representative flow cytometry plots (a) and percentages (b) of live or dead cells after Annexin V and PI staining of indicated cell lines treated for 48 h with indicated antibiotics ($n$ = 3 drug treatments) **c.** Tumour size quantified by measuring the relative photon flux of the luminescence signal at the indicated time points in SCC25- (left panel) or VDH01 (right panel)-derived tumours (SCC25 CTR: n = 8 mice, AMOX: n = 9 mice, DOX: n = 8 mice, DOX: n = 9 mice; VDH01 CTR, TIG, DOX: n = 8 mice, AMOX: n = 9 mice). Shown is mean +/- SD (b,c).

# Reporting Summary

Nature Research wishes to improve the reproducibility of the work that we publish. This form provides structure for consistency and transparency in reporting. For further information on Nature Research policies, see our Editorial Policies and the Editorial Policy Checklist.

## Statistics

For all statistical analyses, confirm that the following items are present in the figure legend, table legend, main text, or Methods section.

| n/a | Confirmed | |
|---|---|---|
| ☐ | ☒ | The exact sample size (*n*) for each experimental group/condition, given as a discrete number and unit of measurement |
| ☐ | ☒ | A statement on whether measurements were taken from distinct samples or whether the same sample was measured repeatedly |
| ☐ | ☒ | The statistical test(s) used AND whether they are one- or two-sided<br>*Only common tests should be described solely by name; describe more complex techniques in the Methods section.* |
| ☒ | ☐ | A description of all covariates tested |
| ☐ | ☒ | A description of any assumptions or corrections, such as tests of normality and adjustment for multiple comparisons |
| ☐ | ☒ | A full description of the statistical parameters including central tendency (e.g. means) or other basic estimates (e.g. regression coefficient) AND variation (e.g. standard deviation) or associated estimates of uncertainty (e.g. confidence intervals) |
| ☐ | ☒ | For null hypothesis testing, the test statistic (e.g. *F*, *t*, *r*) with confidence intervals, effect sizes, degrees of freedom and *P* value noted<br>*Give P values as exact values whenever suitable.* |
| ☒ | ☐ | For Bayesian analysis, information on the choice of priors and Markov chain Monte Carlo settings |
| ☐ | ☒ | For hierarchical and complex designs, identification of the appropriate level for tests and full reporting of outcomes |
| ☐ | ☒ | Estimates of effect sizes (e.g. Cohen's *d*, Pearson's *r*), indicating how they were calculated |

*Our web collection on statistics for biologists contains articles on many of the points above.*

## Software and code

Policy information about availability of computer code

| Data collection | IVIS Spectrum bioluminescence images were acquired with Living Image v4.4 (PerkinElmer).<br>Confocal microscope images were acquired with LAS X v1.8.1.13759 (Leica).<br>Flow cytometry data were collected using BD FACSDiva 8.0.<br>Electron Microscopy of cell sections were observed in a Zeiss EM 910 at 120kV (Carl Zeiss, Oberkochen, Germany) and micrographs taken using a slow scan CCD camera (TRS, Moorenweis, Germany).<br>The collection of the oxygen consumption rate and acidification rate values were done on the software Wave Desktop 2.6.1.<br>For RT-qPCR, QuantStudio 5 qPCR software (Applied Biosystems) was used.<br>Metabolic data was acquired using Empower3 software suite (Waters). |
|---|---|
| Data analysis | Graphpad Prism 9 was used for the statistics and generation of graphs.<br>Flow cytometry data analysis was done with FlowJo V10 (Treestar).<br>Bioluminescence data were analyzed using Living Image software V4.4.<br>Finch TV1.5 was used to analyse bisulfite treated sequences.<br>Proteomics raw files were analyzed using Proteome Discoverer (PD) 2.4 software (ThermoFisher Scientific), SequestHT node and Percolator.<br>Gene Ontologies were determined using GSEA v4.0.3 software, ToppGene (https://toppgene.cchmc.org/), or GOrilla (http://cbl-gorilla.cs.technion.ac.il/).<br>Clusteranalyses was done using Clustvis 2.0 (https://biit.cs.ut.ee/clustvis/).<br>Quantification of signal from immunofluorescence images was done with ImageJ2 FIJI Version 2.1.0/153c.<br>RNAseq analysis used the following software: FastX toolkit 0.0.13; Homertools 4.7; STAR v2.3; PicardTools 1.78; featureCounts v1.4.5-p1; R v3.3.2 EdgeR v3.16; DESeq2(v1.4.1).<br>For in silico analyses of TCGA-HNSCC tumours, the following softwares were used: EdgeR, "GSVA" package in R, maxstat in R.<br>For Bisulfite sequencing analysis, the sequences were trimmed with 'TrimGalore !' and aligned with Bismark. The R packages RSamtools, GenomicAlignments, and VIM. |

Custom perl script to compare groups in RNA-seq input table can be found here: https://zenodo.org/search?page=1&size=20&q=6518420

For manuscripts utilizing custom algorithms or software that are central to the research but not yet described in published literature, software must be made available to editors and reviewers. We strongly encourage code deposition in a community repository (e.g. GitHub). See the Nature Research guidelines for submitting code & software for further information.

## Data

Policy information about availability of data

All manuscripts must include a data availability statement. This statement should provide the following information, where applicable:

- Accession codes, unique identifiers, or web links for publicly available datasets
- A list of figures that have associated raw data
- A description of any restrictions on data availability

Quantitative proteomics data are available on PRIDE (PXD021835). RNA sequencing data using VDH01 and VDH15 cells are available on EGA under the accession number EGAS00001004765 including the attached studies EGAD00001008743 and EGAD00001008742. All other sequencing data are deposited on GEO under the accession number GSE201993. Results are in part based on TCGA-HNSC (accession number phs000178) downloaded from TCGA (https://portal.gdc.cancer.gov).

# Field-specific reporting

Please select the one below that is the best fit for your research. If you are not sure, read the appropriate sections before making your selection.

☒ Life sciences ☐ Behavioural & social sciences ☐ Ecological, evolutionary & environmental sciences

For a reference copy of the document with all sections, see nature.com/documents/nr-reporting-summary-flat.pdf

# Life sciences study design

All studies must disclose on these points even when the disclosure is negative.

| | |
|---|---|
| Sample size | The number of orthotopic transplantation assays per condition was chosen to be a minimum of 5 biological replicates per condition. This number is based on previous experiences in animal models and publications (e.g. Pascual et al., Nature, 2017). All experiments were generally powered to detect differences greater than 20% at a significance of p<0.05. For targeted and genome-wide gene expression analyses of cultured cells, our study used a minimum of 3 replicates (one replicate = one independent infection). This sample size was estimated using previous NGS datasets (e.g. Blanco et al. 2016 Nature; Selmi et al. 2021 NAR). Flow cytometry, RT-qPCRs and seahorse assays were additionally performed in at least 3 technical replicates. |
| Data exclusions | No data were excluded. |
| Replication | All experiments were repeated at least twice and all repeats were successful. |
| Randomization | Experimental animals were randomly assigned to each experimental cohorts. For cell culture experiments, cells were equally distributed into multi-plate wells and the treatment condition was randomly applied. No additional controls for covariates was performed as the mice were age- and gender-matched and cells from the same passages were used for the experiments. |
| Blinding | Animal, cellular, flow cytometry and fluorescent quantifications were performed in a blinded manner. Bisulfite sequenced samples were processed in a blinded manner. |

# Reporting for specific materials, systems and methods

We require information from authors about some types of materials, experimental systems and methods used in many studies. Here, indicate whether each material, system or method listed is relevant to your study. If you are not sure if a list item applies to your research, read the appropriate section before selecting a response.

## Materials & experimental systems

| n/a | Involved in the study |
|---|---|
| ☐ | ☒ Antibodies |
| ☐ | ☒ Eukaryotic cell lines |
| ☒ | ☐ Palaeontology and archaeology |
| ☐ | ☒ Animals and other organisms |
| ☐ | ☒ Human research participants |
| ☒ | ☐ Clinical data |
| ☒ | ☐ Dual use research of concern |

## Methods

| n/a | Involved in the study |
|---|---|
| ☒ | ☐ ChIP-seq |
| ☐ | ☒ Flow cytometry |
| ☒ | ☐ MRI-based neuroimaging |

## Antibodies

| | |
|---|---|
| Antibodies used | CD44 (Thermofisher, #14-0441-82, IM7, Lot#2093235 , 1:200), MTCO2 (Abcam, ab91317, Rabbit polyclonal, 1:200), MTCO1 |

| Antibodies used | (Thermofisher, Rabbit Polyclonal, PA5-26688, Lot#SA100601BX, 1:200), cytokeratin 10 (Biolegend, PRB-159P, Poly19054, Lot#B284664, 1:200) and filaggrin (Covance, PRB-417P-100, Poly19058, Lot#B257576, 1:200), HSP90 (Santa Cruz, sc-13119, F-8, Lot#J2616, 1:1000) . NSUN3 (Genetex, GTX46175, Rabbit Polyclonal, Lot#822004446 1:100) or anti-GLUT1 (Abcam, Ab15309, Rabbit polyclonal, Lot#GR3266142-1, 1:100). The specificity of antibody staining was confirmed by IHC staining with using a rabbit IgG isotype control antibody (Cell Signaling Technology, #3900, DA1E, 1:1000). PE-Cy7-conjugated CD44 (BD Pharmingen, #560533, G44-26, Lot#0037983, 1:300), FITC- or eFluor 660- conjugated CD36 (BD Bioscience, 1:500, , #555454 and #50-0369-42, NL07, Lot#2303260, Thermofisher). |
|---|---|
| Validation | All antibodies are commercially available and have been validated in previously published studies:<br><br>Anti-CD44 (#14-0441-82, Thermofisher). This monoclonal antibody recognizes all human CD44 isoforms (clone IM7) and is reported in 213 studies. https://www.thermofisher.com/antibody/product/CD44-Antibody-clone-IM7-Monoclonal/14-0441-82<br><br>anti-MTCO2 (ab91317, Abcam). This polyclonal antibody recognizes the human MT-CO2, is validated by the manufacturer for WB application and is used in 7 different pubblished studies. https://www.abcam.com/mtco2-antibody-ab91317.html<br><br>anti-MTCO1 (PA5-26688, Thermofisher). This polyclonal antibody recognizes the human MTCO1 protein, is validated by the manufacturer for WB, IHC and immunofluorescence applications. https://www.thermofisher.com/antibody/product/MTCO1-Antibody-Polyclonal/PA5-26688<br><br>Anti-Cytokeratin 10 (PRB-159P, Bioilegend). This polyclonal antibody recognizes the human protein cytokeratin 10 and is validated by the manufacturer. https://www.biolegend.com/en-us/products/keratin-10-polyclonal-antibody-purified-10952<br><br>Anti-filaggrin (PRB-417P-100, Covance). This polyclonal antibody recognizes the human protein of filaggrin and is validated by the manufacturer. https://www.biolegend.com/en-us/products/filaggrin-polyclonal-antibody-purified-10943<br><br>anti-NSUN3 (Genetex, GTX46175). This polyclonal antibody recognizes the human protein NSUN3 and has been validated by us through NSUN3-knock-down experiments in this study and the manufacturer for WB application. https://www.genetex.com/Product/Detail/NSUN3-antibody-C-term/GTX46175<br><br>anti-GLUT1 (Abcam, Ab15309). This polyclonal antibody recognizes the human GLUT1 protein and has been used in 103 different published studies. https://www.abcam.com/glucose-transporter-glut1-antibody-ab15309.html<br><br>anti-HSP90 ((Santa Cruz, sc-13119). This monoclonal antibody recognizes the human HSP90 protein and has been used in 567 different published studies. https://www.scbt.com/fr/p/hsp-90alpha-beta-antibody-f-8<br><br>IgG isotype control antibody (DA1E, Cell Signaling Technology). This isotype control antibody  are used to estimate the non specific binding of target primary antibodies due to Fc receptor binding or protein-protein interaction. It is validated by the manufacturer. https://www.cellsignal.com/products/primary-antibodies/rabbit-da1e-mab-igg-xp-isotype-control/3900?Ntk=Products&Ntt=3900<br><br>PE-Cy7-conjugated CD44 (BD Pharmingen, #560533). This conjugated antibody react with the human protein of CD44 and is validated by the manufacturer for flow cytometry. https://www.bdbiosciences.com/br/applications/research/t-cell-immunology/t-follicular-helper-tfh-cells/surface-markers/human/pe-cy7-mouse-anti-human-cd44-g44-26-also-known-as-c26/p/560533<br><br>FITC- or eFluor 660- conjugated CD36 (BD Bioscience, #555454 and 50-0369-42, Thermofisher). This conjugated antibody recognizes the human CD36 protein and is validated by the manufacturer for flow cytometry application. https://www.bdbiosciences.com/eu/applications/research/stem-cell-research/hematopoietic-stem-cell-markers/human/negative-markers/fitc-mouse-anti-human-cd36-cb38-also-known-as-nl07/p/555454. https://www.thermofisher.com/antibody/product/CD36-Antibody-clone-eBioNL07-NL07-Monoclonal/50-0369-42. |

## Eukaryotic cell lines

Policy information about cell lines

| Cell line source(s) | SCC25, FaDu, and LentiX 293T were obtained from ATCC (https://www.lgcstandards-atcc.org). NHEK cell lines were obtained from Promocell (https://promocell.com/product/normal-human-epidermal-keratinocytes-nhek/). Biological samples to generate the patient-derived lines VDH01 and VDH15 were obtained from patients from the Hospital Vall d'Hebron (Barcelona, Spain) under informed consent and approval of the Bank of Tumour Committees of the hospital according to Spanish ethical regulations. The study followed the guidelines of the Declaration of Helsinki, and patient identity and pathological specimens remained anonymous in the context of the study. |
|---|---|
| Authentication | No other independent authentification was performed |
| Mycoplasma contamination | All cell lines were test for mycoplasma contamination on a regular basis. Cells used in this study were tested negative. |
| Commonly misidentified lines<br>(See ICLAC register) | No commonly misidentified lines were used in this study |

## Animals and other organisms

Policy information about studies involving animals; ARRIVE guidelines recommended for reporting animal research

| Laboratory animals | 6-8 weeks old female NSG (NOD.Cg-PrkdcSCIDII2rgtm1Wjl/SzJ) from Jackson Laboratory (Cat#0055579) were used in this study.  All |
|---|---|

| Laboratory animals | animals are kept in compliance with Annex III of EU Directive 2021/63 EU from birth to death. All mice are group housed under specific pathogen-free conditions in individually ventilated cages adapted to their body weight and behavioural biology. Food, water and nesting material are provided ad libitum. Cages are placed under a 12 hours light-dark cycle and room temperature, humidity, air change/hour, and noise is maintained and monitored. None of the mice were involved in any previous procedures before the study.

Daily health checks by visual inspection were performed by the animal keepers. In case of abnormalities, individual animals are labelled as "daily observation" in the database and an automated e-mail is sent to the experimenters and the respective veterinarian/person on site to ensure that no animal is kept beyond the humane endpoints defined in the respective animal licence. |
|---|---|
| Wild animals | No wild animals were used. |
| Field-collected samples | No field collected samples were used. |
| Ethics oversight | All mice were housed in the DKFZ Central Animal Laboratory. All mouse husbandry and experiments were carried out according to the local ethics committee (DKFZ and the regional council of Baden-Württemberg state, Regierungsprasidium Karlsruhe) under the terms and condition of the animal license G-351/19. |

Note that full information on the approval of the study protocol must also be provided in the manuscript.

# Human research participants

Policy information about studies involving human research participants

| Population characteristics | For this study tumour sections of 78 patients of the HIPO-HNC cohort were used for IHC staining. The median age at the time of diagnose was 61.1 years (range 39.7-82.5 years), most patients were males (n=62, 79.5%) and had a history of smoking (n=57, 73.1%). A history of alcohol abuse was recorded for 41 patients (52.6%), and HPV16-related tumors (n=24, 30.8%) were almost exclusively found in the subgroup of oropharyngeal squamous cell carcinoma (OPSCC). All patients were treated by surgery and most had adjuvant radiotherapy with (n=32, 41.0%) or without platinum-based chemotherapy (n=26, 33.3%) The median follow-up period was 39.93 months (range 1.73-78.27 months). |
|---|---|
| Recruitment | Patients of the Heidelberg Center for Personalized Oncology-Head and Neck Cancer (HIPO-HNC) cohort were treated between 2012 and 2016 at the University Hospital Heidelberg, Germany, and the cohort consists primarily of advanced HNSCC without evidence of distant metastasis at the time of diagnosis. |
| Ethics oversight | Patient samples were obtained and analyzed under protocols S-206/2011 and S-220/2016, approved by the Ethics Committee of Heidelberg University, with written informed consent from all participants. This study was conducted in accordance with the Declaration of Helsinki. |

Note that full information on the approval of the study protocol must also be provided in the manuscript.

# Flow Cytometry

## Plots

Confirm that:

☐ The axis labels state the marker and fluorochrome used (e.g. CD4-FITC).

☒ The axis scales are clearly visible. Include numbers along axes only for bottom left plot of group (a 'group' is an analysis of identical markers).

☐ All plots are contour plots with outliers or pseudocolor plots.

☒ A numerical value for number of cells or percentage (with statistics) is provided.

## Methodology

| Sample preparation | Cells were trypsinized, washed 2 times with PBS, and fixed for 10 minutes with paraformaldehyde 1% in PBS. Following 2 additional PBS washes, cells were incubated with combinations of antibodies for 30minutes.
For cell sorting, cells trypsinized, washed 2 times with PBS and incubated with combinations of antibodies for 30minutes. After incubation, cells were washed twice in PBS and analyzed or flow sorted. |
|---|---|
| Instrument | BD LSRFortessaTM analyzer or a Cell sorter (BD Biosciences). |
| Software | FlowJo V10 software, BD FACSDiva 8.0. |
| Cell population abundance | The abundance of the relevant cell populations within post-sort fractions was 80-100% in experiments. |
| Gating strategy | Cells were first gated regarding FSC/SCC to eliminate debris, followed by FSC-H/FSC-A gating to exclude the doublet. The markers of interest were used to identify the subpopulation to analyse.
For FACS sorting, cells were selected on the basis of their forward and side scatter excluding cellular debris. Doublets and dead cells were eliminated by DAPI or propidium iodide. |

☒ Tick this box to confirm that a figure exemplifying the gating strategy is provided in the Supplementary Information.

