## [Peer Review File · Nature]

Manuscript Title: Mitochondrial RNA modifications shape metabolic plasticity in metastasis

Reviewer Comments & Author Rebuttals

Reviewer Reports on the Initial Version:

Referee expertise:

Referee #1: cancer metabolism

Referee #2: oral cancer, metastasis

Referee #3: mitochondrial translation

Referee #4: mitochondrial metabolism

Referees' comments:

Referee #1 (Remarks to the Author):

In this manuscript the authors put forward evidence that dynamic changes in mitochondrial translation are necessary for primary tumor invasion and successful metastasis. They find that conditional loss of a tRNA methylation enzyme, NSUN3, impairs mitochondrial translation and causes homeostatic changes in cellular adaptation to reduced protein synthesis. Furthermore, they show that the resulting alterations in mitochondrial function cause a metabolic shift towards glycolysis which is inhibitory to cell migration and survival, but not proliferation. Through various orthotopic models, the authors support that this mechanism is both physiologically and therapeutically relevant. The mechanism is exciting especially because NSUN3 expression seems to be dynamically altered during metastasis and seems to be predictive of patient outcome.

Major Comments:

1. While several in vivo experiments have been performed, mouse numbers are very low and often only show a trend rather than statistical significance.
 - a. Figure 2B and D: Panel B seems to indicate that 10 control animals had LN but in the mets/PT ratio only 4 points seem to be shown for control animals. What about the other 6 met-PT pairs?
 - b. Figure 4N and O: The mouse numbers with 4-5 animals are very low and it seems the changes between oxphos high and low are not statistically significant. Yet this experiment is crucial for the paper and thus needs to be repeated with more mice
 - c. Figure E-G: The mouse numbers with 4-5 animals are very low. It seems that when considering the mets/PT ratio also AMOX is showing a strong trend in metastasis reduction. Thus, the experiment needs to be repeat to ensure that the antibiotics targeting translation indeed have a distinct effect over AMOX. Moreover, Can the authors exclude off-target affects from the antibiotics used?
2. NSUN3 deletion also seems to downregulate CD36. The authors need to demonstrate that the

effect they see is because of mitochondrial protein translation and not because of CD36 downregulation

3. The organoid data of the authors support the idea of a dynamic NSUN3 regulation. Is this also true in vivo for metastasis? The authors should compare NSUN3 expression in early metastatic versus late metastatic lesions to confirm their organoid data. Is it possible to check NSUN3 expression in CTCs?

Minor Comments:

1. Figure 1C should also use a mitochondria-specific loading control (i.e. TOM20, TIMM23) that supports mitochondrial proteins are reduced in shNSUN3 cells due to translation and not differences in mitochondrial number.

2. Figure 1H: Number in the labels for the conditions are unclear i.e. VDH15 #5, is this NSUN3 knockdown #5? Consider Ctr vs shNSUN3.

3. Line 90 the authors implicate enhanced ubiquitination as a mechanism that allows cells to adapt to low mitochondria translation rates. This was not supported by experimental evidence. Thus, the authors should show altered protein degradation in their model system or rephrase this.

4. Figure 2C combines the sh1/2 into a single line, the authors fail to mention how this data was handled or analyzed. Was the average photon flux of sh1 and 2 mice compared to control?

5. Figure 3 F-I: It is likely that the signal from CMXRos is highest at the surface of the tumoroids due to the inability of the compound to penetrate the spheroid. In line 135 the authors do not acknowledge this. Although it is clear the shNSUN3 reduces mitochondrial membrane potential, and their further in vivo immunohistochemical data support this, this caveat of the assay cannot be excluded.

6. The authors use several times the word "flexibility" however the word "plasticity" is more appropriate for their context.

Referee #2 (Remarks to the Author):

This is an interesting paper in which the authors show that 5-methylcytosine (m5C) in mitochondrial tRNA^{Met} is sufficient to repress mitochondrial translation and trigger the switch from oxidative phosphorylation (OXPHOS) to glycolysis. The authors demonstrate an important link between repression of NSUN3 and down regulation of both CD44 and CD36⁺ cell populations. The authors do not however, show convincingly that glycolytic tumour cells fail to metastasize and the results are predominantly limited to a 2 monoclonal cell lines in an orthotopic mouse model of human oral cancer with limited metastatic events. The "metabolic signature" in the TCGA data is problematic and the pharmacologic inhibition of mitochondrial translation is diminished but does not block metastases and is difficult to interpret.

1. There appears to be a flaw in the way figure 2 is interpreted. First, only 2 cell lines were transduced with very different outcomes in terms of tumor growth. The more substantial effect in the number and size of metastatic implants to lymph nodes and lung is in the SCC25 tumor which showed inhibition of growth and thus the differences in metastases could be attributed to this fact alone. If I read this correctly, in b the authors record any metastatic event in a small number of animals which is meaningless and in d, a small reduction in a diminishingly small number of events between 0 and

0.15 per primary tumor (ie 1 in 15 animals or less showed a metastatic lesion in a lymph node). From an interpretive point of view, the effect is not absolute at all yet the authors write “We concluded that efficient mitochondrial translation was specifically required for tumour metastasis”. It is certainly not “required” and when efficient mitochondrial translation does not occur it may be “contribute”. Likewise for the organoid data in figure 3, where efficient mitochondrial translation may contribute but is not “required” for tumour cell invasion. This also has implications for the clinical relevance since any metastases even if microscopic will eventually lead to progression and ultimately death.

2. The authors demonstrate an important link between repression of NSUN3 and down regulation of both CD44 and CD36 mRNA levels (Figure 4). Indeed, flow sorting further confirmed that the number of the CD44H/CD36H cells was about three-fold lower in the absence NSUN3 in two independent OSCC lines (Figure 4). Again, the conclusions are overdrawn when the authors’ first state that repressing mitochondrial translation was sufficient to diminish the metastasis-initiating cell population and later that repression of mitochondrial translation was sufficient to abolish the metastasis inducing tumour cell population.

3. In figure 5, the GSEA analysis of the top (j) or bottom (k) 150 genes in NSUN3-depleted primary tumours shows a very large overlap. The data in the orthotopic animals also showed a small effect on lymph nodes metastases. Lung metastases showed a larger effect and would predict for overall survival which was not measured. The gene signature data in figure 5, I was over fitted to find the best cutoff instead of separating the cohort into a test and validation set or seeking validation in an independent cohort.

4. The antibiotic treatments in figure 7 show a modest effect, reducing a very small number of events further (as discussed in number 1 above). In contrast to the earlier experiments however, the graph now shows the maximum number of metastatic events even smaller for VDH1 as 0.04 or 1 in 25 animals in the control and slightly less in treated animals. These results have to be reproduced in a larger number of animals with more metastatic events and further confirmed in more robust syngeneic models.

Referee #3 (Remarks to the Author):

In this manuscript, Sylvain Delaunay et al. demonstrate that inhibition of mitochondrial translation strongly impairs metastasis. Particularly, by inhibiting mitochondrial translation through shRNA-mediated silencing of the RNA methyltransferase NSUN3, otherwise glycolytic tumor cells from primary tumors were not able to metastasize. Moreover, they demonstrate that a distinct subpopulation of tumor cells rely on high OXPHOS activity to mediate invasion and metastasis. Moreover, inhibition of mitochondrial translation via antibiotic treatment inhibited metastasis, suggesting that blocking mitochondrial translation might be a promising pharmacological approach to counteract metastasis. Overall, the data and their presentation are of high quality, and the study reveals a conceptually exciting pathway in which metabolic reprogramming and particularly the activation of respiration is a key event for metastasis. Given that this work addresses an important

aspect of cancer biology with obvious implications for human health, this study will be of great interest to the broad readership of nature. However, there are some points that need to be addressed to clarify a few critical points:

Major points:

- The authors indicate in Figure 1 C that protein levels of the complex IV subunits MTCO1 and MTCO2 are reduced via shRNA against NSUN3 and measured overall mitochondrial translation via flow cytometric quantification of O-Propargyl-Puromycin. It would be important to test the abundance of other respiratory chain subunits containing mitochondrially encoded subunits, ie complex I, complex III and ATPsynthase
- In Figure 1 L, the authors present data regarding cell viability of the FaDu cell line, and despite a slight reduction of cells alive (at least for shRNA #1) the authors state that cellular viability is unaffected by depletion of NSUN3. It would be important to add statistical analysis to validate this conclusion. Further, the method used for analyzing cellular viability is not described in the manuscript, which should be added. Here, it would be important to choose a method that is not depending on metabolic activity (e.g. not trypan blue staining), as the changes in metabolism mediated by inhibition of OXPHOS activity might falsify the result. Further, as mitochondria play a pivotal role in mediating apoptotic cell death, an assay that includes necrotic as well as apoptotic cell death should be applied (e.g. PI/AnnexinV staining or similar). These analyses should be done with all three cell types and clearly presented. This is important, as a possible explanation for the reduction of metastasis by inhibition of mitochondrial translation might be that the leader cells, which show high OXPHOS activity, might be particularly sensitive to inhibition of mitochondrial translation and die, which would impair metastasis. Alternatively, inhibition of mitochondrial translation could impair metabolic reprogramming, thereby inhibiting the formation of activated leader cells. Both scenarios are highly interesting and it would be important to know which is correct. This is also relevant for the data presented in Figures 5a-l, where absence of the leader cells due to toxicity of NSUN3 depletion would be sufficient to explain the lack in expression of mitochondrial genes (which would be induced in these cells due to metabolic reprogramming).
- In Figure S1, the authors monitored mitochondrial morphology and quantified the number of cristae per mitochondrion via electron microscopy. Although this approach is appropriate for the analysis and quantification of submitochondrial structures, additional fluorescence microscopic analysis of mitochondria is required to validate mitochondrial morphology in vivo.
- In line 361 the authors state that their study shows a specific effect of certain antibiotics on tumors. While the authors clearly demonstrate that inhibition of mitochondrial translation counteracts metastasis, an antibiotic treatment that reduces OXPHOS activity will not be specific for cancer cells, unless specifically targeted to tumors (e.g. via nanoparticle delivery). Hence, a potential treatment with the antibiotics might cause severe general cytotoxicity as OXPHOS activity will be inhibited in healthy cells, which rely on cellular respiration for energy conversion. Although the aim of this study is to provide mechanistic insights into inhibition of metastasis, and not to provide a therapeutic solution against it, it would also be important to add a short paragraph briefly describing such limitations.

Minor Concerns:

- To make the size of tumouroids comparable in Fig. 3 d, please add respective scale bars.
- Please also add scale bars to Fig. 7 c
- The term “mitochondrial activity” in line 135 (et seqq.) is somehow misleading, as these organelles are of course also involved in many other (metabolic) processes in addition respiration.

Mitochondrial membrane potential (or MPP as defined in the line above) instead could be more accurate.

Referee #4 (Remarks to the Author):

This paper demonstrates that metastatic cells rely on mitochondrial translation more so than primary tumor. This is not a new finding. A recent study CRISPR screen demonstrated that “Mitochondrial ribosome and mitochondria-associated genes were identified as top gene sets associated with metastasis-specific lethality” (See PMID: 33239425). Furthermore, this study demonstrate that doxycycline reduced metastasis effectively but not primary tumor growth. Moreover, there are other studies that demonstrate reliance on mitochondria for metastasis (PMID: 2524103). It is important to note that in vivo screens on primary tumors also yielded genes involved in mitochondrial metabolism is necessary for growth (PMID: 33152324; PMID: 33152323).

Minor points: There should be cDNA rescue for shRNA.

2020-10-19395

In response to referees' concerns, we have made the following changes to our manuscript.

Reviewer comments:

Referee #1:

In this manuscript the authors put forward evidence that dynamic changes in mitochondrial translation are necessary for primary tumor invasion and successful metastasis. They find that conditional loss of a tRNA methylation enzyme, NSUN3, impairs mitochondrial translation and causes homeostatic changes in cellular adaptation to reduced protein synthesis. Furthermore, they show that the resulting alterations in mitochondrial function cause a metabolic shift towards glycolysis which is inhibitory to cell migration and survival, but not proliferation. Through various orthotopic models, the authors support that this mechanism is both physiologically and therapeutically relevant. The mechanism is exciting especially because NSUN3 expression seems to be dynamically altered during metastasis and seems to be predictive of patient outcome.

We thank the referee for finding our data exciting and for the very constructive criticism to improve our study.

Major Comments:

1. While several *in vivo* experiments have been performed, mouse numbers are very low and often only show a trend rather than statistical significance.
 - a. Figure 2B and D: Panel B seems to indicate that 10 control animals had LN but in the mets/PT ratio only 4 points seem to be shown for control animals. What about the other 6 met-PT pairs?
 - b. Figure 4N and O: The mouse numbers with 4-5 animals are very low and it seems the changes between oxphos high and low are not statistically significant. Yet this experiment is crucial for the paper and thus needs to be repeated with more mice.
 - c. Figure E-G: The mouse numbers with 4-5 animals are very low. It seems that when considering the mets/PT ratio also AMOX is showing a strong trend in metastasis reduction. Thus, the experiment needs to be repeat to ensure that the antibiotics targeting translation indeed have a distinct effect over AMOX. Moreover, Can the authors exclude off-target affects from the antibiotics used?

As requested by the referee, we have now increased the mouse numbers in all orthotopic transplantation assays (Figure 2a-c; Figure 6d-f; Figure S3a-f; Figure S12c; Supplementary Table 1). Moreover, each experiment was repeated in at least three different metastatic human OSCC lines using two different shRNAs. In addition, we now provide rescue experiments using NSUN3 over-expressing lines. A summary of all orthotopic transplantation assays is now provided in Supplementary Table 1.

Please note that all human OSCC cancer lines form a primary tumour with 100% penetrance, but only about 60-80% of those develop lymph node metastasis within the 21 days of the experiment. This depends on the specific OSCC line. Therefore, some are not appearing in the tumour-metastasis matched analyses.

By increasing the number of orthotopic transplantation assays in the antibiotic treatment experiment, we now confirm no significant difference between vehicle treated and AMOX treated tumours using two different OSCC lines (*Figure 6d-f; Figure S12c*).

The referee is correct, we cannot fully exclude off-target effects. However, we find no differences in cell viability in response to the antibiotic treatment (*Figure S12b*). More importantly, we also did not observe any of the known putative side effects in the mice such as swelling of the face or muzzle, skin rashes or diarrhoea. We have now included this information into the Discussion.

2. *NSUN3 deletion also seems to downregulate CD36. The authors need to demonstrate that the effect they see is because of mitochondrial protein translation and not because of CD36 downregulation.*

The referee raises an excellent point. To address this comment, we over-expressed CD36 in NSUN3-expressing and -lacking tumour cells, and repeated the Seahorse and 3D invasion assays. We now demonstrate that CD36-expression is not sufficient to rescue the metabolic differences or invasion in the absence of NSUN3 (*Figure 4lm-o; Figure S8f-n*).

3. *The organoid data of the authors support the idea of a dynamic NSUN3 regulation. Is this also true in vivo for metastasis? The authors should compare NSUN3 expression in early metastatic versus late metastatic lesions to confirm their organoid data. Is it possible to check NSUN3 expression in CTCs?*

The referee is correct, we indeed find a successive up-regulation of NSUN3 protein levels in patients with early versus late metastatic lesions (*Figure 5a,b; Figure S10a, b*).

Analysing NSUN3 functions in CTCs is an interesting suggestion. It is highly likely that mitochondria will play a role in CTCs to protect from cell death and shear stresses for instance. Interestingly, we find that primary tumours and their matching metastasis share the same level of heterogeneity with respect to CD36/CD44 and NSUN3 expression (*Figure 4h; Figure S10c, d*). Thus, even if CTCs change their metabolic requirements in the blood system, the secondary tumour re-establishes a similar metabolic profile to its matched primary tumour. Our data therefore indicate that the proportion of metabolically distinct populations within tumours is genetically determined.

Minor Comments:

1. *Figure 1C should also use a mitochondria-specific loading control (i.e. TOM20, TIMM23) that supports mitochondrial proteins are reduced in shNSUN3 cells due to translation and not differences in mitochondrial number.*

To exclude differences in mitochondrial number, we quantified the mitochondrial DNA copy numbers in NSUN3-depleted cells and find no significant differences (*Figure S2b*).

2. *Figure 1H: Number in the labels for the conditions are unclear i.e. VDH15 #5, is this NSUN3 knockdown #5? Consider Ctr vs shNSUN3.*

This mistake has now been corrected and the shRNA are consistently labelled throughout the manuscript.

3. *Line 90 the authors implicate enhanced ubiquitination as a mechanism that allows cells to adapt to low mitochondria translation rates. This was not supported by experimental evidence. Thus, the authors should show altered protein degradation in their model system or rephrase this.*

We agree with the referee and have omitted these data from the manuscript.

4. *Figure 2C combines the sh1/2 into a single line, the authors fail to mention how this data was handled or analyzed. Was the average photon flux of sh1 and 2 mice compared to control?*

Yes, the photon flux was always compared to the respective control cells. To be consistent, we now consistently separated the data from two independent shRNAs.

5. *Figure 3 F-I: It is likely that the signal from CMXROS is highest at the surface of the tumoroids due to the inability of the compound to penetrate the spheroid. In line 135 the authors do not acknowledge this. Although it is clear the shNSUN3 reduces mitochondrial membrane potential, and their further in vivo immunohistochemical data support this, this caveat of the assay cannot be excluded.*

To exclude the possibility that CMXROS was not penetrating equally through the tumouroid, we now provide sections from the spheres and confirm higher levels of CMXRAS at the edges of the 3D-cultured cells (*Figure S6e-k*).

5. *The authors use several times the word “flexibility” however the word “plasticity” is more appropriate for their context.*

We agree with the referee and have replaced “flexibility” with the word “plasticity”.

Referee #2:

This is an interesting paper in which the authors show that of 5-methylcytosine (m5C) in mitochondrial tRNAMet is sufficient to repress mitochondrial translation and trigger the switch from oxidative phosphorylation (OXPHOS) to glycolysis. The authors demonstrate a n important link between repression of NSUN3 and down regulation of both CD44 and CD36 + cell populations. The authors do not however, show convincingly that glycolytic tumour cells fail to metastasize and the results are predominantly limited to a 2 monoclonal cell lines in an orthotopic mouse model of human oral cancer with limited metastatic events. The “metabolic signature “ in the TCGA data is problematic and the pharmacologic inhibition of mitochondrial translation is diminished but does not block metastases and is difficult to interpret.

We thank the referee for finding our manuscript interesting.

The main concern of this referee seems to be that the orthotopic transplantation assay analysed a low number of metastatic events. This was partly a mis-understanding how the analysis was performed (see detailed comments below). However, we now also increased the number of orthotopic transplantation assays. In addition, we included a third independent patient-derived OSCC line. In total, we used 175 animals resulting in 96 metastatic events. The precise numbers are now provided in *Supplementary Table 1*.

We respectfully disagree with the statement that our data do not convincingly show that glycolytic tumour cells fail to metastasize. The chance to develop a lymph node metastasis dropped from on average 75% in controls down to 20% in the absence of a functional NSUN3 protein. We believe that this finding is highly clinically relevant, given that 90% of cancer patients die because of the metastases but not the primary tumour.

Please note that we are not generating knockout cells. We are reducing NSUN3 expression levels through shRNAs or by over-expressing a dominant negative enzyme. This means that some cells will escape the knockdown. The data might be therefore more variable, but nevertheless they are significant. We believe that down-regulation or inhibition rather than genetic deletion of the enzyme is far more meaningful for any putative therapeutic approach, as no drug treatment in cancer patients will achieve complete knockout of the enzyme.

1. *There appears to be a flaw in the way figure 2 is interpreted. First, only 2 cell lines were transduced with very different outcomes in terms of tumor growth. The more substantial effect in the number and size of metastatic implants to lymph nodes and lung is in the SCC25 tumor which showed inhibition of growth and thus the differences in metastases could be attributed to this fact alone. If I read this correctly, in b the authors record any metastatic event in a small number of animals which is meaningless and in d, a small reduction in a diminishingly small number of events between 0 and 0.15 per primary tumor (ie 1 in 15 animals or less showed a metastatic lesion in a lymph node). From an interpretive point of view, the effect is not absolute at all yet the authors write “We concluded that efficient mitochondrial translation was specifically required for tumour metastasis”. It is certainly not “required” and when efficient mitochondrial translation does not occur it may be “contribute”. Likewise for the organoid data in figure 3, where efficient mitochondrial translation may contribute but is not “required” for tumour cell invasion. This also has implications for the clinical relevance since any metastases even if microscopic will eventually lead to progression and ultimately death.*

The referee is correct, the outcome of tumour growth differed in the two cancer lines. Similar to tumours in patients, we expect see different levels of heterogeneity in the orthotopically transplanted tumours. To ensure the reproducibility of our *in vivo* data, we now included a third patient-derived OSCC line.

SCC25-derived primary tumours indeed showed some inhibition in growth, which might affect the development of metastasis. Precisely for that reason, we have normalized the size of the metastasis to its matching primary tumour (*Figure 2c; Figure S3c; Figure 6f*).

We apologize that our description on how some of the analyses was performed lacked clarity. The statement that “1 in 15 animals or less showed a metastatic lesion in a lymph node ” is incorrect. The transplanted cells formed primary tumours with a 100% penetrance (e.g. *Figure 2b; PT*), which resulted in about 75% of cases in lymph node metastases in controls. In contrast, NSUN3-deficient tumour cells only formed metastases in 20% of the cases (e.g. *Figure 2b; LN-Met*). The actual number of animals and tumours is now provided in *Supplementary Table 1*. In *Figure 2c, Figure S3c, and Figure 6f*, we account for tumour size (not tumour number) to exclude the possibility that we observed less metastases because the primary tumour was smaller. We have now re-written the text to better clarify how the analysis was done.

As mentioned above, we used knockdown tools but not knockout tools to down-regulate NSUN3-expression. Some NSUN3-expressing cells will escape the shRNA infections, even after selection. In fact, the small number of matching metastases (LN-Met; see graph on the right) we observed after transplanting NSUN3-depleted cells contained proportionally the same number of CD36/CD44 cells than the primary tumours (PTU) formed by control cells (grey). This data strongly indicates that those cells escaped the knockdown.

We respectfully disagree with the statement that our data is from limited clinical relevance. Depletion or inhibition of NSUN3 reduced the chance of developing a metastasis from in average 75% down to 20%. We believe that is highly clinically relevant, given that 90% of cancer patients die of metastases, but not the primary tumour.

Moreover, NSUN3 is a very promising drug target, because it is a stand-alone enzyme solely responsible for mitochondrial m⁵C formation. In addition, we already provided the proof-of-concept that mitochondrial inhibition reduces the formation of metastasis by using FDA-approved drugs.

2. The authors demonstrate an important link between repression of NSUN3 and down regulation of both CD44 and CD36 mRNA levels (*Figure 4*). Indeed, flow sorting further confirmed that the number of the CD44H/CD36H cells was about three-fold lower in the absence NSUN3 in two independent OSCC lines (*Figure 4*). Again, the conclusions are overdrawn when the authors' first state that repressing mitochondrial translation was sufficient to diminish the metastasis-initiating cell population and later that repression of mitochondrial translation was sufficient to abolish the metastasis inducing tumour cell population.

As outlined under point 1, we cannot expect an all or nothing response in our system. We believe that a three-fold significant reduction in metastasis-initiating CD44^H/CD36^H cells is convincing.

To address the referee's concern, we now use the precise effect sizes when describing the data in the results section.

In addition, we provide new data demonstrating that CD36 requires NSUN3 to activate invasion (*Figure 4m-o; Figure S8k-n*).

3. *In figure 5, the GSEA analysis of the top (j) or bottom (k) 150 genes in NSUN3-depleted primary tumours shows a very large overlap. The data in the orthotopic animals also showed a small effect on lymph nodes metastases. Lung metastases showed a larger effect and would predict for overall survival which was not measured. The gene signature data in figure 5, I was over fitted to find the best cutoff instead of separating the cohort into a test and validation set or seeking validation in an independent cohort.*

We agree with the referee. Because the data directly obtained from our patient cohort is more clinically relevant, we omitted those figure panels in former Figure 5 from the revised version of the manuscript.

Please note that there is not sufficient data about lung metastasis available in the HNSC TCGA dataset (only 1 out of 528 samples).

4. *The antibiotic treatments in figure 7 show a modest effect, reducing a very small number of events further (as discussed in number 1 above). In contrast to the earlier experiments however, the graph now shows the maximum number of metastatic events even smaller for VDH1 as 0.04 or 1 in 25 animals in the control and slightly less in treated animals. These results have to be reproduced in a larger number of animals with more metastatic events and further confirmed in more robust syngeneic models.*

We refer to our response under point 1 and 2, and again apologize for the confusion how parts of our analyses have been performed. As requested by the referee, we have increased the animal numbers in all orthotopic transplantation assays including the drug treatment (see *Figure 6d-f; Supplementary Table 1*).

Please note that we use the antibiotic as a mean to specifically reduce mitochondrial but not cytoplasmatic protein synthesis. Therefore, we needed to eliminate any side-effects the drugs may have on the immune system. For that reason, NSG were the ideal model system to answer our research question.

Referee #3:

In this manuscript, Sylvain Delaunay et al. demonstrate that inhibition of mitochondrial translation strongly impairs metastasis. Particularly, by inhibiting mitochondrial translation through shRNA-mediated silencing of the RNA methyltransferase NSUN3, other wise glycolytic tumor cells form primary tumors were not able to metastasize. Moreover, they demonstrate that

a distinct subpopulation of tumor cells rely on high OXPHOS activity to mediate invasion and metastasis. Moreover, inhibition of mitochondrial translation via antibiotic treatment inhibited metastasis, suggesting that blocking mitochondrial translation might be a promising pharmacological approach to counteract metastasis. Overall, the data and their presentation are of high quality, and the study reveals a conceptually exciting pathway in which metabolic reprogramming and particularly the activation of respiration is a key event for metastasis. Given that this work addresses an important aspect of cancer biology with obvious implications for human health, this study will be of great interest to the broad readership of nature. However, there are some points that need to be addressed to clarify a few critical points:

We agree with the referee that our study reveals a conceptually exciting pathway, in which metabolic reprogramming and particularly the activation of respiration is a key event for metastasis.

Major points:

- The authors indicate in Figure 1 C that protein levels of the complex IV subunits MTCO1 and MTCO2 are reduced via shRNA against NSUN3 and measured overall mitochondrial translation via flow cytometric quantification of O-Propargyl-Puromycin. It would be important to test the abundance of other respiratory chain subunits containing mitochondrially encoded subunits, ie complex I, complex III and ATPsynthase*

We agree with the referee that the abundance of other respiratory chain subunits should be measured in addition to the OP-puro experiments. Therefore, we also show a decrease of all mitochondria-encoded subunits on RNA level in Figure 2g. Vice versa, we find mitochondrial inner membrane proteins up-regulated on protein level in cell populations expressing the highest levels of NSUN3 (i.e. the CD44^H/CD36^H population; Figure S8a).

- In Figure 1 L, the authors present data regarding cell viability of the FaDu cell line, and despite a slight reduction of cells alive (at least for shRNA #1) the authors state that cellular viability is unaffected by depletion of NSUN3. It would be important to add statistical analysis to validate this conclusion. Further, the method used for analyzing cellular viability is not described in the manuscript, which should be added. Here, it would be important to choose a method that is not depending on metabolic activity (e.g. not trypan blue staining), as the changes in metabolism mediated by inhibition of OXPHOS activity might falsify the result. Further, as mitochondria play a pivotal role in mediating apoptotic cell death, an assay that includes necrotic as well as apoptotic cell death should be applied (e.g. PI/AnnexinV staining or similar). These analyses should be done with all three cell types and clearly presented. This is important, as a possible explanation for the reduction of metastasis by inhibition of mitochondrial translation might be that the leader cells, which show high OXPHOS activity, might be particularly sensitive to inhibition of mitochondrial translation and die, which would impair metastasis. Alternatively, inhibition of mitochondrial translation could impair metabolic reprogramming, thereby inhibiting the formation of activated leader cells. Both scenarios are highly interesting and it would be important to know which is correct. This is also relevant for the data presented in Figures 5a-I, where absence of the leader cells due to toxicity of NSUN3 depletion would be sufficient to explain the lack in expression of mitochondrial genes (which would be induced in these cells due to metabolic reprogramming).*

The referee raises a very important point, and we have now included cell viability assays using PI/AnnexinV staining in all relevant experiments (*Figure S2a; Fig. S12b*). We do not find cell viability to be affected.

As requested by the referee, we also analysed whether the metastasis-initiating cell population (CD44^H/CD36^H) is undergoing cell death in the absence of NSUN3. We find cell viability to be unaffected by depletion of NSUN3 (*Figure 4p*).

• *In Figure S1, the authors monitored mitochondrial morphology and quantified the number of cristae per mitochondrion via electron microscopy. Although this approach is appropriate for the analysis and quantification of submitochondrial structures, additional fluorescence microscopic analysis of mitochondria is required to validate mitochondrial morphology in vivo.*

To address the referee's comment, we now include fluorescence images of the mitochondria in cells expressing wild-type or methylation-deficient NSUN3 constructs. This allowed us to visualise the mitochondria and to confirm that mitochondrial RNA methylation directly regulates the mitochondrial shape (*Figure S1n, o; Figure S2c, d*).

• *In line 361 the authors state that their study shows a specific effect of certain antibiotics on tumors. While the authors clearly demonstrate that inhibition of mitochondrial translation counteracts metastasis, an antibiotic treatment that reduces OXPHOS activity will not be specific for cancer cells, unless specifically targeted to tumors (e.g. via nanoparticle delivery). Hence, a potential treatment with the antibiotics might cause severe general cytotoxicity as OXPHOS activity will be inhibited in healthy cells, which rely on cellular respiration for energy conversion. Although the aim of this study is to provide mechanistic insights into inhibition of metastasis, and not to provide a therapeutic solution against it, it would also be important to add a short paragraph briefly describing such limitations.*

The referee is correct, any therapeutic approach will not specifically target cancer cells. However, we do know from the mouse experiments that the treatment with mitochondrial translation inhibitory drugs does not cause any major side effect. Over the course of the antibiotic treatment, the mice did not display any of the known side effects the antibiotics may cause such as swelling of the face or muzzle, skin rashes or diarrhoea.

Moreover, we do have strong indication for potential side effects because we identified patients carrying loss-of-function mutations of NSUN3. These patients survive, yet they present with combined mitochondrial respiratory chain complex deficiency ¹.

We have now included a paragraph describing these findings into the Discussion.

Minor Concerns:

• *To make the size of tumouroids comparable in Fig. 3 d, please add respective scale bars.*

The scale bar is now added (now *Figure 3l*).

• *Please also add scale bars to Fig. 7 c*

The scale bar is now added (now *Figure 6c*).

- The term “mitochondrial activity” in line 135 (*et seqq.*) is somehow misleading, as these organelles are of course also involved in many other (metabolic) processes in addition respiration. Mitochondrial membrane potential (or MMP as defined in the line above) instead could be more accurate.

We agree with the referee and have replaced “mitochondrial activity” with “mitochondrial membrane potential” when we refer to the MMP measurements.

Referee #4:

This paper demonstrates that metastatic cells rely on mitochondrial translation more so than primary tumor. This is not a new finding. A recent study CRISPR screen demonstrated that “Mitochondrial ribosome and mitochondria-associated genes were identified as top gene sets associated with metastasis-specific lethality” (See PMID: 33239425). Furthermore, this study demonstrate that doxycycline reduced metastasis effectively but not primary tumor growth. Moreover, there are other studies that demonstrate reliance on mitochondria for metastasis (PMID: 2524103). It is important to note that in vivo screens on primary tumors also yielded genes involved in mitochondrial metabolism is necessary for growth (PMID: 33152324; PMID: 33152323).

We strongly disagree with the referee that our study is not novel. Our major discovery is that the deposition of one single mitochondrial RNA modification directly triggers the metabolic reprogramming required for metastasis. This is a fundamentally novel concept, which in addition, offers novel therapeutic treatment strategies as NSUN3 is a stand-alone enzyme mediating mitochondrial RNA methylation¹⁻³. Thus, we also identified NSUN3 and mitochondrial m⁵C as novel and promising anti-cancer drug targets.

We apologize however, that the previous version of our manuscript clearly failed to convey this message. Therefore, we have now extensively re-written our manuscript and in addition, provide new data demonstrating that mitochondrial tRNA methylation directly regulates metastasis capacity of primary tumour cells.

Moreover, we outline below how our study conceptually differs from the above-mentioned publications.

PMID: 33239425: The referee points to a study that identified mitochondria-related genes as the top gene set *associated* with metastasis-specific lethality⁴. Besides the fact that this study does not identify any underlying mechanism, it also does not test whether the observation is relevant for cancer patients. Conceptual differences further include that this study compares non-metastatic to metastatic lung adenocarcinoma cell lines. The authors find that the metastatic cell lines exhibited reduced mitochondrial functions when compared to non-metastatic lines. This is the exact opposite to our findings when analysing intrinsic subpopulations of primary tumours formed by three different OSCC

cancer lines *in vivo*. Finally, none of our observations are caused by cell death, but are due to the intrinsic metabolic plasticity of cancer cells.

PMID: 2524103: We believe the reviewer refers to PMID: 25241037 ⁵. We are aware of this study and it was already cited in our previous version of the manuscript. We refer to our paragraph found in the discussion: “*A direct regulatory role for the mitochondrial RNA modification m5C in determining tumour cell behaviour was unexpected. While mitochondrial and cytosolic translation are known to be rapidly, dynamically and synchronously regulated, it is widely assumed that cytosolic translation processes control mitochondrial translation unidirectionally. For instance, invasive breast cancer cells rely on the transcription coactivator peroxisome proliferator-activated receptor gamma, coactivator 1 alpha (PGC-1α) to enhance oxidative phosphorylation and mitochondrial biogenesis to undergo metastasis*”.

PMID: 33152324: As the authors themselves state, this study is a “*comparative compendium of metabolic essentialities of pancreatic cancer cells grown in culture or as tumors, and reveals potential targets that could be exploited for therapy.*” ⁶ This study does not address the metabolic requirements for metastasis.

PMID: 33152323: Similar to the study above, this manuscript finds a dependency of pancreatic cancer cells on heme synthesis for tumour growth ⁷. This study does also not address the metabolic requirement for metastasis.

In sum, we agree with the referee that there certainly are studies pointing to an important role of mitochondria in metastasis. However, it is currently unknown how these changes in mitochondrial functions are regulated and whether they are cause or consequence of engaging in the metastatic cascade. In contrast, we provide a direct and novel mechanism how mitochondrial functions metabolically reprogram tumour cells to allow metastasis.

Minor points: There should be cDNA rescue for shRNA.

We agree with the referee. Rescue experiments are now provided throughout the manuscript.

References

- 1 Van Haute, L. *et al.* Deficient methylation and formylation of mt-tRNA(Met) wobble cytosine in a patient carrying mutations in NSUN3. *Nat Commun* **7**, 12039, doi:10.1038/ncomms12039 (2016).
- 2 Nakano, S. *et al.* NSUN3 methylase initiates 5-formylcytidine biogenesis in human mitochondrial tRNA(Met). *Nature chemical biology* **12**, 546-551, doi:10.1038/nchembio.2099 (2016).
- 3 Haag, S. *et al.* NSUN3 and ABH1 modify the wobble position of mt-tRNA^{Met} to expand codon recognition in mitochondrial translation. *The EMBO journal*, doi:10.15252/embj.201694885 (2016).
- 4 Chuang, C. H. *et al.* Altered Mitochondria Functionality Defines a Metastatic Cell State in Lung Cancer and Creates an Exploitable Vulnerability. *Cancer Res* **81**, 567-579, doi:10.1158/0008-5472.CAN-20-1865 (2021).
- 5 LeBleu, V. S. *et al.* PGC-1alpha mediates mitochondrial biogenesis and oxidative phosphorylation in cancer cells to promote metastasis. *Nat Cell Biol* **16**, 992-1003, 1001-1015, doi:10.1038/ncb3039 (2014).
- 6 Zhu, X. G. *et al.* Functional Genomics In Vivo Reveal Metabolic Dependencies of Pancreatic Cancer Cells. *Cell metabolism* **33**, 211-221 e216, doi:10.1016/j.cmet.2020.10.017 (2021).
- 7 Biancur, D. E. *et al.* Functional Genomics Identifies Metabolic Vulnerabilities in Pancreatic Cancer. *Cell metabolism* **33**, 199-210 e198, doi:10.1016/j.cmet.2020.10.018 (2021).

Reviewer Reports on the First Revision:

Referees' comments:

Referee #1 (Remarks to the Author):

The authors have addressed my points and have significantly strengthened this really exciting manuscript. I have one remaining minor point that needs to be addressed based on the newly presented data. In my opinion the CD36 overexpression experiment is very important and the authors clearly show that the overexpression does not rescue OCR and does not rescue palmitate induced invasion. However, while in Extended Data Figure 8k, CD36 does not increase sphere growth, it seems to me that in the same panel shNSUN3 does not decrease sphere growth compared to control w/o CD36 OE. Thus it seems no conclusion is possible. The meaning of this experiment should be clarified in the text.

Referee #2 (Remarks to the Author):

I am focusing my attention on the response to my previous comments. I believe the paper is much improved but the interpretation and clinical relevance still has to be toned down substantially.

1. Thank you for the clarification. The data and graphs are much better and the reduction in metastases is now more interpretable. The explanation regarding the heterogenous expression of the shRNA adds a plausible explanation for the varied results. I remain cautious about over interpretation even with data from 3 cell lines. Again, I point out that a 75% to 20% reduction in metastases is an important finding but I can not agree with the language that it is "specifically required"!! For example (top of page 7) these scientific overstatements remain and are simply not acceptable.

2. The results are clear and now more accurately described. Again, I agree that one doesn't expect an "all or nothing" effects but the interpretation and wording should not reflect an "all or nothing" effect.

3. OK

4. Now more clear and their is a nice effect. Again, the language needs to be moderated to reflect the true results.

Referee #3 (Remarks to the Author):

In this revised version the authors have addressed all the points we have raised. The manuscript has further improved and we feel that this study is now a very important contribution to the field.

Referee #4 (Remarks to the Author):

The paper is acceptable conditional on significant editing to make the message clearer and the

interpretation correct with respect to mitochondrial metabolism. In fact, on reading the revision it is clear to this reviewer that the authors have found something super interesting but fail to highlight it. NSUN3 loss does not significantly impact mitochondrial ATP generation rather changes CD36.

Things that must be edited:

(1) “We concluded that NSUN3-mediated tRNA methylation was required to maintain high levels of mitochondrial translation rates and to fuel oxidative phosphorylation”. “A switch to glycolysis explained why primary tumour growth was unaffected by loss of NSUN3.”

-These statements are not supported by their own data. The data in Figure 1E shows that coupled respiration (oligomycin sensitive; ATP generation linked to oxygen consumption) is similar in NSUN3 null cells compared to WT cells. I am not sure how they got figure 1F. I think Figure 1F is maximal respiration, which is different. NSUN3 does not limit basal or coupled respiration but maximal respiration. The important piece is coupled respiration (ATP generation) that is not different!

(2) The connection to CD36 is the most interesting aspect. This should be highlighted in abstract and in the paper. How NSUN3 KD was sufficient to down-regulate CD44 and CD36 is super interesting!!! It is not due to any bioenergetic defects as noted above. It is likely changes in ROS, mtDNA or TCA cycle metabolite release that serves as a signaling molecule.

(3) This paragraph and data should be removed. “Metabolic plasticity allows fast cellular adaptations to cues and stresses in the tumour environment. To understand how dynamic the switch between OXPHOS and glycolysis is, we cultured flow sorted tumour cells according to high (H) and low (L) mitochondrial membrane potential (MMP) as tumouroids (Figure 4q; Figure S9a-d). Tumouroids derived from MMP-high or -low cells were comparable in size (Figure S9a-d).” when we re-analysed the mitochondria’s membrane potential in the cultured tumouroids, we measured no difference in their MMP (Figure S9a-f). To test how fast the tumour cells re-established their characteristic metabolic profile, we flow sorted tumour cells, and then analysed their membrane potential in a time course (Figure 4q, r). The tumour cells re- established their original proportion of cells using OXPHOS or glycolysis in less than 48 hours (Figure 4r)

-This paragraph above and data do not add any value and is an artifact of the culture conditions. Once you remove cells from in vivo environment then cells adapt to high glucose culture conditions. In the future, they need to do these sorts of experiment in human or mouse plasma like media. For example, traditional media is abundant with glucose and not contain lactate or ketone bodies that promote oxidative phosphorylation.

(4) Please put the doxycycline data as supplementary data. Please tone down this aspect. It is not that helpful for the paper. There are lot of papers that have targeted mitochondrial DNA translation or transcription to diminish both primary tumors and metastasis. It is not needed for the central message.

Referee #5 (Remarks to the Author):

In the revised version of their manuscript, Delaunay et al describe involvement of mitochondrial 5-

methylcytosine transferase NSUN3 in metastatic propagation of cancer cells. This is a great study in the field, even if multiple metabolic diseases have been already associated to NSUN3 deficiency. There is still one major point to be clarified.

NSUN3 converts C34 into m5C34 which is further oxidized to f5C34 in the second reaction catalyzed by ABH1 (ALKBH1). This latter modification (f5C34) was shown to be required for correct AUA and AUG decoding. In the previous studies, modification analysis of mito tRNAMet was also performed by bisulfite sequencing with the conclusion that about 1/3 of C34 is unmodified, 1/3 is converted to m5C and the last 1/3 is f5C (PMID: 27356879). Only m5C is resistant to bisulfite deamination and can be detected, C34 and f5C34 are converted to U and not visible. From the BS data shown in Figure 1A one can conclude that about 80% of C34 is converted to m5C (which is deamination-resistant) only minor fraction is C34 or f5C34, which is really necessary for decoding. Does this mean that ABH1 activity is already compromised in the cell lines used for this study? I believe that the initial modification status of mito tRNAMet should be better characterized, as well as the residual level of f5C34 in NSUN3 shRNA treated cells. Figure 1A shows the residual m5C34 level, but this may not be directly correlated with f5C34 level and only f5C34 nucleotide is really crucial for mitochondrial translation. Authors may consider alternative variants of BS, like fCAB-Seq and RedBS-Seq which were also used to distinguish different modified species in related studies. In the same line, in figure 1A authors give results for only 15 sequenced plasmids (out of 25 used?) and this low number is largely insufficient for reliable statistics. Why only low-throughput BS was used here, the use of deep-sequencing is a common standard in the field now.

Related to this point, some expressions used in the text should be clarified. It is stated in the introduction (I45-50) "The biogenesis of f5C34 is initiated through formation of 5-methylcytosine (m5C) by NSUN3 and completed by ALKBH1", this is correct. However, in the description of the results (I75 and I80) "we asked whether loss of m5C decreased global mitochondrial translation rates..." and "The metabolic consequence of reduced m5C levels in mt-tRNAMet was a lower oxidative phosphorylation...". These statements lead to misinterpretation of the m5C role here. Residual m5C34 level only defines potential f5C34 level in tRNA, unless authors provide experimental data that m5C34 can efficiently replace f5C34 in translation.

Minor point:

Cloverleaf structure of tRNA in Figure 1A has only 6 basepaired nucleotides in the CCA stem. This is indeed the case of initiator tRNAMet in many species, but mito tRNAMet used here has classical number of basepairs. This should be corrected.

Nature 2020-10-19395A-Z

In response to referees' comments, we have made the following changes to our manuscript.

Reviewers' comments:

Reviewer #1:

The authors have addressed my points and have significantly strengthen this really exciting manuscript. I have one remaining minor point that needs to be addressed based on the newly presented data. In my opinion the CD36 overexpression experiment is very important and the authors clearly show that the overexpression does not rescue OCR and does not rescue palmitate induced invasion. However, while in Extended Data Figure 8k, CD36 does not increase sphere growth, it seems to me that in the same panel shNSUN3 does not decrease sphere growth compared to control w/o CD36 OE. Thus it seems no conclusion is possible. The meaning of this experiment should be clarified in the text.

We thank the referee for finding our manuscript really exciting.

The author is correct, both CD36-overexpression and NSUN3-depletion do not alter sphere growth (now *Figure S9k* and *Figure S6e-l*). These data are in line with our finding that also primary tumour growth is largely unaffected by depletion of NSUN3 (*Figure 3a* and *Figure S4a*). In conclusion, NSUN3-dependent mitochondrial functions drive metastasis but not primary tumour growth *in vitro* and *in vivo*.

As requested by the referee, we have edited the text for clarity.

Referee #2:

I am focusing my attention on the response to my previous comments. I believe the paper is much improved but the interpretation and clinical relevance still has to be toned down substantially.

As the referee requested, we have further toned down the interpretation and clinical relevance in the abstract and results section.

1. Thank you for the clarification. The data and graphs are much better and the reduction in metastases is now more interpretable. The explanation regarding the heterogenous expression of the shRNA adds a plausible explanation for the varied results. I remain cautious about over interpretation even with data from 3 cell lines. Again, I point out that a 75% to 20% reduction in metastases is an important finding but I can not agree with the language that it is "specifically required"!! For example (top of page 7) these scientific overstatements remain and are simply not acceptable.

We apologize that our language in the previous version of our manuscript lacked precision. To address the referee's concern, we now refer to effect sizes rather than making general statements in the results section.

2. The results are clear and now more accurately described. Again, I agree that one doesn't expect an "all or nothing" effects but the interpretation and wording should not reflect an "all or nothing" effect.

We fully agree with the referee, and we have edited the results section demonstrating that CD36-signalling involves the mitochondrial tRNA modifications accordingly.

3. OK

4. Now more clear and their is a nice effect. Again, the language needs to be moderated to reflect the true results.

We agree with the referee that the antibiotic treatments have a nice effect on tumour metastasis.

To address the referee's concern that the language needed to be moderated, we have edited the respective results section and now refer to effect sizes instead of making general statements.

Referee #3:

In this revised version the authors have addressed all the points we have raised. The manuscript has further improved and we feel that this study is now a very important contribution to the field.

We thank the referees for finding that our study is an important contribution to the field.

Referee #4:

The paper is acceptable conditional on significant editing to make the message clearer and the interpretation correct with respect to mitochondrial metabolism. In fact, on reading the revision it is clear to this reviewer that the authors have found something super interesting but fail to highlight it. NSUN3 loss does not significantly impact mitochondrial ATP generation rather changes CD36.

We are very glad to hear that the referee now finds our study super interesting, and we apologise that our previous version of the manuscript lacked clarity.

As the referee requested, we have added new data and edited the manuscript to better highlight the connection between NSUN3 and CD36.

Things that must be edited:

(1) “We concluded that NSUN3-mediated tRNA methylation was required to maintain high levels of mitochondrial translation rates and to fuel oxidative phosphorylation”. “A switch to glycolysis explained why primary tumour growth was unaffected by loss of NSUN3.”-These statements are not supported by their own data. The data in Figure 1E shows that coupled respiration (oligomycin sensitive; ATP generation linked to oxygen consumption) is similar in NSUN3 null cells compared to WT cells. I am not sure how they got figure 1F. I think Figure 1F is maximal respiration, which is different. NSUN3 does not limit basal or coupled respiration but maximal respiration. The important piece is coupled respiration (ATP generation) that is not different!

As requested by the referee we have edited the text for clarity and emphasize that maximal respiration was limited in the absence of NSUN3 (Figure 2).

We agree with the referee that in steady state mitochondrial ATP generation was not deleteriously limited by deletion of NSUN3 (Figure 2). We edited the text accordingly and provide all corresponding p-values in the Source Data File (see tables Fig. 2d,e, Fig. S2j-o). Because the effect on coupled respiration varied between the cancer lines and even within one cancer line in the seahorse assay (now Figure 2d,e; Figure S2j,k), we now measured ATP production directly. We confirmed a slight but significant reduction of mitochondrial ATP production in all NSUN3-deficient conditions, while ATP production from glycolysis was enhanced (Figure 2g). We conclude that NSUN3-deficient cancer cells mainly rely on glycolysis for energy production (Figure 2f,g; Fig. 3d,e,h; Figure S2g,l-o; Figure S4g) and adapted mitochondrial function and structure (Figure 2h-k; Figure S3f-k) without affecting cell viability and cell growth (Figure 5p-r; Figure S3a and Figure 3a; Figure S4a; Figure S6e-k; Figure S9k).

In addition, we have now re-written our manuscript to better highlight our main finding that loss of NSUN3 reduced mRNA translation of mitochondrial encoded OXPHOS proteins (Figure 2a,b and Figure 3f-h; Figure S2a and Figure S5g-i). Because the levels of m⁵C and f⁵C in mt-tRNA^{Met} are rate limiting for mitochondrial encoded subunits of the OXPHOS complex, the metabolic plasticity of cancer cells in response to CD36-signalling is impaired.

(2) The connection to CD36 is the most interesting aspect. This should be highlighted in abstract and in the paper. How NSUN3 KD was sufficient to down-regulate CD44 and CD36 is super interesting!!! It is not due to any bioenergetic defects as noted above. It is likely changes in ROS, mtDNA or TCA cycle metabolite release that serves as a signaling molecule.

We fully agree with the referee, the fact that NSUN3-depletion alone was sufficient to down-regulate CD36 and CD44 is super interesting. As requested by the referee, we now highlighted the connection between m⁵C and CD36 in the title and the abstract. We have now also made this connection clearer in the results section.

We thank the referee for the very valuable suggestions how NSUN3 KD may down-regulate CD44 and CD36. We propose that the mitochondrial tRNA modifications (m⁵C and f⁵C) are themselves limiting CD36-driven metastasis for the following reasons. (i) Depletion of NSUN3 does not cause changes in mtDNA levels (Figure S3b). (ii) ROS production remains either unchanged or decreased in the absence of NSUN3 (Figure

S3c), as expected from previous studies (Trixl et al., 2018). (iii) Overall TCA cycle metabolites decreased in methylation-deficient cells (Figure 2c). In conclusion, the mitochondrial tRNA modifications m⁵C and f⁵C themselves act as a molecular rheostat orchestrating mitochondrial functions in response to CD36 signalling.

(3) This paragraph and data should be removed. “Metabolic plasticity allows fast cellular adaptations to cues and stresses in the tumour environment. To understand how dynamic the switch between OXPHOS and glycolysis is, we cultured flow sorted tumour cells according to high (H) and low (L) mitochondrial membrane potential (MMP) as tumouroids (Figure 4q; Figure S9a-d). Tumouroids derived from MMP-high or -low cells were comparable in size (Figure S9a-d).” when we re-analysed the mitochondria’s membrane potential in the cultured tumouroids, we measured no difference in their MMP (Figure S9a-f). To test how fast the tumour cells re-established their characteristic metabolic profile, we flow sorted tumour cells, and then analysed their membrane potential in a time course (Figure 4q, r). The tumour cells re-established their original proportion of cells using OXPHOS or glycolysis in less than 48 hours (Figure 4r) -This paragraph above and data do not add any value and is an artifact of the culture conditions. Once you remove cells from in vivo environment then cells adapt to high glucose culture conditions. In the future, they need to do these sorts of experiment in human or mouse plasma like media. For example, traditional media is abundant with glucose and not contain lactate or ketone bodies that promote oxidative phosphorylation.

We agree with the referee that these experiments are not essential for the main message of our manuscript, and we have deleted the paragraph.

(4) Please put the doxycycline data as supplementary data. Please tone down this aspect. It is not that helpful for the paper. There are lot of papers that have targeted mitochondrial DNA translation or transcription to diminish both primary tumors and metastasis. It is not needed for the central message.

As the referee requested, we have now toned down the doxycycline data, and we apologize that our revised version still lacked clarity why we performed the antibiotic experiments.

We fully agree with the referee that there are plenty of studies targeting mitochondrial transcription and protein synthesis to diminish most steps of carcinogenesis such as tumour initiation, progression and metastasis; and that this is indeed not our central message. We now make this clear at the beginning of the corresponding results section in the revised version of our manuscript (page 16; lines 350-357).

However, in stark contrast to these studies, we find that loss of NSUN3 only reduced tumour metastasis without affecting cell viability or primary tumour initiation or growth. Since reducing mitochondrial translation by inhibiting m⁵C and f⁵C formation in mt-tRNA^{Met} did not affect tumourigenesis more generally, we needed to exclude that other, unknown mitochondrial unrelated functions of NSUN3 contributed to the tumour cell invasion and metastasis phenotype.

We speculated that if regulating mitochondrial translation by modifying mt-tRNA^{Met} was indeed the only function of NSUN3 within the metastatic cascade, then we should be able to recapitulate all our observations using pharmacological drugs that specifically inhibit

mitochondrial translation without affecting cytoplasmic protein synthesis. Thus, the antibiotics served as a tool to confirm NSUN3-driven functions in metastasis. Inhibition of mitochondrial translation indeed recapitulated all observed NSUN3-dependent functions: **(i)** reduction of maximal respiration (*Figure S11c,d*), **(ii)** enhanced glucose uptake (*Figure S11g,h*); **(iii)** reduction of invasion (*Figure 7b,c*; *Figure S11e,f*), **(iv)** down-regulation of CD36 protein expression (*Figure S11i*), **(v)** decreased number of CD36-driven metastasis-initiating tumour cells (*Figure S11j-m*), and consequently **(vi)** inhibition of metastasis into the lymph nodes *in vivo* (*Figure 7d-f*).

In summary the antibiotic treatment experiments are important for two reasons: **(1)** We confirm that inhibition of mitochondrial translation fully recapitulated NSUN3 loss-of-function *in vitro* and *in vivo*. **(2)** We also confirmed that inhibition of mitochondrial mRNA translation after the tumour has been formed still reduced metastasis. This is important because we further excluded that differences in primary tumour development and growth contributed to impaired metastasis. We have now edited the text for clarity.

Referee #5:

In the revised version of their manuscript, Delaunay et al describe involvement of mitochondrial 5-methylcytosine transferase NSUN3 in metastatic propagation of cancer cells. This is a great study in the field, even if multiple metabolic diseases have been already associated to NSUN3 deficiency.

We thank the referee for pointing out that our study is a great contribution to the field.

There is still one major point to be clarified.

NSUN3 converts C34 into m5C34 which is further oxidized to f5C34 in the second reaction catalyzed by ABH1 (ALKBH1). This latter modification (f5C34) was shown to be required for correct AUA and AUG decoding. In the previous studies, modification analysis of mito tRNAMet was also performed by bisulfite sequencing with the conclusion that about 1/3 of C34 is unmodified, 1/3 is converted to m5C and the last 1/3 is f5C (PMID: 27356879). Only m5C is resistant to bisulfite deamination and can be detected, C34 and f5C34 are converted to U and not visible. From the BS data shown in Figure 1A one can conclude that about 80% of C34 is converted to m5C (which is deamination-resistant) only minor fraction is C34 or f5C34, which is really necessary for decoding. Does this mean that ABH1 activity is already compromised in the cell lines used for this study? I believe that the initial modification status of mito tRNAMet should be better characterized, as well as the residual level of f5C34 in NSUN3 shRNA treated cells. Figure 1A shows the residual m5C34 level, but this may not be directly correlated with f5C34 level and only f5C34 nucleotide is really crucial for mitochondrial translation. Authors may consider alternative variants of BS, like fCAB-Seq and RedBS-Seq which were also used to distinguish different modified species in related studies. In the same line, in figure 1A authors give results for only 15 sequenced plasmids (out of 25 used?) and this low number is largely insufficient for reliable statistics. Why only low-throughput BS was used here, the use of deep-sequencing is a common standard in the field now.

The referee is correct, it has previously been shown that C34 in the mitochondrial tRNA^{Meth} is about 30% unmodified, 30% m⁵C modified and 30% f⁵C modified (Van

Haute et al., 2016). In contrast, we find 50 to 70% m⁵C in tRNA^{Met} depending on the cell line (*Figure 1c-n; Figure S1g*). However, the modification levels vary quite substantially between studies and may be cell type-specific (see table below) (Haag et al., 2016; Nakano et al., 2016; Trixl et al., 2018; Van Haute et al., 2016).

	Cell line	C unmodified	m ⁵ C (hm ⁵ C)	f ⁵ C
Haag et al., 2016	Hela cells	0%	40%	60%
Nakano et al., 2016	HEK293T cells	0%	0%	100%
Van Haute et al., 2016	Human Fibroblasts	32%	30%	38%
Trixl et al., 2018	ES cells	15%	25%	60%

To address the referee's concern that the high level of m⁵C may indicate a loss of f⁵C due to compromised ALKBH1 in cancer cells, we now quantified f⁵C levels by combining fCAB- with bisulfite RNA-sequencing (*Figure 1a-n; Figure S1a-d*). Parallel fCAB and bisulfite sequencing confirmed the overall very high modification levels at C34 mt-tRNA^{Met} (*Figure 1a-j*).

Together, our data reveal f⁵C-modification levels between 25 to 40% in mt-tRNA^{Met} depending on the cell line (*Figure 1c-l*), which is in line with previous studies. In contrast to previous studies, we find 50-75% m⁵C and rather low levels of unmodified cytosines (*Figure 1c-l*). Importantly, we now directly show a residual f⁵C level of only 2% in NSUN3-depleted cancer cells (*Figure 1m,n*). In conclusion, f⁵C is present in cancer cells and is removed by deletion of NSUN3, as expected.

Moreover, we provide new data demonstrating that ALKBH1 activity is not compromised in the cell lines used in this study. (i) RNA expression of both NSUN3 and ALKBH1 is significantly higher in cancer cells than normal skin cells (*Figure S1i*). (ii) RNA expression levels of ALKBH1 remained largely unaffected by depletion of NSUN3 (*Figure S1j*). (iii) Both NSUN3 and ALKBH1 proteins can be detected in the majority of cancer cell lines (*Figure S1k*). (iv) NSUN3 and ALKBH1 RNA levels significantly correlate in tumour samples and normal tissues (*Figure S1l,m*).

While we do not have an explanation why m⁵C is the major modification at mt-tRNA^{Met} in the tested normal and cancer cells, the difference in modification levels are likely to be cell type-specific. As the referee points out, previous studies have been shown that f⁵C34 is important for decoding (Bilbille et al., 2011; Takemoto et al., 2009), by promoting tRNA-binding to AUG and AUA codons. Whether m⁵C34 has similar or complementary functions cannot be entirely ruled out. For precisely that reason, we chose to remove m⁵C by depleting NSUN3 because this will also block f⁵C formation.

Related to this point, some expressions used in the text should be clarified. It is stated in the introduction (145-50) "The biogenesis of f⁵C34 is initiated through formation of 5-methylcytosine (m⁵C) by NSUN3 and completed by ALKBH1", this is correct. However, in the description of the results (175 and 180) "we asked whether loss of m⁵C decreased global mitochondrial translation rates..." and "The metabolic consequence of reduced m⁵C levels in mt-tRNA^{Met} was a lower oxidative phosphorylation...". These statements lead to misinterpretation of the m⁵C role here. Residual m⁵C34 level only defines potential f⁵C34 level in tRNA, unless authors provide experimental data that m⁵C34 can efficiently replace f⁵C34 in translation.

We agree with the referee. We have now clarified in the text that the effects on translation is caused by removal of both modifications.

Minor point:

Cloverleaf structure of tRNA in Figure 1A has only 6 basepaired nucleotides in the CCA stem. This is indeed the case of initiator tRNA^{Met} in many species, but mito tRNA^{Met} used here has classical number of basepairs. This should be corrected.

We apologize for this oversight and have now included the correct structure and sequence of mt-tRNA^{Met} into *Figure 1* and *Figure S1h* according to Suzuki and colleagues (Suzuki et al., 2020).

References

- Bilbille, Y., Gustilo, E.M., Harris, K.A., Jones, C.N., Lusic, H., Kaiser, R.J., Delaney, M.O., Spremulli, L.L., Deiters, A., and Agris, P.F. (2011). The human mitochondrial tRNA^{Met}: structure/function relationship of a unique modification in the decoding of unconventional codons. *J Mol Biol* 406, 257-274.
- Haag, S., Sloan, K.E., Ranjan, N., Warda, A.S., Kretschmer, J., Blessing, C., Hubner, B., Seikowski, J., Dennerlein, S., Rehling, P., et al. (2016). NSUN3 and ABH1 modify the wobble position of mt-tRNA^{Met} to expand codon recognition in mitochondrial translation. *The EMBO journal*.
- Nakano, S., Suzuki, T., Kawarada, L., Iwata, H., Asano, K., and Suzuki, T. (2016). NSUN3 methylase initiates 5-formylcytidine biogenesis in human mitochondrial tRNA(Met). *Nature chemical biology* 12, 546-551.
- Suzuki, T., Yashiro, Y., Kikuchi, I., Ishigami, Y., Saito, H., Matsuzawa, I., Okada, S., Mito, M., Iwasaki, S., Ma, D., et al. (2020). Complete chemical structures of human mitochondrial tRNAs. *Nat Commun* 11, 4269.
- Takemoto, C., Spremulli, L.L., Benkowski, L.A., Ueda, T., Yokogawa, T., and Watanabe, K. (2009). Unconventional decoding of the AUA codon as methionine by mitochondrial tRNA^{Met} with the anticodon f5CAU as revealed with a mitochondrial in vitro translation system. *Nucleic Acids Res* 37, 1616-1627.
- Trixl, L., Amort, T., Wille, A., Zinni, M., Ebner, S., Hechenberger, C., Eichin, F., Gabriel, H., Schoberleitner, I., Huang, A., et al. (2018). RNA cytosine methyltransferase Nsun3 regulates embryonic stem cell differentiation by promoting mitochondrial activity. *Cell Mol Life Sci* 75, 1483-1497.
- Van Haute, L., Dietmann, S., Kremer, L., Hussain, S., Pearce, S.F., Powell, C.A., Rorbach, J., Lantaff, R., Blanco, S., Sauer, S., et al. (2016). Deficient methylation and formylation of mt-tRNA(Met) wobble cytosine in a patient carrying mutations in NSUN3. *Nat Commun* 7, 12039.

Reviewer Reports on the Second Revision:

Referees' comments:

Referee #4 (Remarks to the Author):

I am satisfied.

Referee #5 (Remarks to the Author):

Authors perfectly satisfied my concerns about m5C/f5C levels and provided new data reinforcing their conclusions.

No other comments from my side.

Author Rebuttals to Second Revision:

Responses to the referees

5th May 2022

Nature 2020-10-19395B

We thank the referees for their constructive comments. No further changes needed to be made to our manuscript.

Referees' comments:

Referee #4 (Remarks to the Author):

I am satisfied.

Referee #5 (Remarks to the Author):

*Authors perfectly satisfied my concerns about m5C/f5C levels and provided new data reinforcing their conclusions.
No other comments from my side.*